# FAST STACKELBERG DETERMINISTIC ACTOR-CRITIC

## ABSTRACT

Most advanced Actor-Critic (AC) approaches update the actor and critic concurrently through (stochastic) Gradient Descents (GD), which may be trapped into bad local optimality due to the instability of these simultaneous updating schemes. Stackelberg AC learning scheme alleviates these limitations by adding a compensated indirect gradient terms to the GD. However, the indirect gradient terms are time-consuming to calculate, and the convergence rate is also relatively slow. To alleviates these challenges, we find that in the Deterministic Policy Gradient family, by removing the terms that contain Hessian matrices and adopting the block diagonal approximation technique to approximate the remaining inverse matrices, we can construct an approximated Stackelberg AC learning scheme that is easy to compute and fast to converge. Experiments reveal that ours outperform SOTAs in terms of average returns under acceptable training time.

## 1 INTRODUCTION

Reinforcement learning (RL) has been widely adopted for solving various complicated sequential problems such as games (Mnih et al., 2013) and robotics applications (Kober et al., 2013). In particular, the actor-critic (AC) approach has achieved remarkable performance in many real-world tasks (Arulkumaran et al., 2017). Most AC methods perform stochastic gradient descent on the actor and critic simultaneously. This can be regarded as performing Gradient Descent Ascent (GDA) (Zheng et al., 2021a;b; Wen et al., 2021), which is known to suffer from convergence to limit cycles or bad locally optimal solutions (Wang et al., 2019; Jin et al., 2020; Balduzzi et al., 2018). Moreover, as mentioned in Yang et al. (2019), the GDA training scheme is fragile, which could lead to biased critic and may not converge.

To address these issues, Zheng et al. (2021a) and Wen et al. (2021) reformulate the actor-critic problem as a bilevel optimization problem, and introduce a leader-follower (Stackelberg) training scheme (Fiez et al., 2020). However, this requires the computation of several indirect gradient terms (including the inverse Hessian matrices), which are difficult to compute or accurately approximated. Moreover, empirically, this Stackelberg training scheme converges slower than GDA in the environments where GDA is not trapped into limited cycles or bad local optimums, since the indirect gradient terms pull back the parameters to stay within the best response areas to guarantee that the leader's loss is always small (Wang et al., 2019). To reduce the computational complexity, Shewchuk (2007) introduces the use of conjugate gradient (CG) descent, which simplifies the computation of the approximated inverse Hessian to the solving of a linear system. However, to have good performance, a sufficient number of inner iterations is still required to obtain a good approximation of the Hessian matrices. Empirically, its convergence is still not fast.

Among these actor-critic methods, the deterministic policy gradient (DPG) family (Silver et al., 2014; Lillicrap et al., 2016; Fujimoto et al., 2018; Haarnoja et al., 2018) has some advantages over other methods: if we assume the actor to be the follower, and the critic to be the leader, the gradient of the follower can be divided into several terms, some are closely related to the best response function which slows down the training process. Also, they include Hessian matrices which are hard to compute (We name it as the TD-related term which is formally discussed in Sec. 4). Based on this observation, we propose Fast Stackelberg Deep Deterministic Policy Gradient (FSDDPG), which reduces these TD-related terms to improve the convergence rate and reduce training time. Moreover, together with the block diagonal approximation technique to approximate the remaining matrices, it is possible to further reduce the training complexity from $O(n^3 m^3)$ to $O(n^2 m^2)$, where $n$ is the number of layers and $m$ is the number of neurons in each layer. Experiments reveal that with our approximated methods outperform SOTAs in terms of average returns under acceptable training time.

## 2 RELATED WORKS

*Actor Critic.* The AC methods are widely adopted in solving complicated sequential problems (Sutton & Barto, 2018). Among these methods, the DPG family are prevalent since they can be both off and on policy and able to be deployed in discrete time and continuous time environments, which are efficient and suit various environments (Lillicrap et al., 2016). However, most of the ACs use simultaneously update rule, i.e., the actor and critic update with the same learning rate, which are not always stable. Empirically, the AC may find better solutions when the critic is updated more frequently than the actor (Wen et al., 2021), and we also show it in our experiments in Sec. 6. Theoretically, most of the stable training methods of GDAs are based on the Two Time Scale Updates (TTSU) scheme (Wu et al., 2020; Dalal et al., 2018). Although the simultaneously update rule for AC can also be proven convergence (Fu et al., 2020), it may not suit all the environments since AC (GDA) may trapped into limited cycles and bad local optimality (Zhang et al., 2020; Wang et al., 2019). Regarding TTSU, since it requires the learning rate of the follower to be slower than that of the leader, its convergence rate is slow consequently (Kaledin et al., 2020).

As the AC training schemes can be also represented as a bi-level optimization process, it is naturally to introduce Stackelberg training scheme to AC as the Stackelberg Actor Critic to stabilize training (Zheng et al., 2021a; Wen et al., 2021). However, the gradient of the follower is time consuming to calculate, and the convergence rate is relatively slow. Thus, in many environments, Stackelberg training may perform worse than vanilla AC under acceptable training time (Wen et al., 2021).

*Gradient-based bi-level optimization.* The gradient-based bi-level optimization is used to minimize/maximize a cost function defined in terms of the optimal solution to another cost function (Sinha et al., 2017). Currently, the idea of differential Stackelberg (leader-follower) game is introduced to convert these problems into the differential (Stackelberg) game problems (Fiez et al., 2020). To solve the differential Stackelberg game, the Stackelberg gradient descent is introduced by leveraging the implicit function theorem (Ji et al., 2021; Fiez et al., 2020; Grazzi et al., 2020), which has already be applied into different areas, e.g., GAN (Fiez et al., 2020), hyper-parameter optimization (Lorraine et al., 2020), as well as meta learning (Rajeswaran et al., 2019).

*Approximating (inverse) Hessian.* Calculating the (inverse) Hessian is time-consuming, especially for the deep neural networks (Martens & Grosse, 2015; Botev et al., 2017). To address this challenge, the conjugate gradient approximation (Rajeswaran et al., 2019; Mehra, 1969; Ghadimi & Wang, 2018), block diagonal approximation (Martens & Grosse, 2015; Botev et al., 2017), as well as the approximate Newton methods (Martınez, 2000; Xu et al., 2016) are proposed. Here, we choose the block diagonal approximation, since it is easy to be implemented and fast to compute while maintaining a good performance. Another method to approximate Hessian is the Gauss Newton matrix (Schraudolph, 2002), which is generally used to approximate Hessian for the Mean Squared loss (MSE) (Martens, 2020). Gauss Newton matrix outperforms Hessian matrix for its fast computing, and stability since it is always positive semi-definite (Chen, 2011). However, it still lacks the obvious justification why dropping the Hessian term helps convergence (Martens, 2020).

## 3 PROBLEM FORMULATION

A Markov decision process (MDP) can represented by the tuple $\langle S, A, R, P(S|S, A), \gamma \rangle$, where $s \in S$ is the state, $a \in A$ is the action, $p(s'|s, a)$ is the transition probability, and $r(s, a) \in [R_{\min}, R_{\max}]$ is the bounded reward function. $\rho^\beta(s)$ is the discounted state distribution of $s$ conducted by a behavior policy $\beta$ [1]. $\pi : S \times A \to \Delta(A)$ is the policy, where $\Delta(A)$ is the simplex of action set $A$. $Q^\pi$ is the state-action value function w.r.t. $\pi$: $Q^\pi(s, a) := \mathbb{E}_\pi[\sum_k \gamma^k r_k | s, a]$. Here, we parameterize $Q$ and $\pi$ with $w \in \Omega$ and $\theta \in \Theta$, respectively ($Q_\omega$ and $\pi_\theta$), where $\Omega$ and $\Theta$ are the parameter spaces.

Following (Zheng et al., 2021a), we formulate the actor-critic as a differential Stackelberg game (Fiez et al., 2020):

$$\max_\theta \quad J(\theta, w^*(\theta)) \tag{1}$$
$$\text{s.t.} \quad w^*(\theta) = \arg\min_w L(\theta, w),$$

where $J(\theta, w) = \mathbb{E}_{s \sim \rho(s), a \sim \pi(\cdot|s)}[Q_w^{\pi_\theta}(s, a)]$, $L(\theta, w) = \mathbb{E}_{s \sim \rho(s), a \sim \pi(\cdot|s)}\left[(Q_w^{\pi_\theta}(s, a) - \perp [Q^{\pi_\theta}(s, a)])^2\right]$, which is the temporal difference (TD) error, and $\perp (\cdot)$ is the stop-gradient operator to achieve semi-gradient method (Sutton & Barto, 2018).

---

[1]For compactness, we omit the superscript of $\rho^\beta(s)$ as $\rho(s)$ with a little abuse of notations.

Stackelberg learning (or approximate implicit differentiation) (Fiez et al., 2020; Ji et al., 2021) is a framework for solving the bilevel optimization problems (Ji et al., 2021; Fiez et al., 2020; Grazzi et al., 2020), and can be used on most AC variants (e.g., DDPG (Lillicrap et al., 2016), TD3 (Fujimoto et al., 2018) and SAC (Haarnoja et al., 2018)). Using it on (1), we obtain (Fiez et al., 2020):

$$\theta \leftarrow \theta + \alpha \frac{\partial J(\theta, w^*(\theta))}{\partial \theta}, \quad w \leftarrow w - \alpha_w \frac{\partial L(\theta, w)}{\partial w}, \tag{2}$$

where $\frac{\partial J(\theta, w^*(\theta))}{\partial \theta} = \frac{\partial J(\theta, w)}{\partial \theta} - \left( \frac{\partial^2 L(\theta, w)}{\partial \theta \partial w} \right) \left( \frac{\partial^2 L(\theta, w)}{\partial w^2} \right)^{-1} \frac{\partial J(\theta, w)}{\partial w}$, and $\alpha, \alpha_\omega$ are learning rates for the actor and critic, respectively.[2] Here,

$$\left( \frac{\partial^2 L(\theta, w)}{\partial \theta \partial w} \right) \left( \frac{\partial^2 L(\theta, w)}{\partial w^2} \right)^{-1} \frac{\partial J(\theta, w)}{\partial w} \tag{3}$$

is known as the indirect gradient term (Lorraine et al., 2020).

Empirically, AC with Stackelberg training does not improve the performance of ACs in many environments (Wen et al., 2021). This is also confirmed by our experiments in Sec. 6. This is mainly due to the fact that Stackelberg training converges slowly, and thus it performs worse than vanilla AC when AC does not be trapped into limited cycles and some bad local optimality. Also, inverting the $n \times n$ Hessian matrix in (3) takes $O(n^3)$ time, where $n$ is the matrix size and is equal to the number of neural network parameters. Although Conjugate Gradient (CG) can be used to reduce the computing complexity (Zheng et al., 2021a), it still needs sufficient steps for CG to obtain a precise approximation of the Hessian matrix.

In this paper, we focus on improving Stackelberg learning in DDPG (Lillicrap et al., 2016) and TD3 (Fujimoto et al., 2018). DDPG and TD3 have been widely adopted for solving complicated continuous control environments (Wang et al., 2020). In DPG family, $\pi(a|s)$ degenerates to $\pi(a|s) = \delta(a - \pi_\theta(s))$, where $\pi_\theta(s) : S \rightarrow A$ is the deterministic policy, and $\delta(\cdot)$ is the Dirac-Delta function. $L(\theta, w)$ becomes $\mathbb{E}_{\rho(s)} \left[ (Q_w^{\pi_\theta}(s, a) - \perp [Q^{\pi_\theta}(s, \pi_\theta(s))])^2 \right]$, where $Q_w^{\pi_\theta}(s, a) := Q_w(s, a)$, s.t. $a = \pi_\theta(s)$. $J(\theta, w)$ becomes $\mathbb{E}_{s \sim \rho(s)}[Q_w^{\pi_\theta}(s, a)]$.

Notice that the practical DPG based methods update the critic by using $\hat{L}(\theta, w) = \mathbb{E}_{\rho(s), a \sim \beta} \left[ (Q_w^{\pi_\theta}(s, a) - \perp [Q^{\pi_\theta}(s, a)])^2 \right]$ [3] (Lillicrap et al., 2016; Fujimoto et al., 2018). We follow the critic's loss, and the modified update scheme becomes:

$$\theta \leftarrow \theta + \alpha \frac{\partial J(\theta, w^*(\theta))}{\partial \theta}, \quad w \leftarrow w - \alpha_w \frac{\partial \hat{L}(\theta, w)}{\partial w}, \tag{4}$$

Comparing with Eq. (2), the update scheme for policy $\theta$ maintains the same. The main reason is that $\nabla_\theta \hat{L}(\theta, w) = \mathbb{E}_{\rho(s), a \sim \beta} \left[ \nabla_\theta (Q_w^{\pi_\theta}(s, a) - Q^{\pi_\theta}(s, a))^2 \right]$ is not well-defined when $a \neq \pi(s)$, and thus we still use $L(\theta, w)$ to update the policy.

To help our analysis, we denote algorithms that replace the vanilla policy gradient in Eq. (4) to DPG in DDPG (resp. TD3) as SDDPG and (resp. STD3), respectively.

## 4 PROPOSED METHOD

To alleviate the two challenges in Section 3, we find that the indirect gradient term for the DPG family (Silver et al., 2014) can be divided into several terms. The TD-related terms are closely related to the TD loss of the critic and contain Hessian matrices.

When the TD loss is small, these TD-related terms can be eliminated (Sec. 4.1). As discussed in (Wang et al., 2019), the indirect gradient term pulls the follower to stay along in the best response area, which may hinder the update scheme and slows convergence: Formally speaking, the best response function $b(w) : \Omega \rightarrow \Theta$ is the implicit function defined by $\frac{\partial L(\theta, w)}{\partial w} = 0$. It can be shown that $\nabla_w b(w) = \frac{\partial^2 L(\theta, w)}{\partial \theta \partial w} \frac{\partial^2 L(\theta, w)}{\partial w^2}^{-1}$ (Fiez et al., 2020). Hence, we have $-\alpha \frac{\partial^2 L(\theta, w)}{\partial \theta \partial w} \frac{\partial^2 L(\theta, w)}{\partial w^2}^{-1} \frac{\partial J(\theta, w)}{\partial w} = \nabla_w b(w) \left( -\alpha \frac{\partial J(\theta, w)}{\partial w} \right)$. If $\theta_t \approx b(w_t)$ ($\theta_t$ is closed to the best

---

[2]For simplicity of notations, we write $\partial w^2 := \partial w^\top \partial w$ and $\partial \theta \partial w := \partial \theta^\top \partial w$.

[3]For simplification, we omit $\perp (\cdot)$ in $\perp [Q^{\pi_\theta}(s, a)]$.

response parameters $b(w_t)$ ), then $\theta_t - \nabla_{w_t} b(w_t) \left( \alpha \frac{\partial J(\theta_t, w_t)}{\partial w_t} \right) = b(w_t) - \nabla_{w_t} b(w_t)(w_{t+1} - w_t) \approx b(w_{t+1})$, where $\approx$ is according to the Taylor's series approximation. It means that LHS tries to keep follower in the best response area. Therefore, since the convergence rate is slow with the indirect gradient term; while the training may be not stable without it (Wang et al., 2019), could we find a balance that maintain the good properties for both fast convergence and stability by removing some terms which are closely related to the best response indicator $L(\theta, w)$?

Here, we observe that the TD-related term is a good option because it contains the residue term $Q_w^{\pi_\theta}(s, a) - Q^{\pi_\theta}(s, a)$ that are closely connected to $L(\theta, w)$, which indicates the distance to best response. Thus, removing them could reduce the influence of the best response in indirect gradient to the follower (actor). Another advantage is that since (partial of) the Hessian matrices $\frac{\partial^2 Q_w^{\pi_\theta}(s, a)}{\partial w^2}$ and $\frac{\partial^2 Q_w^{\pi_\theta}(s, a)}{\partial w \partial \theta}$ are hard to compute, reducing them can help improve the computing time.

But, one still needs to compute the inverse matrix, which dominates the time complexity for computing the gradients. To simplify the computation, in Sec. 4.2 we propose to use a block-diagonal approximation to approximate the inverse matrix.

## 4.1 DERIVATIVES OF $\frac{\partial^2 L(\theta, w)}{\partial \theta \partial w}$ AND $\frac{\partial^2 L(\theta, w)}{\partial w^2}$

First, we show how to simplify the indirect gradient term in (3) by proposing the following proposition:

**Proposition 1.** *For SDDPG and STD3, $\frac{\partial^2 L(\theta, w)}{\partial \theta \partial w}$ and $\frac{\partial^2 L(\theta, w)}{\partial w^2}$ can be further explicitly expressed as:*

$$
\frac{\partial^2 L(\theta, w)}{\partial \theta \partial w} = 2\mathbb{E}_{\rho(s)} \left[ \left( \frac{\partial}{\partial w} Q_w^{\pi_\theta}(s, a) \frac{\partial w}{\partial \theta} \right)^\top \left( \frac{\partial}{\partial w} Q_w^{\pi_\theta}(s, a) \right) \right.
$$
$$
\left. + (Q_w^{\pi_\theta}(s, a) - Q^{\pi_\theta}(s, a)) \left( \frac{\partial}{\partial \theta^\top} \frac{\partial}{\partial w} Q_w^{\pi_\theta}(s, a) \right) \right], \tag{5}
$$

$$
\frac{\partial^2 L(\theta, w)}{\partial w^2} = \mathbb{E}_{\rho(s)} \left( 2 \frac{\partial Q_w^{\pi_\theta}(s, a)}{\partial w} \frac{\partial Q_w^{\pi_\theta}(s, a)^\top}{\partial w} + 2 (Q_w^{\pi_\theta}(s, a) - Q^{\pi_\theta}(s, a)) \frac{\partial^2 Q_w^{\pi_\theta}(s, a)}{\partial w^2} \right). \tag{6}
$$

Proofs can be found in Appendix A. Notice that here we use the chain rules of composite functions $Q_w^{\pi_\theta}(s, a)$, which is different from the original DPG form (Silver et al., 2014). Our form is more suitable for our bi-level optimization settings. Discussion can be found in Remark 1 in Appendix A.

Also note that both (5) and (6) contain the residue term $Q_w^{\pi_\theta}(s, a) - Q^{\pi_\theta}(s, a)$, which can be ignored if $\mathbb{E}_{\rho(s)} \left[ (Q_w^{\pi_\theta}(s, a) - Q^{\pi_\theta}(s, a))^2 \right]$ is small. Eqs. (5) and (6) then degenerate to:

$$
\frac{\partial^2 L(\theta, w)}{\partial \theta \partial w} \approx 2\mathbb{E}_{\rho(s)} \left( \frac{\partial}{\partial w} Q_w^{\pi_\theta}(s, a) \frac{\partial w}{\partial \theta} \right)^\top \left( \frac{\partial}{\partial w} Q_w^{\pi_\theta}(s, a) \right) := k_1(\theta, w),
$$

$$
\frac{\partial^2 L(\theta, w)}{\partial w^2} \approx \mathbb{E}_{\rho(s)} \left( 2 \frac{\partial Q_w^{\pi_\theta}(s, a)}{\partial w} \frac{\partial Q_w^{\pi_\theta}(s, a)^\top}{\partial w} \right) := k_2(\theta, w).
$$

Substituting $k_1(\theta, w), k_2(\theta, w)$ into (4), the follower's approximated update scheme becomes[4]:

$$
\theta_{t+1} \leftarrow \theta_t + \alpha \left[ \frac{\partial J(\theta_t, w_t)}{\partial \theta_t} - k_1 k_2^{-1} \frac{\partial J(\theta_t, w_t)}{\partial w_t} \right], \tag{7}
$$

The leader update scheme is still the same as in (4). Intuitively, (7) reduces the TD-related terms which consequently reduce the time from $O(m^3 n^3 T)$ to $O(\max\{m^2 n^2, mno^2\} T)$ to compute the gradients. Details are shown in Sec. 4.3.

Notice that $k_2(\theta, w)$ is also known as the Gauss Newton matrix (Schraudolph, 2002). Besides fast computing, the Gauss Newton matrix is always Positive Semi-Definite (PSD) which helps better convergence comparing with the Hessian term $\frac{\partial^2 L(\theta, w)}{\partial w^2}$, which may not guarantee PSD (Martens, 2020). This gives us a numerical analysis perspective why the our approximation works.

We will show that it improves the convergence rate theoretically and empirically in Section 5.

---

[4]For simplification, we use $k_1$ and $k_2$ to denote $k_1(w, \theta)$ and $k_2(w, \theta)$.

## 4.2 Approximated Inverse $k_2$

$k_2(\theta, w)$ and $k_1(\theta, w)$ do not include any Hessian term, which avoids $O(m^3 n)$ for computing partial of the Hessian matrix. However, computing the gradients of the FSDDPG is still expensive since inverting $k_2$ ($nm \times nm$ size matrix) usually takes $O(m^3 n^3)$ time. [5]

To accelerate the computation of $k_2^{-1}$, we use a block-diagonal approximation scheme:

$k_2 \simeq \hat{k} := \hat{k}_1 \oplus \hat{k}_2 \oplus \ldots \oplus \hat{k}_d$, where $\hat{k}_i$ is the $i$th diagonal block of $k_2$, $\oplus$ is the direct sum operator, and $d \in \mathbb{Z}^+$ is the number of diagonal blocks which can be set manually. Define $o := d|nm$, where $|$ is the integer division operator as the diagonal degree. Also notice that the time complexity of inverting a block diagonal matrix

---

**Algorithm 1** FSDDPG algorithm

---

Randomly initialize $Q(s, a|w)$ and $\pi(a|s, \theta)$.
Initialize target network $Q'$, $\pi'$, and replay buffer $R$.
**for** episode = 1, M **do**
    Using $\pi$ to interact with environment to obtain transitions $(s_t, a_t, r_t, s_{t+1})$.
    $R = R \cup (s_t, a_t, r_t, s_{t+1})$.
    Sample $N$ transitions from $R$.
    Update the critic using $\frac{\partial \hat{L}(\theta, w)}{\partial w}$.
    Update the actor policy using the sampled policy gradient: $\frac{\partial J(\theta, w)}{\partial \theta} - k_1 k_2^{-1} \frac{\partial J(\theta, w)}{\partial w}$
    Update the target networks.
**end for**

---

is equal to the sum of inverting each block (Boyd et al., 2004). Thus the time complexity of inverting the matrix $k_2$ is reduced from $O\left((nm)^3\right)$ to $O\left(d(\frac{nm}{d})^3\right) = O\left(\frac{mn}{o}(o)^3\right) = O\left(mno^2\right)$. [6]

Since inverting $k_2$ dominates the time complexity, when it is reduced, time complexity reduces to $O\left(\max\{m^2 n^2, mno^2\}t\right)$ consequently. More details can be found in Appendix C.

## 4.3 Time complexity for computing the gradient

The time complexity analysis table is shown in Tab. 1. SDDPG is the Stackelberg learning method directly implemented to DDPG (as mentioned in Sec. 3), and SDDPG-CG is the SDDPG with Conjugate Gradient (CG) technique to approximate for the inverse Hessian matrix (Zheng et al., 2021a; Wen et al., 2021). SDDPG-BD is SDDPG with the block diagnal technique.

From empirical studies, we found that taking $o = 1$ (just taking the diagonal) is sufficient to be well-performing. Also, $\epsilon$ is always small to guarantee performance. Thus, in practical implementation, ours is faster than SDDPG-CG.

Notice that the diagonal block technique may not suit vanilla SDDPG well. The main reason is that the term $\frac{\partial^2 L(\theta, w)}{\partial \theta \partial w}$ still contains (partial) of the Hessian which needs $O(n^2 m^3)$ to compute, which have been removed in ours. Thus, the time complexity of SDDPG with diagonal block technique (SDDPG-DB) is larger than ours when $o$ is small. Moreover, the mainstream deep learning framework (e.g., PyTorch (Paszke et al., 2019)) does not support calculate $diag^o(\frac{\partial^2 L(\theta, w)}{\partial \theta \partial w})$ in parallel through GPU, which makes it hard to implement in practice. Details are in Appendix G. Therefore, the diagonal block technique is an approach that design for our reduction.

Our method can also adopt CG to approximate the $k_2^{-1} \frac{\partial J(\theta_t, w_t)}{\partial w_t}$, which solves the linear system $k_2 x = \frac{\partial J(\theta_t, w_t)}{\partial w_t}$, where $x$ is the vector to be solved. We name this method as FSDDPG-CG (ours).

Table 1: Time complexities of calculating the follower's gradient for SDDPG, FSDDPG (ours), SDDPG-CG, and SDDPG-BD. Here, $n$ is the size of layer, and $m$ is the number of neurons in each layer. $T$ is the total number of episodes. $o \in [1, mn]$ is the diagonal degree. $\epsilon$ is the $\epsilon$-optimal solutions for CG.

| | SDDPG | FSDDPG | SDDPG-CG | SDDPG-BD |
|---|---|---|---|---|
| Time complexity | $O\left(m^3 n^3 T\right)$ | $O\left(\max\{m^2 n^2, mno^2\}T\right)$ | $O\left(m^2 n^2 T/\epsilon\right)$ | $O\left(\max\left\{nmo^2, n^2 m^3\right\} T\right)$ |

## 5 Theoretical Analysis

This section focuses on some theoretical results of our FSDDPG.

---

[5] In this subsection, we regards $k_2$ as a $nm \times nm$ matrix since $\theta$ and $w$ are fixed.

[6] We assume $\frac{nm}{d}$ is a interger.

Before we go deeper into our analysis, we define some useful notations that help the analysis:

$$\xi(\boldsymbol{\theta}) := \begin{bmatrix} \frac{\partial -J(\theta,w)}{\partial \theta} - k_1(\boldsymbol{\theta})\, k_2^{-1}(\boldsymbol{\theta})\, \frac{\partial -J(\theta,w)}{\partial w} \\ \tau \frac{\partial \hat{L}(\theta,w)}{\partial w} \end{bmatrix}, \text{ is the dynamics of our FSDDPG, } \tau := \frac{\alpha_w}{\alpha} \text{ is}$$

defined as the time separation.

we also define the Jacobian of the dynamics of our method as:

$$J_{\mathcal{S}}(\boldsymbol{\theta}) := \begin{bmatrix} \frac{\partial}{\partial \theta}\left( \frac{\partial -J(\theta,w)}{\partial \theta} - k_1 k_2^{-1} \frac{\partial -J(\theta,w)}{\partial w} \right) & \frac{\partial}{\partial w}\left( \frac{\partial -J(\theta,w)}{\partial \theta} - k_1 k_2^{-1} \frac{\partial -J(\theta,w)}{\partial w} \right) \\ \tau \frac{\partial^2 \hat{L}(\theta,w)}{\partial \theta \partial w} & \tau \frac{\partial^2 \hat{L}(\theta,w)}{\partial w^2} \end{bmatrix}$$

and $S(\boldsymbol{\theta}^*) := \frac{1}{2}\left( J_{\mathcal{S}}(\boldsymbol{\theta}^*)^\top + J_{\mathcal{S}}(\boldsymbol{\theta}^*) \right)$, where $\boldsymbol{\theta} := [\theta, w]$ ($\boldsymbol{\theta}^* := [\theta^*, w^*]$ is a fixed point), and $T(\boldsymbol{\theta}^*) = (J_{\mathcal{S}}(\boldsymbol{\theta}^*)^\top J_{\mathcal{S}}(\boldsymbol{\theta}^*))$ which helps our analysis.

Now, we prove that our method can converge to a fixed point.

**Theorem 1.** *When $\pi \in C^q(S \otimes \Theta, \mathbb{R})$ and $Q \in C^q(S \otimes A \otimes W, \mathbb{R})$, $q \geq 2$. For a fixed point $\theta^*$ and $w^*$ such that $J_{\mathcal{S}}(\boldsymbol{\theta}^*)^\top + J_{\mathcal{S}}(\boldsymbol{\theta}^*)$ is positive-definite, the our method with learning rate $\alpha = \frac{\sqrt{\upsilon}}{\psi}$ converges locally with a rate of $O\left(\left(1 - \frac{\upsilon}{2\psi}\right)^{T/2}\right)$, where $\upsilon = \lambda_{\min}^2(S(\boldsymbol{\theta}^*))$, $\psi = \lambda_{\max}(T(\boldsymbol{\theta}^*))$.*

The proof sketch follows Theorem 5 in (Fiez et al., 2020). To compare the convergence rate of FSD-DPG (ours) and SDDPG, we firstly define linear actor as $\pi = \varphi(s)\theta$, where $\varphi(s)$ is the embedding (abstract state), $\theta$ is the learnable parameters. The linear critic is $Q(s,a) = \begin{bmatrix} \varphi(s) \\ \varphi(s)\theta \end{bmatrix} w$ , where $w$ is the learnable parameters for critic.

We now show that for linear actor and critic, under some mild assumptions, the convergence rate of FSDDPG is equivalent to or faster than that of SDDPG, which is $O\left(\left(1 - \frac{\hat{\upsilon}}{2\hat{\psi}}\right)^{T/2}\right)$ (The proof is similar to Theorem 1 and we provide details in Appendix D), where $\hat{\upsilon} = \lambda_{\min}^2\left(\hat{S}(\boldsymbol{\theta}^*)\right)$, $\hat{\psi} = \lambda_{\max}\left(\hat{T}(\boldsymbol{\theta}^*)\right)$, and $\hat{S}(\cdot)$ as wells as $\hat{T}(\cdot)$ are defined similar to $S(\cdot)$ and $T(\cdot)$ by replacing $k_1 k_2^{-1}$ with $\left(\frac{\partial^2 L(\theta,w)}{\partial \theta \partial w}\right)\left(\frac{\partial^2 L(\theta,w)}{\partial w^2}\right)^{-1}$ in $J_S(\cdot)$, respectively.

**Theorem 2.** *For linear actor and critic, if $w^\intercal A \begin{bmatrix} \varphi(s) \\ \varphi(s)\theta \end{bmatrix} \succcurlyeq 0$, where $A$ is any Positive Semi Definite (PSD) matrix. Then, the convergence rate of FSDDPG (ours) is faster than or equivalent to that of SDDPG.*

The proof sketch relies on Matrix inequality. The motivation behind this is that we can show that $S(\boldsymbol{\theta}^*)$ (resp. $\left(J_{\mathcal{S}}(\boldsymbol{\theta}^*)^\top J_{\mathcal{S}}(\boldsymbol{\theta}^*)\right)$) with TD-related term minus $S(\boldsymbol{\theta}^*)$ (resp. $\left(J_{\mathcal{S}}(\boldsymbol{\theta}^*)^\top J_{\mathcal{S}}(\boldsymbol{\theta}^*)\right)$) is always PSD (resp. NSD), which can imply the relationship in eigenvalues of $S(\boldsymbol{\theta}^*)$ and $\left(J_{\mathcal{S}}(\boldsymbol{\theta}^*)^\top J_{\mathcal{S}}(\boldsymbol{\theta}^*)\right)$. Notice that since the convergence rate is decided by these eigenvalues (Theorem 1), we can thus obtain the result.

We can use toy example 1 as an application of Theorem 2 (environment setting is in Sec. 6.1 ). After calculation, we find that the the theoretical convergence of SDDPG is near $O(1^{T/2})$, while FSDDPG is $O(0.75^{T/2})$. This coincide with our empirical result that ours is faster than SDDPG. Details are in Appendix D.

The following we show some properties of our method (Details are in Appendix F):

We show the connections between Eq. (2), FSDDPG (ours) and SDDPG:

**Corollary 1.** *If $\langle \theta^*, w^* \rangle$ is the optimal solution for SDDPG. Then, $\langle \theta^*, w^* \rangle$ is also the optimal solutions for Eq. (2) as well as FSDDPG.*

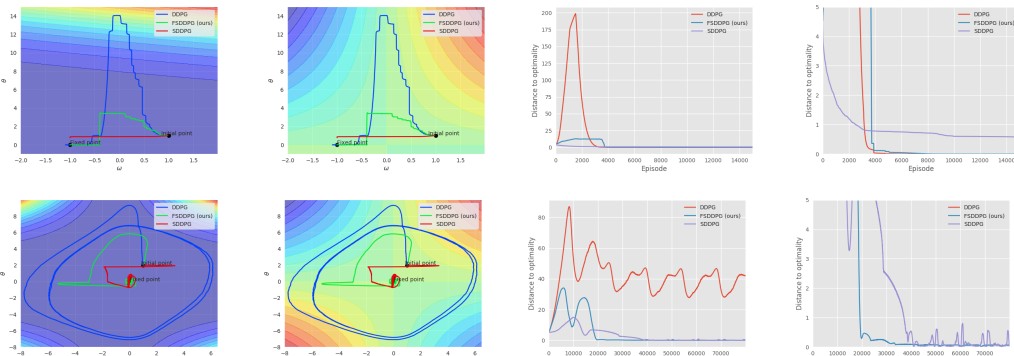

Figure 1: Toy examples. The first row is the toy example 1 and the second row is the toy example 2. The first column is the trajectories of $\theta$ and $w$ (The contours are for $L(\theta, w)$ value). The second column is the trajectories of $\theta$ and $w$ w.r.t different episodes (The contours are for $J(\theta, w)$ value). The third column is the training curves. The forth column is the enlarged part of the training curves.

We also show that our method can avoid strict saddle point almost surely under mild assumption in Appendix F.

## 6 EXPERIMENTS

### 6.1 EXPERIMENTS ON TOY EXAMPLES

*Environment Setup.* Inspired by (Zheng et al., 2021a), we conduct two toy examples to illustrate that our methods are fast convergence and do not being easily trapped into bad local optimality and limited cycles. In the first example, we set both actor and critic as a one-dimension scale $\theta$ and $w$ respectively. The reward is set to $r(\theta) = -\theta^2$. Thus, $\hat{L}(\theta, w) = L(\theta, w) = (w\theta^2 + \theta^2)^2$, $J(\theta, w) = w\theta^2$. The optimal solution of the the first toy example is $\theta^* = 0, w^* = -1$. The second example is similar except that we set $\hat{L}(\theta, w) = L(\theta, w) = (w\theta + \theta^2)^2$, $J(\theta, w) = w\theta$. The optimal solution is $\theta^* = 0, w^* = 0$. The initial point for the two examples are $\theta = 1, w = 1$ and $\theta = 2, w = 2$, respectively. The distance to optimality is defined as $(w - w^*)^2 + (\theta - \theta^*)^2$. Toy example 2 is harder than Toy example 1 since the best response w.r.t $w$ are $w = -1$ and $w = \theta$ respectively, i.e., $w^*$ is a constant in toy example 1, while $w^*$ needs to trace the value of $\theta$ in toy example 2.

*Baselines.* We compare our methods (without block diagonal approximation technique) with for DDPG (Lillicrap et al., 2016), and SDDPG (Zheng et al., 2021a). Notice that due to both the parameters for actor and critic are one-dimension.

*Results.* As shown in Fig. 1, for the first example, DDPG and FSDDPG (ours) converge faster than SDDPG, while DDPG is much more unstable than ours and SDDPG since it moves very far away from the goal. For the second example, DDPG is trapped into the limited cycle, while both FSDDPG (ours) and SDDPG finds the optimal efficiently. Also, ours converges faster than SDDPG. These results reveal that DDPG converges fast yet relative unstable. SDDPG is stable but converges slowly. Ours method combines both the advantages of DDPG and SDDPG, i.e., it converges fast while maintain the stability during training. These results coincide with the assumption that eliminating the TD-related term can improve the convergence rate in Sec. 4.

We also visualize the trajectories on the contours of the $J(\theta, w^*)$ value as in Appendix H. Results find that SDDPG tries to stay in the best response area, while ours are somewhat drifts from best response area, but it returns back only after a few episodes. DDPG could drives very far from the best response area.

### 6.2 EXPERIMENTS ON MUJOCO

*Environment Setup.* We conduct our experiments in standard physical simulator MuJoCo (Todorov et al., 2012), including Ant, HalfCheetah, Humanoid, InvertedDoublePendulum, InvertedPendulum, Swimmer, Reacher, and Walker. The network sizes are $16 \times 16$ for all the methods to make SDDPG and STD3 trainable, larger networks make SDDPG and STD3 hard to train. All the experiments are conducted on a single GeForce RTX 2080 Ti GPU and Intel(R) Xeon(R) CPU E5-2680, and

Table 2: Ablations on running times in different degrees for diagonal block approximation.

| Average Running times (seconds) of each episode | FSDDPG-DEG 1 | FSDDPG-DEG 2 | FSDDPG-DEG 4 |
|---|---|---|---|
| Swimmer | 22.73 | 164.07 | 180.29 |

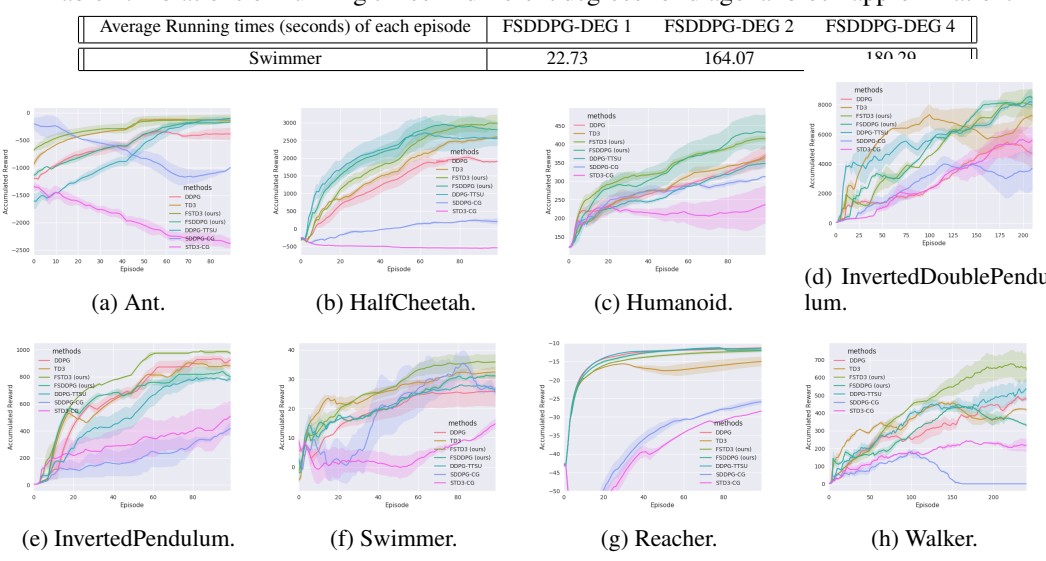

(a) Ant.  (b) HalfCheetah.  (c) Humanoid.  (d) InvertedDoublePendulum.

(e) InvertedPendulum.  (f) Swimmer.  (g) Reacher.  (h) Walker.

Figure 3: Results for various methods on MuJoCo.

implemented by PyTorch (Paszke et al., 2019). More details on the environments and the hyper-parameters can be found in Appendix G.

*Baselines.* We uses DDPG (Lillicrap et al., 2016) TD3 (Fujimoto et al., 2018), DDPG-TTSU, SDDPG-CG, STD3-CG (Wen et al., 2021; Zheng et al., 2021a;b) as our baselines:

1) DDPG is a deep learning implementations of DPG.

2) TD3 is the DDPG method with twined delayed critics to avoid high variance.

3) DDPG-TTSU is similar to DDPG except that the critic is updated much more frequently than actor. In our experiments, we set the frequency ratio as $10 : 1$.

4) SDDPG & STD3 are the Stackelberg learning of DDPG and TD3, respectively.

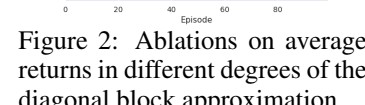

Figure 2: Ablations on average returns in different degrees of the diagonal block approximation.

5) SDDPG-CG & STD3-CG are the SDDPG and STD3 with conjugate gradient to approximate the inverse Hessian, respectively.

Details on the hyper-parameters can be found in Appendix G.

*Results.* As shown in Fig. 3, FSDDPG (ours) reaches the best performance in Humanoid, while FSTD3 (ours) reaches the best performance in 6 out of 8 environments (Ant, HalfCheetah, Inverted-DoublePendulum, InvertedPendulum, Swimmer, and Walker). More specifically, comparing FSD-DPG with DDPG, and FSTD3 with TD3, FSDDPG outperforms DDPG in 6 out of 8 environments (Ant, HalfCheetah, Humanoid, InvertedDoublePendulum, Swimmer, and Walker); FSTD3 outperforms TD3 in all of the environments. Results indicate that our approximated Stackelberg scheme does improve the performance.

For SDDPG-CG and STD3-CG, they do not perform well enough in many environments comparing with ours due to the slow convergence rate. But they could perform better than DDPG and TD3 for some environments that DDPG and TD3 are not converge well (e.g., Swimmer and InvertedDoublePendulum for DDPG).

For DDPG-TTSU, its convergence rate is comparable to FSDDPG and DDPG in all environments. But notice that the critic needs to be updated for ten times in each episode, its actual convergence rate is slower.

We conduct extra experiments in terms of the training times. We use CPUs to testify the training time since precise GPU time per thread is not easy to estimate. As shown in Fig. 4, we find that

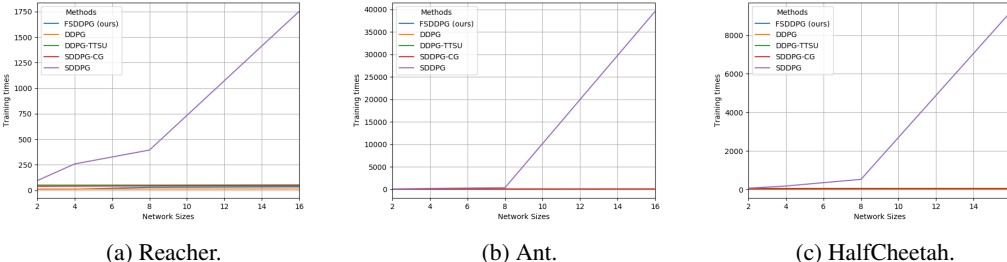

(a) Reacher.      (b) Ant.      (c) HalfCheetah.

Figure 4: Average running time per episode of different methods. The enlarged training time per episode of different methods is shown in Appendix H.

Table 3: Ablations of training time per episode on different Hessian approximation approaches.

| Average training times (seconds) of each episode | FSDDPG (ours) | FSDDPG-CG (ours) | SDDPG-CG | SDDPG-DB | SDDPG |
|---|---|---|---|---|---|
| HalfCheetah | 13.70 | 29.27 | 44.58 | $\geq 2000$ | $\geq 2000$ |
| Inverted Double Pendulum | 12.19 | 28.45 | 43.58 | 446 | 1780 |
| Inverted Pendulum | 15.25 | 30.96 | 43.69 | 339 | 1268 |

the training time of SDDPG is far longer than others. Also, DDPG-TTSU and SDDPG-CG are all requires more training time than ours. This reveals that our methods are fast to compute.

*Ablations.* We firstly compare the performance of different Hessian approximation methods, including FSDDPG-CG and SDDPG-BD (SDDPG with the diagonal block approximation) in Fig. 5. Results show that the ours with CG perform better than ours with DB approximation in 2 out of 3 environments. However, as shown in Tab 3, ours with CG requires twice training times than ours with DB approximation to reach that performance. Thus, we choose the diagonal block approximation as our final method since it balances the tradeoff between times and performance well. Combining Fig. 5 and 3, we find that ours with CG performs much better than SDDPG-CG. Thus, it reveals that the TD-related terms do affect the convergence rate in complicated environments.

In terms of SDDPG-DB, from Tab. 3, we find that SDDPG-DB is still slow in both computing and convergence comparing with FSDDPG, revealing that the block diagonal method may not suit SDDPG well, which testify our claim in Sec. 4.3. Also, regarding the average returns, as shown in Fig. 5, SDDPG outperforms SDDPG-DB. Together with the result of FSDDPG, we conclude that the improvement of performance is mainly based on removing the TD-related terms rather than using the diagonal block approximation.

We also testify how the change of degrees $o$ affects the performance. As shown in Fig. 2 and Tab. 2, the increasing of the degrees costs more training time but leads to better performance. Moreover, we run the SDDPG (without approximation), and the results reveal that SDDPG is relatively slow. Finally, we visualize the losses of critics to see whether the leaders are in best response areas, and the results coincide with the results in Fig. 1. Details are in Appendix H.

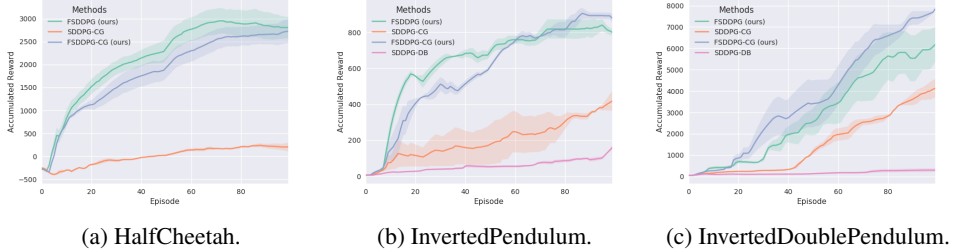

(a) HalfCheetah.      (b) InvertedPendulum.      (c) InvertedDoublePendulum.

Figure 5: Ablations of average returns on different Hessian approximation approaches. SDDPG-CG (ours) does not be conducted in HalfCheetah due to its long training time.

## 7 CONCLUSION

This paper aims to mitigate the challenges of high complexity and slow convergence rate in current Stackelberg actor critic scheme. Specifically, we propose an approximated Stackelberg Deterministic Policy Gradient that removes TD-related terms to improve convergence rate, and together with the block diagonal approximation technique to further reduce the computing times. Experiments show that ours outperform SOTAs in terms of average returns under acceptable training times.

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

## A  DERIVATIVES

This section we discuss the derivatives.

We firstly introduce a lemma that helps our analysis.

**Lemma 1.** *We have $\nabla_\theta \mathbb{E}_{\rho(s)}\left[Q_w^{\pi_\theta}(s, \pi_\theta(s))\right] = \mathbb{E}_{\rho(s)}\left[\nabla_\theta \pi_\theta(s)\left[\nabla_a Q_w^{\pi_\theta}(s, a)\right]_{a=\pi_\theta(s)}\right].$*

*Proof.* By using the Leibniz integral rule to $\nabla_\theta \mathbb{E}_{\rho(s)}\left[Q_w^{\pi_\theta}(s, \pi_\theta(s))\right]$, we have:

$$\nabla_\theta \mathbb{E}_{\rho(s)}\left[Q_w^{\pi_\theta}(s, \pi_\theta(s))\right] = \mathbb{E}_{\rho(s)}\left[\nabla_\theta Q_w^{\pi_\theta}(s, \pi_\theta(s))\right] = \mathbb{E}_{\rho(s)}\left[\nabla_\theta \pi_\theta(s)\left[\nabla_a Q_w^{\pi_\theta}(s, a)\right]_{a=\pi_\theta(s)}\right],$$

which achieves our desired result. □

This lemma reveals that our off-policy gradient is different from the gradient derived in (Silver et al., 2014). Remark 1 discusses why they are different, Lemma 2 discusses when these two methods have the same results.

Now, we derive the derivatives using Lemma 1.

**Proposition 1.** *For SDDPG and STD3, $\frac{\partial^2 L(\theta,w)}{\partial \theta \partial w}$ and $\frac{\partial^2 L(\theta,w)}{\partial w^2}$ can be further explicitly expressed as:*

$$
\begin{aligned}
\frac{\partial^2 L(\theta, w)}{\partial \theta \partial w} &= 2\mathbb{E}_{\rho(s)}\left[\left(\frac{\partial}{\partial w}Q_w^{\pi_\theta}(s, a)\frac{\partial w}{\partial \theta}\right)^\top\left(\frac{\partial}{\partial w}Q_w^{\pi_\theta}(s, a)\right)\right. \\
&\qquad + \left. (Q_w^{\pi_\theta}(s, a) - Q^{\pi_\theta}(s, a))\left(\frac{\partial}{\partial \theta^\top}\frac{\partial}{\partial w}Q_w^{\pi_\theta}(s, a)\right)\right],
\end{aligned}
\tag{8}
$$

$$
\frac{\partial^2 L(\theta, w)}{\partial w^2} = \mathbb{E}_{\rho(s)}\left(2\frac{\partial Q_w^{\pi_\theta}(s, a)}{\partial w}\frac{\partial Q_w^{\pi_\theta}(s, a)^\top}{\partial w} + 2\left(Q_w^{\pi_\theta}(s, a) - Q^{\pi_\theta}(s, a)\right)\frac{\partial^2 Q_w^{\pi_\theta}(s, a)}{\partial w^2}\right).
\tag{9}
$$

*Proof.* Based on the basic technique of matrix derivatives and the integral rule of interchanging the integration and differentiation, we have:

$$
\begin{aligned}
\frac{\partial^2 L(\theta,w)}{\partial w^2} &= \mathbb{E}_{\rho(s)}\left[\frac{\partial}{\partial w^2}\left(Q_w^{\pi_\theta}(s, a) - Q^{\pi_\theta}(s, a)\right)^2\right] \\
&= 2\mathbb{E}_{\rho(s)}\left[\frac{\partial}{\partial w^\top}\left[\left(Q_w^{\pi_\theta}(s, a) - Q^{\pi_\theta}(s, a)\right)\frac{\partial}{\partial w}Q_w^{\pi_\theta}(s, a)\right]\right] \\
&= 2\mathbb{E}_{\rho(s)}\left[\left(\frac{\partial Q_w^{\pi_\theta}(s,a)}{\partial w^\top}\frac{\partial Q_w^{\pi_\theta}(s,a)}{\partial w} + \left(Q_w^{\pi_\theta}(s, a) - Q^{\pi_\theta}(s, a)\right)\frac{\partial^2 Q_w^{\pi_\theta}(s,a)}{\partial w^2}\right)\right]
\end{aligned}
\tag{10}
$$

and

$$
\begin{aligned}
\frac{\partial^2 L(\theta,w)}{\partial \theta \partial w} &= \frac{\partial}{\partial \theta^\top}\mathbb{E}_{\rho(s)}\left[\frac{\partial}{\partial w}\left(Q_w^{\pi_\theta}(s, a) - \perp(Q^{\pi_\theta}(s, a))\right)^2\right] \\
&= 2\frac{\partial}{\partial \theta^\top}\mathbb{E}_{\rho(s)}\left[\left[\left(Q_w^{\pi_\theta}(s, a) - \perp(Q^{\pi_\theta}(s, a))\right)^\top\left(\frac{\partial}{\partial w}Q_w^{\pi_\theta}(s, a)\right)\right]\right] \\
&= 2\mathbb{E}_{\rho(s)}\left[\frac{\partial}{\partial \theta^\top}\left\{\left(Q_w^{\pi_\theta}(s, a) - \perp(Q^{\pi_\theta}(s, a))\right)^\top\left(\frac{\partial}{\partial w}Q_w^{\pi_\theta}(s, a)\right)\right\}\right] \\
&= 2\mathbb{E}_{\rho(s)}\left[\left(\frac{\partial}{\partial \theta^\top}Q_w^{\pi_\theta}(s, a)\right)^\top\left(\frac{\partial}{\partial w}Q_w^{\pi_\theta}(s, a)\right)\right. \\
&\quad + \left.\left(Q_w^{\pi_\theta}(s, a) - Q^{\pi_\theta}(s, a)\right)^\top\left(\frac{\partial}{\partial \theta^\top}\frac{\partial}{\partial w}Q_w^{\pi_\theta}(s, a)\right)\right] \\
&= 2\mathbb{E}_{\rho(s)}\left[\left(\frac{\partial}{\partial \theta^\top}Q_w^{\pi_\theta}(s, a)\right)\left(\frac{\partial}{\partial w}Q_w^{\pi_\theta}(s, a)\right)\right. \\
&\quad + \left.\left(Q_w^{\pi_\theta}(s, a) - \perp(Q^{\pi_\theta}(s, a))\right)\left(\frac{\partial}{\partial \theta^\top}\frac{\partial}{\partial w}Q_w^{\pi_\theta}(s, a)\right)\right] \\
&= 2\mathbb{E}_{\rho(s)}\left[\left(\frac{\partial}{\partial w}Q_w^{\pi_\theta}(s, a)\frac{\partial w}{\partial \theta}\right)^\top\left(\frac{\partial}{\partial w}Q_w^{\pi_\theta}(s, a)\right)\right. \\
&\quad + \left.\left(Q_w^{\pi_\theta}(s, a) - \perp(Q^{\pi_\theta}(s, a))\right)\left(\frac{\partial}{\partial \theta^\top}\frac{\partial}{\partial w}Q_w^{\pi_\theta}(s, a)\right)\right]
\end{aligned}
$$

The last equation is due to the fact that $\frac{\partial}{\partial \theta^\top} Q_w^{\pi_\theta}(s,a) = \left(\frac{\partial}{\partial \theta} Q_w^{\pi_\theta}(s,a)\right)^\top = \left(\frac{\partial}{\partial w} Q_w^{\pi_\theta}(s,a)\frac{\partial w}{\partial \theta}\right)^\top$.

With a little abuse of the notations, for the last equation, we use $Q^{\pi_\theta}(s,a)$ to replace $\perp (Q^{\pi_\theta}(s,a))$.

$\square$

**Remark 1.** *Here, comparing with off-policy DPG, our derived policy gradient is different. In (Silver et al., 2014), under some assumptions (mentioned in Appendix A in (Silver et al., 2014)), the (deterministic) policy gradient is derived from $Q^{\pi_\theta}(s,a)$ as*

$$
\begin{aligned}
\nabla_\theta \mathbb{E}_{\rho(s)}[Q^{\pi_\theta}(s,\pi_\theta(s))] &= \mathbb{E}_{\rho(s)}[\nabla_\theta \int_A [\pi(a|s)Q^{\pi_\theta}(s,a)]da]\\
&= \mathbb{E}_{\rho(s)}[\int_A \nabla_\theta \pi(a|s)Q^{\pi_\theta}(s,a)da + \int_A \pi(a|s)\nabla_\theta Q^{\pi_\theta}(s,a)da]\\
&= \mathbb{E}_{\rho(s)}[\nabla_\theta \pi_\theta(s)\left[\nabla_a Q^{\pi_\theta}(s,a)\right]_{a=\pi_\theta(s)} + \left[\nabla_\theta Q^{\pi_\theta}(s,a)\right]_{a=\pi_\theta(s)}]
\end{aligned}
\tag{11}
$$

*The second equation is based on the integral rule of interchanging integration and differentiation, and the chain rule. The last equation is based on the property of the Dirac-Delta function. Then $Q^{\pi_\theta}(s,\pi_\theta(s))$ is approximated by $Q_w^{\pi_\theta}(s,\pi_\theta(s))$ to make the Q value trackable.*

*For ours, $Q_w^{\pi_\theta}(s,a)$ has been already introduced when the objective is formed. So, by the integral rule of interchanging the integration and differentiation, $\nabla_\theta \mathbb{E}_{\rho(s)}[Q_w^{\pi_\theta}(s,\pi_\theta(s))] = \mathbb{E}_{\rho(s)}[\nabla_\theta Q_w^{\pi_\theta}(s,\pi_\theta(s))] = \mathbb{E}_{\rho(s)}\left[\nabla_\theta \pi_\theta(s)\left[\nabla_a Q_w^{\pi_\theta}(s,a)\right]_{a=\pi_\theta(s)}\right]$. Therefore, our gradient is exact and does not have extra term.*

*The reader may ask: why the two proofs have different results? The main reason is that in the second equation of Eq. (11), in order to use the integral rule of interchanging the integration and differentiation [7], $Q_w^{\pi_\theta}(s,a)$ should be a continuous function w.r.t $\theta$, $\forall\, a \in A$. But according to our definition, it is only differentiable w.r.t $\theta$ when $a = \pi(s)$. Therefore, we can not use the technique as DPG. The following we show that when Q satisfies some conditions, the gap can be bridged:*

**Lemma 2.** *If Q is always a continuous function w.r.t $\theta$, $\forall\, a \in A$, which w.l.o.g. is represented as $Q(s,a,\theta)$. The derivatives of $\mathbb{E}_{\rho(s)}[Q(s,\pi_\theta(s),\theta)]$ derived by (Silver et al., 2014) and ours are equal.*

*Proof.* First of all, following the same steps in off-policy DPG, we have:

$$
\begin{aligned}
\nabla_\theta \mathbb{E}_{\rho(s)}[Q(s,\pi_\theta(s),\theta)] &= \\
&= \mathbb{E}_{\rho(s)}\left[\nabla_\theta \int_A \pi(a|s)Q(s,a,\theta)da\right]\\
&= \mathbb{E}_{\rho(s)}\left[\int_A \nabla_\theta \pi(a|s)Q(s,a,\theta)da + \int_A \pi(a|s)\nabla_\theta Q(s,a,\theta)da\right]\\
&\overset{(a)}{=} \mathbb{E}_{\rho(s)}\left[\int_A \nabla_\theta \pi(a|s)Q(s,a,\theta)da\right] + \mathbb{E}_{\rho(s)}[\nabla_\theta Q(s,a,\theta)]\\
&\overset{(b)}{=} \mathbb{E}_{\rho(s)}\left[\nabla_\theta \pi_\theta(s)\left[\nabla_a Q(s,a,\theta)\right]_{a=\pi_\theta(s)} + \nabla_\theta Q(s,a,\theta)\right]
\end{aligned}
$$

(a) follows the property of Dirac-Delta distribution. (b) follows the the last equation in Eq. (15) in (Silver et al., 2014). The result can also be found in Corollary 3 in (Whiteson, 2020).

For ours, following the chain rule, we have:

$$
\nabla_\theta \mathbb{E}_{\rho(s)}[Q(s,\pi_\theta(s),\theta)] = \mathbb{E}_{\rho(s)}\left[\nabla_\theta Q(s,a,\theta) + \nabla_\theta \pi_\theta(s)\left[\nabla_a Q(s,a,\theta)\right]_{a=\pi_\theta(s)}\right].
$$

Then, we can easily check the two derivatives are equal, which is our desired result. $\square$

Here, we show the reason that the stop-gradient operator is necessary in $L(\theta,w)$: because $\frac{\partial}{\partial \theta}\mathbb{E}_{\rho(s)}\left[(Q_w^{\pi_\theta}(s,a) - Q^{\pi_\theta}(s,\pi_\theta(s)))^2\right]$ may not be well-defined since $\frac{\partial}{\partial \theta}Q^{\pi_\theta}(s,\pi_\theta(s))$ does not exist in our settings.

**Lemma 3.** *$\frac{\partial}{\partial \theta}Q^{\pi_\theta}(s,\pi_\theta(s))$ is not well-defined if $r(s,\pi_\theta(s))$ and $p(s'|s,\pi_\theta(s))$ are non-differentiable.*

---

[7]Details can be found in `https://math.stackexchange.com/questions/2530213/when-can-we-interchange-integration-and-differentiation`

*Proof.* We can rewrite the state-action function under the expectation of rewards for different trajectories: $Q^{\pi_\theta}(s_t, \pi_\theta(s_t)) = \mathbb{E}_\tau[\sum_{k=t}^{\infty} \gamma^k r^k | \tau]$ , where $\tau :=$ $p(s_t | s_t, \pi_\theta(s_t)) \prod_{k=t+1}^{\infty} \pi_\theta(s_k) p(s_{k+1}|s_k, \pi_\theta(s_k)) \in \top$ is the probability of trajectories.

$$d\tau := ds_t \prod_{k=t+1}^{\infty} da_k ds_k$$

Then, we have:

$$Q^{\pi_\theta}(s_t, \pi_\theta(s_t)) = \mathbb{E}_\tau[\sum_{k=t}^{\infty} \gamma^k r^k | \tau]$$

$$= \int_{\mathcal{T}} \sum_{k=1}^{\infty} \gamma^k r^k d\tau$$

$$= \int_{\mathcal{T}} p(s_t|s_t, \pi_\theta(s_t)) \prod_{k=t+1}^{\infty} \pi_\theta(s_k) p(s_{k+1}|s_k, \pi_\theta(s_k)) \left( \gamma^t r^t(s, \pi_\theta(s_t)) + \sum_{k=t+1}^{\infty} \gamma^k r^k \right) d\tau$$

$$= \int_{\mathcal{T}} p(s_t|s_t, \pi_\theta(s_t)) \prod_{k=t+1}^{\infty} \pi_\theta(a_k|s_k) p(s_{k+1}|s_k, a) \left( \gamma^t r^t(s, \pi_\theta(s_t)) + \sum_{k=t+1}^{\infty} \gamma^k r^k \right) d\tau$$

$$= \int_{\mathcal{T}} p(s_t|s_t, \pi_\theta(s_t)) \prod_{k=t+1}^{\infty} \pi_\theta(a_k|s_k) p(s_{k+1}|s_k, a) \left( \gamma^t r^t(s, \pi_\theta(s_t)) + \sum_{k=t+1}^{\infty} \gamma^k r^k \right) d\tau$$

$$\overset{(a)}{=} \int_{\mathcal{T}\backslash A} p(s_{t+1}|s_t, \pi_\theta(s_t)) \prod_{k=t+2}^{\infty} \pi_\theta(a_k|s_k) p(s_{k+1}|s_k, a_k)$$

$$\left( \int_A \pi_\theta(a_{t+1}|s_{t+1}) p(s_{t+2}|s_{t+1}, a_{t+1}) \left( \gamma^t r^t(s, \pi_\theta(s_t)) + \sum_{k=t+1}^{\infty} \gamma^k r^k \right) da_{t+1} \right) d(\tau\backslash a_{t+1})$$

$$\overset{(b)}{=} \int_{\mathcal{T}\backslash A} p(s_{t+1}|s_t, \pi_\theta(s_t)) \left( \prod_{k=t+2}^{\infty} \pi_\theta(a_k|s_k) p(s_{k+1}|s_k, a_k) \right)$$

$$(p(s_{t+2}|s_{t+1}, \pi_\theta(s_{t+1}))) \left( \sum_{k=t}^{t+1} \gamma^k r^k(s, \pi_\theta(s_t)) + \sum_{k=t+1}^{\infty} \gamma^k r^k \right) d\tau\backslash a_{t+1}$$

$$= \int_{\mathcal{T}\backslash \left( \cup_{k=t}^{\infty} A \right)} \prod_{k=t}^{\infty} p(s_{k+1}|s_k, \pi_\theta(s_k)) \left( \sum_{k=t}^{\infty} \gamma^k r^k(s, \pi_\theta(s_t)) \right) d\tau\backslash \left( \cup_{k=t}^{\infty} a_{t+1} \right)$$

(a) is based on the Fubini's theorem.

(b) is due to the property of Dirac-delta function.

$d(\tau\backslash a_{t+1})$ means $ds_t ds_{t+1} ( \prod_{k=t+2}^{\infty} da_k ds_k )$.

$\mathcal{T}\backslash A$ means removing the set $A$ from $\mathcal{T}$.

According to the assumptions, $r(s, \pi_\theta(s))$ and $p(s'|s, \pi_\theta(s))$ are non-differentiable w.r.t $\theta$.

Since $Q^{\pi_\theta}(s_t, \pi_\theta(s_t))$ does not contain any differentiable elements, $\frac{\partial}{\partial\theta} Q^{\pi_\theta}(s, \pi_\theta(s))$ are not well-defined.

$\square$

This Lemma reveals that the $\frac{\partial}{\partial\theta^\top} Q^{\pi_\theta}(s, \pi_\theta(s))$ is not well-defined in many cases, since most of the environments are natively non-differentiable w.r.t the parameters of policy, e.g., (Todorov et al., 2012). That is another reason that we need to use the stop-gradient operator.

## B    BLOCK DIAGONAL MATRICES

We firstly define $E := k_2 diag^o(k_2)^{-1} - I$

We now show a technical lemma here:

**Lemma 4.** *(Henderson & Searle, 1981) For two non-degenerate matrices $A$ and $B$, we always have:*

$$(A+B)^{-1} = A^{-1} - A^{-1}B(A+B)^{-1},$$

Now we show that larger $o$ does not mean that the distance will be smaller:

**Lemma 5.** $D(k_2, diag^o(k_2))$ *is not always monotonic decreased w.r.t. o:*

*Proof.* This can be done by finding a counter-example:

For $k_2 = \begin{bmatrix} 1 & 1 & 1 & 2 \\ 1 & 1 & 1 & 3 \\ 1 & 1 & 9 & 4 \\ 5 & 5 & 6 & 1 \end{bmatrix}$, we have $D\left(k_2, diag^1(k_2)\right) = 11.0$, $D\left(k_2, diag^2(k_2)\right) = 8.2$, $D\left(k_2, diag^3(k_2)\right) = 10.7$. $\qquad\square$

But we can always find a sub-series that satisfies the following Lemma:

**Lemma 6.** *When o can be represented as $lx$, where $x, l \in \mathbb{Z}^+$, $D\left(k_2, diag^{lx}(k_2)\right)$ is a strict decreased series with the increasing of $l$.*

*Proof.* $||k_2 - diag^{lx}(k_2)||_F = \sqrt{\sum_{i,j \in G} |k_{ij}|}$, $G = \{i,j | i \neq lxj + k, \ k = 1, 2 \ldots, lx\}$,

$||k_2 - diag^{(l+1)x}(k_2)||_F = \sqrt{\sum_{i,j \in G_1} |k_{ij}|}$, $G_1 = \{i,j | i \neq (l+1)xj + k, \ k = 1, 2 \ldots, (l+1)x\}$

Since $G \subset G_1$, we have:

$$\sqrt{\sum_{i,j \in G} |k_{ij}|} > \sqrt{\sum_{i,j \in G_1} |k_{ij}|}$$

$||k_2 - diag^{lx}(k_2)||_F - ||k_2 - diag^{(l+1)x}(k_2)||_F < 0.$

Since$||k_2 - diag^{mn}(k_2)||_F = 0$. So $D\left(k_2, diag^{lx}(k_2)\right)$ is a strict decreased series with the increasing of $l$. $\qquad\square$

Therefore, we conclude that we can select a series of $o$ that are have the same common factor larger than 1 to guarantee that the distance is $D\left(k_2, diag^{lx}(k_2)\right)$ is a strict decreased (e.g., $2, 4, 6, \ldots$).

## C  TIME COMPLEXITY FOR COMPUTING THE GRADIENT

We assume that critic has $n$ layers with $m$ neurons, while actor has $n_1$ layers with $m_1$. We also define $t$ as the training episodes.

We define $OP_M$ as the time complexity of multiplication two real numbers, $OP_G$ as the time complexity of calculating the gradient of one element w.r.t other, and $OP_A$ as the time complexity of adding two real numbers.

Now we analyze the one step time complexity of different terms.

1) The time complexity of $\frac{\partial}{\partial w} Q_w^{\pi_\theta}(s, a)$ is

$$OP_G\left(m + \sum_{i=2}^{n+1} m^2\right) + OP_M\left(\sum_{i=1}^{n+1} m^2\right) = OP_G\left(m + nm^2\right) + OP_M\left(n + 1\right)m^2.$$

$m + \sum_{i=2}^{n+1} m^2$ is because the the dimension of gradient between the output layer and the last layer is $m$, and thus the dimension of gradient is $m^2$ for each intermediate layer. $\left(\sum_{i=1}^{n+1} m^2\right)$ is because each layer requires $m^2$ times multiplication to form the chain rules. [8]

---

[8] Recall the time complexity of multiplication two matrices with dimensions $m, n$ and $n, p$ is $mnp$.

The time complexity of $\frac{\partial \pi}{\partial \theta}$ is $\left( OP_G \left(m_1 + n_1 m_1^2\right) + OP_M \left(n_1 + 1\right)m_1^2 \right)$, which follows the same analysis of $\frac{\partial}{\partial w} Q_w^{\pi_\theta}(s, a)$.

Combining the above analysis, the time complexity of $\frac{\partial J(\theta, w)}{\partial w}$ is $= s\left( OP_G \left(m + nm^2\right) + OP_M \left(n + 1\right)m^2 \right)$, which follows the same analysis of $\frac{\partial}{\partial w} Q_w^{\pi_\theta}(s, a)$.

2) The time complexity of $\frac{\partial J(\theta, w)}{\partial \theta}$ is

$$
\begin{aligned}
&\left( OP_G \left(m + nm^2\right) + OP_M \left(n + 1\right)m^2 \right) + (OP_G + OP_M)m \\
&+ \left( OP_G \left(m_1 + n_1 m_1^2\right) + OP_M \left(n_1 + 1\right)m_1^2 \right) + OP_M(mn + m_1 n_1)
\end{aligned}
$$

, which is a combination of the time of calculating $\frac{\partial}{\partial w} Q_w^{\pi_\theta}(s, a)$ (require $\left( OP_G \left(m + nm^2\right) + OP_M \left(n + 1\right)m^2 \right)$ times), $\frac{\partial w}{\partial \pi}$ (require $(OP_G + OP_M)m$ times), and $\frac{\partial \pi}{\partial \theta}$ (require $\left( OP_G \left(m_1 + n_1 m_1^2\right) + OP_M \left(n_1 + 1\right)m_1^2 \right)$ times) (since $\frac{\partial J(\theta, w)}{\partial \theta} \approx \sum \frac{\partial}{\partial w} Q_w^{\pi_\theta}(s, a) \frac{\partial w}{\partial \pi} \frac{\partial \pi}{\partial \theta}$).

3) The time complexity of $\mathbb{E} \frac{\partial Q_w^{\pi_\theta}(s, a)}{\partial \theta}$ is exactly the same as $\frac{\partial J(\theta, w)}{\partial \theta}$.

4) For the time complexity of $\frac{\partial^2}{\partial w^2} Q_w^{\pi_\theta}(s, a)$ is

$$
\begin{aligned}
&OP_G \left(m + nm^2\right) + OP_M \left(n + 1\right)m^2 + nm OP_G \left(m + \sum_{i=2}^{n+1} m^2\right) + nm OP_M \left(\sum_{i=1}^{n+1} m^2\right) \\
&= OP_G \left(m + nm^2\right) + OP_M \left(n + 1\right)m^2 + OP_G \left(nm^2 + n^2 m^3\right) + OP_M \, n(n + 1)m^3,
\end{aligned}
$$

which is based on the fact that the Hessian can be decomposed to $\frac{\partial}{\partial w}[\frac{\partial}{\partial w} Q_w^{\pi_\theta}(s, a)]$.

5) For $\frac{\partial}{\partial \theta^\top} \frac{\partial}{\partial w} Q_w^{\pi_\theta}(s, a)$,

$$
\begin{aligned}
&n_1 m_1 \left( OP_G \left(m + nm^2\right) + OP_M \left(n + 1\right)m^2 \right) + n_1 m_1 (OP_G + OP_M)m \\
&+ n_1 m_1 \left( OP_G \left(m_1 + n_1 m_1^2\right) + OP_M \left(n_1 + 1\right)m_1^2 \right) + OP_M(mn + m_1 n_1)
\end{aligned}
$$

6) The time complexity of $\left( \frac{\partial}{\partial w} Q_w^{\pi_\theta}(s, a) \frac{\partial w}{\partial \theta} \right)^\top \left( \frac{\partial}{\partial w} Q_w^{\pi_\theta}(s, a) \right)$ is the summation of $\frac{\partial}{\partial w} Q_w^{\pi_\theta}(s, a) \frac{\partial w}{\partial \theta}$ and $\frac{\partial}{\partial w} Q_w^{\pi_\theta}(s, a)$, and their multiplication.

$$
\begin{aligned}
&\left( OP_G \left(m + nm^2\right) + OP_M \left(n + 1\right)m^2 \right) + (OP_G + OP_M)m \\
&+ \left( OP_G \left(m_1 + n_1 m_1^2\right) + OP_M \left(n_1 + 1\right)m_1^2 \right) + OP_M(mn + m_1 n_1) \\
&+ \left( OP_G \left(m + nm^2\right) + OP_M \left(n + 1\right)m^2 \right) + OP_M(mn m_1 n_1) \\
&= O \left( \max \left\{ nm^2, mnm_1 n_1, n_1 m_1^2 \right\} \right)
\end{aligned}
$$

7) For $\dfrac{\partial Q_w^{\pi_\theta}(s, a)}{\partial w} \dfrac{\partial Q_w^{\pi_\theta}(s, a)^\top}{\partial w}$

$$
\left( OP_G \left(m + nm^2\right) + OP_M \left(n + 1\right)m^2 \right) + OP_M \left(m^2 n^2\right)
$$

Combining them together, for

$$
\mathbb{E}_{\rho(s)} \left[ \left( \frac{\partial}{\partial w} Q_w^{\pi_\theta}(s, a) \frac{\partial w}{\partial \theta} \right)^\top \left( \frac{\partial}{\partial w} Q_w^{\pi_\theta}(s, a) \right) + \left( Q_w^{\pi_\theta}(s, a) - Q^{\pi_\theta}(s, a) \right) \frac{\partial}{\partial \theta^\top} \frac{\partial}{\partial w} Q_w^{\pi_\theta}(s, a) \right],
$$

the time complexity is:

$$
\begin{aligned}
&\left( OP_G \left(m + nm^2\right) + OP_M \left(n + 1\right)m^2 \right) + (OP_G + OP_M)m \\
&+ \left( OP_G \left(m_1 + n_1 m_1^2\right) + OP_M \left(n_1 + 1\right)m_1^2 \right) + OP_M(mn + m_1 n_1) \\
&+ \left( OP_G \left(m + nm^2\right) + OP_M \left(n + 1\right)m^2 \right) + OP_M(mn m_1 n_1) \\
&+ n_1 m_1 \left( OP_G \left(m + nm^2\right) + OP_M \left(n + 1\right)m^2 \right) + n_1 m_1 (OP_G + OP_M)m \\
&+ n_1 m_1 \left( OP_G \left(m_1 + n_1 m_1^2\right) + OP_M \left(n_1 + 1\right)m_1^2 \right) + OP_M(mn + m_1 n_1) \\
&= O \left( \max \left\{ n_1^2 m_1^3, n_1 m_1 nm^2 \right\} \right)
\end{aligned}
$$

For $\mathbb{E}_{\rho(s)}\left[\left(2\frac{\partial Q_w^{\pi_\theta}(s,a)}{\partial w}\frac{\partial Q_w^{\pi_\theta}(s,a)^\top}{\partial w} + 2\left(Q_w^{\pi_\theta}(s,a) - Q^\pi(s,a)\right)\frac{\partial^2 Q_w^{\pi_\theta}(s,a)}{\partial w^2}\right)\right]$, the time complexity is:

$$\underbrace{\left(OP_G\left(m+nm^2\right) + OP_M\left(n+1\right)m^2\right) + OP_M\left(m^2n^2\right)}_{calculate\ k_2}$$

$$+\underbrace{OP_G\left(m+nm^2\right) + OP_M\left(n+1\right)m^2 + OP_G\left(nm^2+n^2m^3\right) + OP_M\,n(n+1)m^3}_{calculate\ \kappa_2},$$

$$= O\left(n^2m^3\right)$$

For inverse $k_2 + \kappa_2$, the time complexity is:

$$\underbrace{\left[OP_G\left(m+nm^2\right) + OP_M\left(n+1\right)m^2\right] + s\left[OP_M n_1^2 m_1^2\right]}_{calculate\ k_2}$$

$$+\underbrace{\left[OP_G\left(m+nm^2\right) + OP_M\left(n+1\right)m^2 + OP_G\left(m^2+nm^3\right) + OP_M\left(n+1\right)m^3\right]}_{calculate\ \kappa_2}$$

$$+\underbrace{O\left(n^3m^3\right)}_{inverse\ matrix}$$

$$= O\left(n^3m^3\right)$$

For inverting $diag^o(k_2)$, the time complexity is:

$$\underbrace{\left[OP_G\left(m+nm^2\right) + OP_M\left(n+1\right)m^2\right] + s\left[OP_M n_1^2 m_1^2\right] + OP_G m^2}_{calculate\ part\ of\ k_2} + \underbrace{O\left(o\frac{n^3m^3}{o^3}t\right)}_{inverse\ matrix} + \underbrace{O\left(o\left(\frac{nm}{o}\right)^2 t\right)}_{extract\ the\ diag^o(k_2)}$$

$$= O\left(\max\left\{nm^2, n_1 m_1^2, \frac{n^3m^3}{o^2}\right\}t\right)$$

For inverting $diag^o(k_2 + \kappa_2)$

$$\underbrace{\left[OP_G\left(m+nm^2\right) + OP_M\left(n+1\right)m^2\right] + s\left[OP_M n_1^2 m_1^2\right] + OP_G m^2}_{calculate\ part\ of\ k_2} +$$

$$+\underbrace{\left[OP_G\left(m+nm^2\right) + OP_M\left(n+1\right)m^2\right] + \left[OP_M n_1^2 m_1^2\right] + OP_G m}_{calculate\ part\ of\ \kappa_2} + \underbrace{O\left(k\frac{n^3m^3}{k^3}t\right)}_{inverse\ matrix} + \underbrace{O\left(k\left(\frac{nm}{k}\right)^2 t\right)}_{extract\ the\ diag^o(k_2)}$$

$$= O\left(\max\left\{nm^2, n_1 m_1^2, \frac{n^3m^3}{k^2}\right\}t\right)$$

Therefore, the time complexity of calculating the actor for SDDPG in Eq. (4) is

$$O\left(\underbrace{\max\left\{nm^2, n_1 m_1^2\right\}t}_{\frac{\partial Q_w^{\pi_\theta}(s,a)}{\partial\theta}and\frac{\partial Q_w^{\pi_\theta}(s,a)}{\partial w}} + \underbrace{\left(m_1 n_1 m^2 n^2\right)t + (m_1 n_1 mn)t}_{multiply\ matrices} + \underbrace{n^3m^3 t + n_1 m_1^2 t}_{get\ k_1 + \kappa_1}\right) =$$

$$O\left(\max\left\{n^3m^3, n_1 m_1^2, m_1 n_1 m^2 n^2\right\}t\right)$$

The complexity of FSDDPG (ours) is

$$O\left(\underbrace{\max\left\{nm^2, n_1 m_1^2\right\}t}_{\frac{\partial Q_w^{\pi_\theta}(s,a)}{\partial\theta}\ and\ \frac{\partial Q_w^{\pi_\theta}(s,a)}{\partial w}} + \underbrace{m^2 n^2 t + \left(om^2 n^2\right)t}_{multiply\ matrices} + \max\left\{nm^2, n_1 m_1^2, o^3\right\}t\right)$$

$$= O\left(\max\left\{n_1^2 m_1^2, m_1 n_1 mn, nm^2, m^2 n^2 o, o^3\right\}t\right)$$

The time complexity of DPG is:

$$O\left(\underbrace{\max\left(nm^2t, n_1m_1^2t\right)}_{\frac{\partial Q_w^{\pi_\theta}(s,a)}{\partial\theta}}\right).$$

The time complexity of SDDPG-CG is:

$$O\left(\underbrace{\max\left\{nm^2, n_1m_1^2\right\}t}_{\frac{\partial Q_w^{\pi_\theta}(s,a)}{\partial\theta}\ and\ \frac{\partial Q_w^{\pi_\theta}(s,a)}{\partial w}} + \underbrace{(m_1n_1mn)\,t}_{multiply\ matrices} + \underbrace{n^2m^2pt}_{get\ inverse\ CG}\right) =$$
$$O\left(\max\left\{n^2m^2p, n_1m_1^2, m_1n_1mn\right\}t\right)$$

where $p$ is the number of iterations of CG in each episode, which is $O(1/\epsilon)$, where $\epsilon$ is the error of CG.

The time complexity of SDDPG-BD (SDDPG with block diagonal approximation) is:

$$O(\underbrace{\max\left\{nm^2, n_1m_1^2\right\}t}_{\frac{\partial Q_w^{\pi_\theta}(s,a)}{\partial\theta}\ and\ \frac{\partial Q_w^{\pi_\theta}(s,a)}{\partial w}} + \underbrace{(mn)t + \left(k\left(\frac{nm}{k}\right)^2\times nm\right)t}_{multiply\ matrices} + \underbrace{\max\left\{nm^2, n_1m_1^2, \frac{n^3m^3}{k^2}\right\}t}_{get\ block\ diagnal\ of\ k_2+\kappa_2}$$
$$+ \underbrace{O\left(\max\left\{n_1^2m_1^3, n_1m_1nm^2\right\}\right)}_{get\ k_1+\kappa_1}) =$$
$$O\left(\max\left\{n_1m_1nm^2, nm^2, \frac{n^3m^3}{o^2}, n_1^2m_1^3\right\}t\right)$$

if $m_1 = m$, $n_1 = n$:

The time complexity of SDDPG is:

$$O\left(\max\left\{n^3m^3, n_1m_1^2, m_1n_1m^2n^2\right\}t\right) = O\left(m^3n^3t\right)$$

The time complexity of FSDDPG (ours) is:

$$O\left(\max\left\{n_1^2m_1^2, m_1n_1mn, nm^2, mno^2, o^3\right\}t\right) = O\left(\max\{m^2n^2, mno^2\}t\right)$$

The time complexity of DDPG is:

$$O\left(\max\left\{n_1^2m_1^2, nm^2\right\}t\right) = O\left(nm^2t\right)$$

The time complexity of SDDPG-CG is:

$$O\left(\max\left\{n^2m^2p, n_1m_1^2, m_1n_1mn\right\}t\right) = O\left(m^2n^2\frac{1}{\epsilon}t\right)$$

The time complexity of SDDPG-DB is:

$$O\left(\max\left\{\frac{n^3m^3}{k^2}, n_1m_1^2, n_1^2m_1^3, n_1m_1nm^2\right\}t\right) = O\left(\max\left\{nmo^2, n^2m^3\right\}t\right)$$

## D CONVERGENCE RATE

This section we discuss the convergence rate.

We firstly show the relationship between SDDPG, FSDDPG, and Eq. (2):

**Corollary 1.** *If $\langle\theta^*, w^*\rangle$ is the optimal solution for SDDPG. Then, $\langle\theta^*, w^*\rangle$ is also the optimal solutions for Eq. (2) as well as FSDDPG.*

*Proof.* We firstly prove the relationship between SDDPG and Eq. (2). The global fixed point satisfies:

$$w^* = \arg\min \mathbb{E}_{s\sim\rho(\cdot),\ a\sim\beta}[(Q_w^{\pi_\theta}(s,a) - Q^{\pi_\theta}(s,a))^2]$$

$\theta^* = \arg\max \mathbb{E}_{s\sim\rho(\cdot)}\left[Q_w^{\pi_\theta}(s,a)\right]$

Since the minimum of $\mathbb{E}_{s\sim\rho(\cdot),\ a\sim\beta}[(Q_w^{\pi_\theta}(s,a) - Q^{\pi_\theta}(s,a))^2]$ is zero,

this implies $\forall s, Q_w^{\pi_\theta}(s,a) = Q^{\pi_\theta}(s,a)$.

Therefore, putting the above equation into the dynamic of Eq. (2), we have

$w^* - \alpha_w \frac{\partial L(\theta,w^*)}{\partial w} = w^* - \alpha_w \mathbb{E}_{s\sim\rho(\cdot)}[2\left(Q_w^{\pi_{\theta^*}}(s,a) - Q^{\pi_{\theta^*}}(s,a)\right)\frac{\partial}{\partial w}Q_{w^*}^{\pi_{\theta^*}}(s,a)] = w^*$

Therefore, $w^*$ is the fixed point. By further using the fact that

$\forall s, Q_w^{\pi_\theta}(s,a) = Q^{\pi_\theta}(s,a) \to \forall s, Q_w^{\pi_\theta}(s,\pi_\theta(s)) = Q^{\pi_\theta}(s,\pi_\theta(s))\,,$

we conclude that $w^*$ is the optimal solution for Eq. (2).

Also notice that the dynamic of $\theta$ is the same for SDDPG and Eq. (2). Thus, $\theta^*$ is also the optimal solution for Eq. (2).

Now, we prove the relationship between SDDPG and FSDDPG. Since we already knew that $\forall s, Q_w^{\pi_\theta}(s,a) = Q^{\pi_\theta}(s,a)$, we can quickly verify that $k_1(\boldsymbol{\theta}) = \left(\frac{\partial^2 L(\theta,w)}{\partial\theta\partial w}\right)$ and $k_2(\boldsymbol{\theta}) = \left(\frac{\partial^2 L(\theta,w)}{\partial w^2}\right)$.

Thus, $\frac{\partial J(\theta,w)}{\partial\theta} - k_1(\boldsymbol{\theta})\, k_2^{-1}(\boldsymbol{\theta})\, \frac{\partial J(\theta,w)}{\partial w} = \frac{\partial J(\theta,w)}{\partial\theta} - \frac{\partial^2 L(\theta,w)}{\partial\theta\partial w}(\frac{\partial^2 L(\theta,w)}{\partial w^2})^{-1}\frac{\partial J(\theta,w)}{\partial w}.$

Thus, the dynamics of SDDPG and FSDDPG is the same. So, it is also optimal solution for FSDDPG.

$\square$

This lemma indicates that if we can find a solution of SDDPG, it is exactly the solution for the standard Stackelberg learning Eq. (2), as well as FSDDPG.

Now, we focus on the convergence rate of FSDDPG and SDDPG:

**Theorem 1.** *When $\pi \in C^q(S \otimes \Theta, \mathbb{R})$ and $Q \in C^q(S \otimes A \otimes W, \mathbb{R})$, $q \geq 2$. For a fixed point $\theta^*$ and $w^*$ such that $J_S(\boldsymbol{\theta}^*)^\top + J_S(\boldsymbol{\theta}^*)$ is positive-definite, the our method with learning rate $\alpha = \frac{\sqrt{\upsilon}}{\psi}$ converges locally with a rate of $O\left(\left(1 - \frac{\upsilon}{2\psi}\right)^{T/2}\right)$, where $\upsilon = \lambda_{\min}^2(S(\boldsymbol{\theta}^*))$, $\psi = \lambda_{\max}\left(J_S(\boldsymbol{\theta}^*)^\top J_S(\boldsymbol{\theta}^*)\right)$.*

*Proof.* Our framework follows Theorem 5 in (Fiez et al., 2020). We firstly have

$(I - \gamma_1 J_S(\boldsymbol{\theta}^*))^\top (I - \gamma_1 J_S(\boldsymbol{\theta}^*))$
$\leq \left(1 - 2\gamma_1\lambda_{\min}(S(\boldsymbol{\theta}^*)) + \gamma_1^2\lambda_{\max}\left(J_{S_\tau}^\top(\boldsymbol{\theta}^*) J_{S_\tau}(\boldsymbol{\theta}^*)\right)\right) I \leq (1 - \upsilon/\psi)I$

Taking the Taylor series of $I - \gamma_2\xi(\boldsymbol{\theta})$ around $\boldsymbol{\theta}^*$, we have

$I - \gamma_2\xi(\boldsymbol{\theta}) = (I - \gamma_1\xi(\boldsymbol{\theta})) + (I - \gamma_1 J_S(\boldsymbol{\theta}^*))(\boldsymbol{\theta} - \boldsymbol{\theta}^*) + R_1(\boldsymbol{\theta} - \boldsymbol{\theta}^*)$

where $R_1(\boldsymbol{\theta} - \boldsymbol{\theta}^*)$ is the remainder term satisfying $R_1(\boldsymbol{\theta} - \boldsymbol{\theta}^*) = O(||\boldsymbol{\theta} - \boldsymbol{\theta}^*||_2)$ as $\boldsymbol{\theta} \to \boldsymbol{\theta}^*$. This implies that there is $\delta$ such that $R_1(\boldsymbol{\theta} - \boldsymbol{\theta}^*) \leq \frac{\upsilon}{4\psi}||\boldsymbol{\theta} - \boldsymbol{\theta}^*||$ whenever $||\boldsymbol{\theta} - \boldsymbol{\theta}^*|| \leq \delta$. Hence

$||I - \gamma_2\xi(\boldsymbol{\theta}) - (I - \gamma_2\xi(\boldsymbol{\theta}^*))||_2 \leq \left(||I - \gamma_1 J_S(\boldsymbol{\theta}^*)||_2 + \frac{\upsilon}{4\psi}\right)||\boldsymbol{\theta} - \boldsymbol{\theta}^*||_2$
$\leq \left(\left(1 - \frac{\upsilon}{\psi}\right)^{1/2} + \frac{\upsilon}{4\psi}\right)||\boldsymbol{\theta} - \boldsymbol{\theta}^*||_2.$

Note that $\upsilon/\psi \in [0,1]$ since $\upsilon \leq \psi$, in fact,

$\upsilon = \lambda_{\min}^2(S(\boldsymbol{\theta}^*)) \leq \lambda_{\max}^2(S(\boldsymbol{\theta}^*)) \leq \lambda_{\max}\left(J_S^\top(\boldsymbol{\theta}^*) J_S(\boldsymbol{\theta}^*)\right) = \psi$

Thus, for any $\boldsymbol{\theta}_0 \in \{\boldsymbol{\theta}|\ ||\boldsymbol{\theta} - \boldsymbol{\theta}^*|| < \delta\}$

$$\|\boldsymbol{\theta}_T - \boldsymbol{\theta}^*\|_2 \leq \left(1 - \frac{\upsilon}{2\psi}\right)^{T/2} \|\boldsymbol{\theta}_0 - \boldsymbol{\theta}^*\|_2$$

so the local rate of convergence is $O\left(\left(1 - \frac{\upsilon}{2\psi}\right)^{T/2}\right)$. $\square$

Before we get deeper into the analysis, we define

$$\hat{\xi}(\boldsymbol{\theta}) := \begin{bmatrix} \frac{\partial -J(\theta,w)}{\partial \theta} - k_1(\boldsymbol{\theta}) \, k_2^{-1}(\boldsymbol{\theta}) \, \frac{\partial -J(\theta,w)}{\partial w} \\ \tau \frac{\partial \hat{L}(\theta,w)}{\partial w} \end{bmatrix}$$

$$\hat{J}_{\mathcal{S}}(\boldsymbol{\theta}) := \begin{bmatrix} \frac{\partial}{\partial \theta}\left(\frac{\partial -J(\theta,w)}{\partial \theta} - \frac{\partial^2 L(\theta,w)}{\partial \theta \partial w} \frac{\partial^2 L(\theta,w)}{\partial w^2}^{-1} \frac{\partial -J(\theta,w)}{\partial w}\right) & \frac{\partial}{\partial w}\left(\frac{\partial -J(\theta,w)}{\partial \theta} - \frac{\partial^2 L(\theta,w)}{\partial \theta \partial w} \frac{\partial^2 L(\theta,w)}{\partial w^2}^{-1} \frac{\partial -J(\theta,w)}{\partial w}\right) \\ \tau \frac{\partial^2 \hat{L}(\theta,w)}{\partial \theta \partial w} & \tau \frac{\partial^2 \hat{L}(\theta,w)}{\partial w^2} \end{bmatrix}$$

$$\hat{S}(\boldsymbol{\theta}^*) := \left(\hat{J}_{\mathcal{S}}^\top(\boldsymbol{\theta}^*) + \hat{J}_{\mathcal{S}}(\boldsymbol{\theta}^*)\right)$$

which decides the dynamic and the Jacobian of SDDPG.

Now, we show that the convergence rate of SDDPG:

**Proposition 2.** *When $\pi \in C^q(S \otimes \Theta, \mathbb{R})$ and $Q \in C^q(S \otimes A \otimes W, \mathbb{R})$, $q \geq 2$. For a fixed point $\theta^*$ and $w^*$ such that $\hat{J}_{\mathcal{S}}(\boldsymbol{\theta}^*)^\top + \hat{J}_{\mathcal{S}}(\boldsymbol{\theta}^*)$ is positive-definite, the our method with learning rate $\hat{\alpha} = \frac{\sqrt{\hat{\upsilon}}}{\hat{\psi}}$ converges locally with a rate of $O\left(\left(1 - \frac{\hat{\upsilon}}{2\hat{\psi}}\right)^{T/2}\right)$, where $\hat{\upsilon} = \lambda_{\min}^2\left(\hat{S}(\boldsymbol{\theta}^*)\right)$, $\hat{\psi} = \lambda_{\max}\left(\hat{J}_{\mathcal{S}}(\boldsymbol{\theta}^*)^\top \hat{J}_{\mathcal{S}}(\boldsymbol{\theta}^*)\right)$.*

*Proof.* Following the same steps in Theorem 1, we can easily check that

$$\left(I - \gamma_1 \hat{J}_{\mathcal{S}}(\boldsymbol{\theta}^*)\right)^\top \left(I - \gamma_1 \hat{J}_{\mathcal{S}}(\boldsymbol{\theta}^*)\right) \leq (1 - \hat{\upsilon}/\hat{\psi})I, \text{ and}$$

$$||I - \gamma_2 \hat{\xi}(\boldsymbol{\theta}) - \left(I - \gamma_2 \hat{\xi}(\boldsymbol{\theta}^*)\right)||_2 \leq \left(\left(1 - \frac{\hat{\upsilon}}{\hat{\psi}}\right)^{1/2} + \frac{\hat{\upsilon}}{4\hat{\psi}}\right)||\boldsymbol{\theta} - \boldsymbol{\theta}^*||_2,$$

and

$$\hat{\upsilon} \leq \hat{\psi}.$$

Thus, for any $\boldsymbol{\theta}_0 \in \{\boldsymbol{\theta}| \, ||\boldsymbol{\theta} - \boldsymbol{\theta}^*|| < \delta\}$

$$\|\boldsymbol{\theta}_T - \boldsymbol{\theta}^*\|_2 \leq \left(1 - \frac{\hat{\upsilon}}{2\hat{\psi}}\right)^{T/2} \|\boldsymbol{\theta}_0 - \boldsymbol{\theta}^*\|_2$$

so the local rate of convergence is $O\left(\left(1 - \frac{\hat{\upsilon}}{2\hat{\psi}}\right)^{T/2}\right)$. $\square$

Now, we show that FSDDPG is faster or equivalent to SDDPG for linear actor critic.

We firstly define linear actor as $\pi = \varphi(s)\theta$, where $\varphi(s)$ is the embedding, $\theta$ is the learnable parameters.

The linear critic is $Q(s,a) = \begin{bmatrix} \varphi(s) \\ \varphi(s)\theta \end{bmatrix} w$, where $w$ is the learnable parameters for critic.

**Theorem 2.** *For linear actor and critic, if $w^\mathsf{T} A \begin{bmatrix} \varphi(s) \\ \varphi(s)\theta \end{bmatrix} \succcurlyeq 0$, where $A$ is any Positive Semi Definite (PSD) matrix. Then, the convergence rate of FSDDPG (ours) is faster than or equivalent to that of SDDPG.*

*Proof.* [9] [10] We show the deriviatives for different terms:

$$\frac{\partial^2 L(\theta,w)}{\partial\theta\partial w} = 2\mathbb{E}_{\rho(s)}\left[\left(\frac{\partial}{\partial w}Q_w^{\pi_\theta}(s,a)\frac{\partial w}{\partial\theta}\right)^\top\left(\frac{\partial}{\partial w}Q_w^{\pi_\theta}(s,a)\right) + (Q_w^{\pi_\theta}(s,a) - Q^{\pi_\theta}(s,a))\left(\frac{\partial}{\partial\theta^\top}\frac{\partial}{\partial w}Q_w^{\pi_\theta}(s,a)\right)\right] \quad,$$

$$= \mathbb{E}_{\rho(s)}\left[\begin{bmatrix}0\\\varphi(s)1^\top\end{bmatrix}^\top\begin{bmatrix}\varphi(s)\\\varphi(s)\theta\end{bmatrix} + \left(\begin{bmatrix}\varphi(s)\\\varphi(s)\theta\end{bmatrix}w - Q^{\pi_\theta}(s,a)\right)\left(\begin{bmatrix}0\\\varphi(s)1^\top\end{bmatrix}\right)\right].$$

,

$$\frac{\partial^2 L(\theta,w)}{\partial w^2} = \mathbb{E}_{\rho(s)}\left[\frac{\partial}{\partial w^2}\left(Q_w^{\pi_\theta}(s,a) - Q^{\pi_\theta}(s,a)\right)^2\right] = \mathbb{E}_{\rho(s)}\left[\frac{\partial}{\partial w}\left[2\left(Q_w^{\pi_\theta}(s,a) - Q^{\pi_\theta}(s,a)\right)\frac{\partial}{\partial w}Q_w^{\pi_\theta}(s,a)\right]\right],$$

$$= \mathbb{E}_{\rho(s)}\left[\frac{\partial}{\partial w}\left[2\left(Q_w^{\pi_\theta}(s,a) - Q^{\pi_\theta}(s,a)\right)\begin{bmatrix}\varphi(s)\\\varphi(s)\theta\end{bmatrix}\right]\right] = \mathbb{E}_{\rho(s)}\left[\frac{\partial}{\partial w}\left[2\left(\begin{bmatrix}\varphi(s)\\\varphi(s)\theta\end{bmatrix}w - Q^{\pi_\theta}(s,a)\right)\begin{bmatrix}\varphi(s)\\\varphi(s)\theta\end{bmatrix}\right]\right],$$

$$= \mathbb{E}_{\rho(s)}\left[2\left(\begin{bmatrix}\varphi(s)\\\varphi(s)\theta\end{bmatrix}\begin{bmatrix}\varphi(s)\\\varphi(s)\theta\end{bmatrix}^\top\right)\right].$$

,

$$\frac{\partial J(\theta,w)}{\partial\theta} = \mathbb{E}_{\rho(s)}\left[\frac{\partial}{\partial\theta}Q_w^{\pi_\theta}(s,a)\right] = \mathbb{E}_{\rho(s)}\left[\begin{bmatrix}0\\\varphi(s)1^\top\end{bmatrix}w\right], \quad \frac{\partial J(\theta,w)}{\partial w} = \mathbb{E}_{\rho(s)}\left[\begin{bmatrix}\varphi(s)\\\varphi(s)\theta\end{bmatrix}\right] \cdot,$$

$$k_1(\theta,w) = \mathbb{E}_{\rho(s)}\left[\left(\frac{\partial}{\partial w}Q_w^{\pi_\theta}(s,a)\frac{\partial w}{\partial\theta}\right)^\top\left(\frac{\partial}{\partial w}Q_w^{\pi_\theta}(s,a)\right)\right] = 2\mathbb{E}_{\rho(s)}\left[\left(\begin{bmatrix}0\\\varphi(s)\end{bmatrix}w\right)^\top\left(\begin{bmatrix}\varphi(s)\\\varphi(s)\theta\end{bmatrix}\right)\right].$$

, and $k_2(\theta,w) = \mathbb{E}_{\rho(s)}\left(2\frac{\partial Q_w^{\pi_\theta}(s,a)}{\partial w}\frac{\partial Q_w^{\pi_\theta}(s,a)^\top}{\partial w}\right) = 2\mathbb{E}_{\rho(s)}\left(\frac{\partial Q_w^{\pi_\theta}(s,a)}{\partial w}\frac{\partial Q_w^{\pi_\theta}(s,a)^\top}{\partial w}\right) = 2\mathbb{E}_{\rho(s)}\left(\begin{bmatrix}\varphi(s)\\\varphi(s)\theta\end{bmatrix}\begin{bmatrix}\varphi(s)\\\varphi(s)\theta\end{bmatrix}^\top\right).$

Now, we put these derivatives into the $J_\mathcal{S}(\boldsymbol{\theta})$:

$$J_\mathcal{S}(\boldsymbol{\theta}) := \begin{bmatrix}\frac{\partial}{\partial\theta}\left(\frac{\partial -J(\theta,w)}{\partial\theta} - k_1 k_2^{-1}\frac{\partial -J(\theta,w)}{\partial w}\right) & \frac{\partial}{\partial w}\left(\frac{\partial -J(\theta,w)}{\partial\theta} - k_1 k_2^{-1}\frac{\partial -J(\theta,w)}{\partial w}\right) \\ \frac{\partial^2 L(\theta,w)}{\partial\theta\partial w} & \frac{\partial^2 L(\theta,w)}{\partial w^2}\end{bmatrix}$$

$$= \begin{bmatrix} A & B \\ C & D \end{bmatrix}.$$

where

$$A = \mathbb{E}_{\rho(s)}\left[\left(\begin{bmatrix}0\\\varphi(s)1^\top\end{bmatrix}w\right)^\top\left(\begin{bmatrix}0\\\varphi(s)1^\top\end{bmatrix}\right)\right]k_2^{-1}\mathbb{E}_{\rho(s)}\left[\begin{bmatrix}\varphi(s)\\\varphi(s)\theta\end{bmatrix}\right]$$

$$+\mathbb{E}_{\rho(s)}\left[\left(\begin{bmatrix}0\\\varphi(s)1^\top\end{bmatrix}w\right)^\top\left(\begin{bmatrix}\varphi(s)\\\varphi(s)\theta\end{bmatrix}\right)\right]\frac{\partial}{\partial\theta}k_2^{-1}\mathbb{E}_{\rho(s)}\left[\begin{bmatrix}\varphi(s)\\\varphi(s)\theta\end{bmatrix}\right],$$

$$+\mathbb{E}_{\rho(s)}\left[\left(\begin{bmatrix}0\\\varphi(s)1^\top\end{bmatrix}w\right)^\top\left(\begin{bmatrix}\varphi(s)\\\varphi(s)\theta\end{bmatrix}\right)\right]k_2^{-1}\mathbb{E}_{\rho(s)}\left[\begin{bmatrix}0\\\varphi(s)1^\top\end{bmatrix}\right]$$

$$B = \mathbb{E}_{\rho(s)}\left[\left(\begin{bmatrix}\varphi(s)\\\varphi(s)\theta\end{bmatrix}\begin{bmatrix}0\\\varphi(s)1^\top\end{bmatrix}\right)\right]k_2^{-1}\mathbb{E}_{\rho(s)}\left[\begin{bmatrix}\varphi(s)\\\varphi(s)\theta\end{bmatrix}\right],$$

$$C = \mathbb{E}_{\rho(s)}\left(\begin{bmatrix}0\\\varphi(s)1^\top\end{bmatrix}^\top\begin{bmatrix}\varphi(s)\\\varphi(s)\theta\end{bmatrix} + \left(\begin{bmatrix}\varphi(s)\\\varphi(s)\theta\end{bmatrix}w - Q^{\pi_\theta}(s,a)\right)\left(\begin{bmatrix}0\\\varphi(s)1^\top\end{bmatrix}\right)\right),$$

$$D = \mathbb{E}_{\rho(s)}\left[2\left(\begin{bmatrix}\varphi(s)\\\varphi(s)\theta\end{bmatrix}\begin{bmatrix}\varphi(s)\\\varphi(s)\theta\end{bmatrix}^\top\right)\right].$$

---

[9]With a little abuse of notations, we use $\theta$ as $\theta^*$ and $w$ as $w^*$ throughout the proof of this theorem.

[10]$A \succcurlyeq B$ means $A - B$ is PSD.

Similarly, for $\hat{J}_{\mathcal{S}}(\boldsymbol{\theta})$, we have:

$$\hat{J}_{\mathcal{S}}(\boldsymbol{\theta}) := \left[ \begin{array}{cc} \frac{\partial}{\partial\theta}\left( \frac{\partial -J(\theta,w)}{\partial\theta} - \frac{\partial^2 L(\theta,w)}{\partial w\partial\theta}\frac{\partial^2 L(\theta,w)}{\partial w^2}^{-1}\frac{\partial -J(\theta,w)}{\partial w} \right) & \frac{\partial}{\partial w}\left( \frac{\partial -J(\theta,w)}{\partial\theta} - \frac{\partial^2 L(\theta,w)}{\partial w\partial\theta}\frac{\partial^2 L(\theta,w)}{\partial w^2}^{-1}\frac{\partial -J(\theta,w)}{\partial w} \right) \\ \frac{\partial^2 L(\theta,w)}{\partial\theta\partial w} & \frac{\partial^2 L(\theta,w)}{\partial w^2} \end{array} \right]$$

$$= \left[ \begin{array}{cc} E & F \\ C & D \end{array} \right]$$

where

$$E = \mathbb{E}_{\rho(s)}\left[ \left( \left[ \begin{array}{c} 0 \\ \varphi(s)1^{\mathsf{T}} \end{array} \right] w \right)^{\mathsf{T}} \left[ \begin{array}{c} 0 \\ \varphi(s)1^{\mathsf{T}} \end{array} \right] + \left[ \begin{array}{c} 0 \\ \varphi(s)1^{\mathsf{T}} \end{array} \right] w \left[ \begin{array}{c} 0 \\ \varphi(s)1^{\mathsf{T}} \end{array} \right] \right] k_2^{-1}\mathbb{E}_{\rho(s)}\left[ \left[ \begin{array}{c} \varphi(s) \\ \varphi(s)\theta \end{array} \right] \right] ,$$

$$+\mathbb{E}_{\rho(s)}\left[ \left( \left[ \begin{array}{c} 0 \\ \varphi(s)1^{\mathsf{T}} \end{array} \right] w \right)^{\mathsf{T}} \left[ \begin{array}{c} \varphi(s) \\ \varphi(s)\theta \end{array} \right] \right] \frac{\partial}{\partial\theta}k_2^{-1}\mathbb{E}_{\rho(s)}\left[ \left[ \begin{array}{c} \varphi(s) \\ \varphi(s)\theta \end{array} \right] \right],$$

$$+\mathbb{E}_{\rho(s)}\left[ \left( \left[ \begin{array}{c} 0 \\ \varphi(s)1^{\mathsf{T}} \end{array} \right] w \right)^{\mathsf{T}} \left[ \begin{array}{c} \varphi(s) \\ \varphi(s)\theta \end{array} \right] \right] k_2^{-1}\mathbb{E}_{\rho(s)}\left[ \left[ \begin{array}{c} 0 \\ \varphi(s)1^{\mathsf{T}} \end{array} \right] \right]$$

$$= A + G,$$

and $G = \mathbb{E}_{\rho(s)}\left[ w^{\mathsf{T}}\left[ \begin{array}{c} 0 \\ \varphi(s)1^{\mathsf{T}} \end{array} \right]^{\mathsf{T}} \left[ \begin{array}{c} 0 \\ \varphi(s)1^{\mathsf{T}} \end{array} \right] \right] k_2^{-1}\mathbb{E}_{\rho(s)}\left[ \left[ \begin{array}{c} \varphi(s) \\ \varphi(s)\theta \end{array} \right] \right],$

$$F = \mathbb{E}_{\rho(s)}\left[ \left[ \begin{array}{c} \varphi(s) \\ \varphi(s)\theta \end{array} \right] \left[ \begin{array}{c} 0 \\ \varphi(s)1^{\mathsf{T}} \end{array} \right] \right] \mathbb{E}_{\rho(s)}\left( \left[ \begin{array}{c} \varphi(s) \\ \varphi(s)\theta \end{array} \right] \left[ \begin{array}{c} \varphi(s) \\ \varphi(s)\theta \end{array} \right]^{\mathsf{T}} \right)^{-1}\mathbb{E}_{\rho(s)}\left[ \left[ \begin{array}{c} \varphi(s) \\ \varphi(s)\theta \end{array} \right] \right],$$

Notice that, we can easily check that:

$B = F = G.$

Thus,

$$S(\boldsymbol{\theta}^*) - \hat{S}(\boldsymbol{\theta}^*) = \frac{1}{2}\left( \left[ \begin{array}{cc} A - E & B - F \\ 0 & 0 \end{array} \right]^{\mathsf{T}} + \left[ \begin{array}{cc} A - E & B - F \\ 0 & 0 \end{array} \right] \right) = $$

$$\frac{1}{2}\left[ \begin{array}{cc} G + (G)^{\mathsf{T}} & 0 \\ 0 & 0 \end{array} \right]$$

Since according to the assumption, we have $G$ to be Positive Semi Definite (PSD). Therefore,

$S(\boldsymbol{\theta}^*) - \hat{S}(\boldsymbol{\theta}^*) \succcurlyeq 0$. According to the property of definite matrices, we have

$\lambda_{min}^2(S(\boldsymbol{\theta}^*)) \geq \lambda_{min}^2\left( \hat{S}(\boldsymbol{\theta}^*) \right).$

For $J_{\mathcal{S}}(\boldsymbol{\theta}^*)^{\mathsf{T}}J_{\mathcal{S}}(\boldsymbol{\theta}^*)$, we have:

$$J_{\mathcal{S}}(\boldsymbol{\theta}^*)^{\mathsf{T}}J_{\mathcal{S}}(\boldsymbol{\theta}^*) = \left[ \begin{array}{cc} A^{\mathsf{T}} & C^{\mathsf{T}} \\ B^{\mathsf{T}} & D^{\mathsf{T}} \end{array} \right]\left[ \begin{array}{cc} A & B \\ C & D \end{array} \right] = \left[ \begin{array}{cc} A^{\mathsf{T}}A + C^{\mathsf{T}}C & B^{\mathsf{T}}A + C^{\mathsf{T}}D \\ AB^{\mathsf{T}} + D^{\mathsf{T}}C & B^{\mathsf{T}}B + D^{\mathsf{T}}D \end{array} \right],$$

and

$$\hat{J}_{\mathcal{S}}(\boldsymbol{\theta}^*)^{\mathsf{T}}\hat{J}_{\mathcal{S}}(\boldsymbol{\theta}^*) = \left[ \begin{array}{cc} E^{\mathsf{T}} & C^{\mathsf{T}} \\ F^{\mathsf{T}} & D^{\mathsf{T}} \end{array} \right]\left[ \begin{array}{cc} E & F \\ C & D \end{array} \right] = \left[ \begin{array}{cc} E^{\mathsf{T}}E + C^{\mathsf{T}}C & F^{\mathsf{T}}E + C^{\mathsf{T}}D \\ EF^{\mathsf{T}} + D^{\mathsf{T}}C & F^{\top}F + D^{\mathsf{T}}D \end{array} \right],$$

Therefore,

$$\hat{J}_{\mathcal{S}}(\boldsymbol{\theta}^*)^{\mathsf{T}}\hat{J}_{\mathcal{S}}(\boldsymbol{\theta}^*) - J_{\mathcal{S}}(\boldsymbol{\theta}^*)^{\mathsf{T}}J_{\mathcal{S}}(\boldsymbol{\theta}^*) = \left[ \begin{array}{cc} E^{\mathsf{T}}E - A^{\mathsf{T}}A & F^{\mathsf{T}}E - B^{\mathsf{T}}A \\ EF^{\mathsf{T}} - AB^{\mathsf{T}} & F^{\mathsf{T}}F - B^{\mathsf{T}}B \end{array} \right] = $$

$$\left[ \begin{array}{cc} E^{\mathsf{T}}E - A^{\mathsf{T}}A & F^{\mathsf{T}}E - B^{\mathsf{T}}A \\ EF^{\mathsf{T}} - AB^{\mathsf{T}} & 0 \end{array} \right],$$

Also notice that $E^{\mathsf{T}}E - A^{\mathsf{T}}A = (A + G)^{\mathsf{T}}(A + G) - A^{\mathsf{T}}A = (A^{\mathsf{T}} + G^{\mathsf{T}})(A + G) - A^{\mathsf{T}}A = G^{\mathsf{T}}A + A^{\mathsf{T}}G + G^{\mathsf{T}}G,$

Based on the property of PSD, we can know that $G^\mathsf{T} G$ and $G^\mathsf{T} A + A^\mathsf{T} G$ are also PSD. Thus, $E^\mathsf{T} E - A^\mathsf{T} A$ is also PSD.

Moreover, we can check that

$EF^\mathsf{T} - AB^\mathsf{T} = GB^\mathsf{T}$, and

$$GB^T = \left( \mathbb{E}_{\rho(s)} \left[ \begin{bmatrix} 0 \\ \varphi(s)1^\mathsf{T} \end{bmatrix} w \begin{bmatrix} 0 \\ \varphi(s)1^\mathsf{T} \end{bmatrix} \right] k_2^{-1} \mathbb{E}_{\rho(s)} \left[ \begin{bmatrix} \varphi(s) \\ \varphi(s)\theta \end{bmatrix} \right] \right)$$
$$\left( \mathbb{E}_{\rho(s)} \left[ \begin{bmatrix} 0 \\ \varphi(s)1^\mathsf{T} \end{bmatrix} w \begin{bmatrix} 0 \\ \varphi(s)1^\mathsf{T} \end{bmatrix} \right] k_2^{-1} \mathbb{E}_{\rho(s)} \left[ \begin{bmatrix} \varphi(s) \\ \varphi(s)\theta \end{bmatrix} \right] \right)^\mathsf{T}$$

$$\overset{a}{\succcurlyeq} 0,$$

Where $a$ is due to the property of PSD .

Thus, we have

$$\hat{J}_\mathcal{S}(\boldsymbol{\theta}^*)^\top \hat{J}_\mathcal{S}(\boldsymbol{\theta}^*) - J_\mathcal{S}(\boldsymbol{\theta}^*)^\top J_\mathcal{S}(\boldsymbol{\theta}^*) \succcurlyeq 0,$$

Therefore, $\lambda_{max}\left( \hat{J}_\mathcal{S}(\boldsymbol{\theta}^*)^\top \hat{J}_\mathcal{S}(\boldsymbol{\theta}^*) \right) \geq \lambda_{max}\left( J_\mathcal{S}(\boldsymbol{\theta}^*)^\top J_\mathcal{S}(\boldsymbol{\theta}^*) \right).$

Thus, $\left( 1 - \dfrac{\lambda_{\min}^2\left( \hat{S}(\boldsymbol{\theta}^*) \right)}{2\lambda_{\max}\left( \hat{J}_\mathcal{S}(\boldsymbol{\theta}^*)^\top \hat{J}_\mathcal{S}(\boldsymbol{\theta}^*) \right)} \right)^{T/2} \geq \left( 1 - \dfrac{\lambda_{\min}^2(S(\boldsymbol{\theta}^*))}{2\lambda_{\max}\left( J_\mathcal{S}(\boldsymbol{\theta}^*)^\top J_\mathcal{S}(\boldsymbol{\theta}^*) \right)} \right)^{T/2}$, which achieves our desired result.

$\square$

Here is an example that helps us to show the convergence rate, and we use toy example 1 as an example:

Recall that in toy example 1, we have $L(\theta, w) = (w\theta^2 + \theta^2)^2$, $J(\theta, w) = w\theta^2$, $\tau = 1$.

And we can easily calculate that:

$$-\frac{\partial J(\theta,w)}{\partial \theta} - \frac{\partial^2 L(\theta,w)}{\partial \theta \partial w} \frac{\partial^2 L(\theta,w)}{\partial w^2}^{-1} \frac{\partial -J(\theta,w)}{\partial w} = -w + \left[ 4(2w+2)\theta^3 \right] \left[ 2\theta^4 \right]^{-1} \theta$$
$$= 3w + 1$$

Thus, we have:

$$\hat{J}_\mathcal{S}(\boldsymbol{\theta}) := \left[ \begin{array}{cc} \frac{\partial}{\partial w}\left( \frac{\partial -J(\theta,w)}{\partial \theta} - \frac{\partial^2 L(\theta,w)}{\partial \theta \partial w}\frac{\partial^2 L(\theta,w)}{\partial w^2}^{-1}\frac{\partial -J(\theta,w)}{\partial w} \right) & \frac{\partial}{\partial \theta}\left( \frac{\partial -J(\theta,w)}{\partial \theta} - \frac{\partial^2 L(\theta,w)}{\partial \theta \partial w}\frac{\partial^2 L(\theta,w)}{\partial w^2}^{-1}\frac{\partial -J(\theta,w)}{\partial w} \right) \\ \frac{\partial^2 L(\theta,w)}{\partial \theta \partial w} & \frac{\partial^2 L(\theta,w)}{\partial w^2} \end{array} \right]$$
$$= \left[ \begin{array}{cc} \frac{\partial}{\partial w}(3w+1) & \frac{\partial}{\partial \theta}(3w+1) \\ 2(w\theta^2 + \theta^2)2\theta & 4\theta^3 \end{array} \right] = \left[ \begin{array}{cc} 1 & 0 \\ 2(w\theta^2 + \theta^2)2\theta & 4\theta^3 \end{array} \right]$$

putting $\theta^* = 0$, $w^* = -1$ into the equation,

$$\hat{J}_\mathcal{S}(\boldsymbol{\theta}^*) = \left[ \begin{array}{cc} 1 & 0 \\ 0 & 0 \end{array} \right]$$

Finally,

$$0.5\left( \hat{J}_\mathcal{S}(\boldsymbol{\theta}^*)^\top + \hat{J}_\mathcal{S}(\boldsymbol{\theta}^*) \right) = 0.5\left( \left[ \begin{array}{cc} 1 & 0 \\ 0 & 0 \end{array} \right] + \left[ \begin{array}{cc} 1 & 0 \\ 0 & 0 \end{array} \right] \right) = \left[ \begin{array}{cc} 1 & 0 \\ 0 & 0 \end{array} \right]$$
$$\hat{J}_\mathcal{S}(\boldsymbol{\theta}^*)^\top \hat{J}_\mathcal{S}(\boldsymbol{\theta}^*) = \left[ \begin{array}{cc} 1 & 0 \\ 0 & 0 \end{array} \right] \times \left[ \begin{array}{cc} 1 & 0 \\ 0 & 0 \end{array} \right] = \left[ \begin{array}{cc} 1 & 0 \\ 0 & 0 \end{array} \right]$$

Thus $\lambda_{\min}^2 = 0$, $\lambda_{max} = 1$, the convergence rate is $O\left( (1-0)^{T/2} \right) = O\left( 1^{T/2} \right)$.

Similarly, we have:

$$-\frac{\partial J(\theta,w)}{\partial \theta} - k_1 k_2^{-1} \frac{\partial -J(\theta,w)}{\partial w} = -w + \theta w \left(\theta^2\right)^{-1} \theta \quad,$$

Thus,

$$J_\mathcal{S}(\boldsymbol{\theta}) := \left[ \begin{array}{cc} \frac{\partial}{\partial w}\left(\frac{-\lambda}{(\theta^2+\lambda)}w\right) & \frac{\partial}{\partial \theta}\left(\frac{-\lambda}{(\theta^2+\lambda)}w\right) \\ \frac{\partial^2 L(\theta,w)}{\partial\theta\partial w} & \frac{\partial^2 L(\theta,w)}{\partial w^2} \end{array} \right]$$

$$= \left[ \begin{array}{cc} \frac{-\lambda}{(\theta^2+\lambda)} & \frac{2\lambda\theta}{(\theta^2+\lambda)^2}w \\ 2(w\theta^2+\theta^2)2\theta & 4\theta^3 \end{array} \right]$$

putting $\theta^* = 0$, $w^* = -1$ into the equation,

$$J_\mathcal{S}(\boldsymbol{\theta}^*) = \left[ \begin{array}{cc} -1 & 0 \\ 0 & 0 \end{array} \right] = \left[ \begin{array}{cc} -1 & 0 \\ 0 & 0 \end{array} \right]$$

Finally

$$0.5\left(J_\mathcal{S}(\boldsymbol{\theta}^*)^\top + J_\mathcal{S}(\boldsymbol{\theta}^*)\right) = \left[ \begin{array}{cc} -1 & 0 \\ 0 & 0 \end{array} \right]$$

$$J_\mathcal{S}(\boldsymbol{\theta}^*)^\top J_\mathcal{S}(\boldsymbol{\theta}^*) = \left[ \begin{array}{cc} -1 & 0 \\ 0 & 0 \end{array} \right] \times \left[ \begin{array}{cc} -1 & 0 \\ 0 & 0 \end{array} \right] = \left[ \begin{array}{cc} 1 & 0 \\ 0 & 0 \end{array} \right]$$

Thus $\lambda_{\min}^2 = 1, \lambda_{max} = 1$ $O\left(\left(1-\frac{1}{2}\right)^{T/2}\right) = O\left(0.5^{T/2}\right)$

## E  TRAJECTORIES OF SDDPG AND FSDDPG.

This section we show the distances of trajectories of SDDPG and FSDDPG are bounded. We firstly introduce an assumption:

**Assumption 1.** $Q_w(s,a)$ *is a linear function.*

This assumption is also adopted in the analysis of temporal difference (Bhandari et al., 2018). In fact, the assumption only serves as foundations of the proofs ine (Bhandari et al., 2018), and it is possible to extend this assumption into the neural networks form (Cai et al., 2019).

Some mild assumptions are given:

**Assumption 2.** *The following Hessian matrices and the Jacobian matrices are bounded.*

$$||\frac{\partial w}{\partial \pi(a|s)}|| \le H_1, ||\frac{\partial}{\partial\theta}\pi_\theta(a\mid s)|| \le H_2, \; ||\frac{\partial^2}{\partial w^2}Q_w^{\pi_\theta}(s,a)|| \le H_3, \; ||\frac{\partial Q_w^{\pi_\theta}(s,a)}{\partial\boldsymbol{\theta}}|| \le H_4$$

$$||\frac{\partial}{\partial\theta}\left(\left(\frac{\partial}{\partial w}Q_w^{\pi_\theta}(s,a)\right)\left(\frac{\partial}{\partial w}Q_w^{\pi_\theta}(s,a)^\top\right)^{-1}\right)|| \le H_5,$$

$$||\frac{\partial}{\partial w}\left(\left(\frac{\partial}{\partial w}Q_w^{\pi_\theta}(s,a)\right)\left(\frac{\partial}{\partial w}Q_w^{\pi_\theta}(s,a)^\top\right)^{-1}\right)|| \le H_6$$

$$||\frac{\partial}{\partial\theta^\top}\frac{\partial}{\partial w}Q_w^{\pi_\theta}(s,a)|| \le H_7$$

$$||\frac{\partial Q_w^{\pi_\theta}(s,a)}{\partial w}|| \le H_8$$

This assumption is standard assumption that the norm of derivatives is bounded, similar to (Ghadimi & Wang, 2018; Reddi et al., 2018). We set $\hat{H} = \max\{H_1, H_2, \ldots, H_8\}$, and $\bar{H} = \min\{H_1, H_2, \ldots, H_8\}$.

For compactness, we define $\hat{k}_1 := \left(\frac{\partial^2 L(\theta,w)}{\partial\theta\partial w}\right)$, and $\hat{k}_2 := \left(\frac{\partial^2 L(\theta,w)}{\partial w^2}\right)$.

We also define $\kappa_1 := \hat{k}_1 - k_1$, and $\kappa_2 := \hat{k}_2 - k_2$ as the residual terms. We also define $\epsilon := ||Q_w^{\pi_\theta}(s,a) - Q^{\pi_\theta}||$.

We also denote the minimum eigenvalue of $\mathbb{E}\left[\frac{\partial Q_w^{\pi_\theta}(s,a)}{\partial w}\frac{\partial Q_w^{\pi_\theta}(s,a)^\top}{\partial w}\right]$ as $\lambda_{\min} :=$ $\lambda_{\min}\left(\mathbb{E}\left[\frac{\partial Q_w^{\pi_\theta}(s,a)}{\partial w}\frac{\partial Q_w^{\pi_\theta}(s,a)^\top}{\partial w}\right]\right).$

**Lemma 7.** *we have* $||\kappa_1|| \le 2\epsilon H_7$ *and* $||\kappa_2|| \le 2\epsilon H_3$.

*Proof.* Based on the definition of $\kappa_1$ and $\kappa_2$ and the property of norm, we have:

$$||\kappa_1|| = ||\mathbb{E}_{s\sim\rho, a\sim\pi_\theta(\cdot|s)}\left[2\left(Q_w^{\pi_\theta}(s,a) - Q^{\pi_\theta}(s,a)\right)\left(\frac{\partial}{\partial\theta^\top}\frac{\partial}{\partial w}Q_w^{\pi_\theta}(s,a)\right)\right]||$$
$$\le 2\epsilon H_7$$

$$||\kappa_2|| = ||\mathbb{E}_{s\sim\rho, a\sim\pi_\theta(\cdot|s)}\left[2\left(Q_w^{\pi_\theta}(s,a) - Q^{\pi_\theta}(s,a)\right)\frac{\partial^2 Q_w^{\pi_\theta}(s,a)}{\partial w^2}\right]||$$
$$\le 2\epsilon H_3$$

$\square$

**Lemma 8.** *For the i.i.d sampling schemes, we have* $\epsilon \le \frac{H}{T}$ *, where $H$ is a positive constant.*

*Proof.* The proof directly follows Sec. 6.1 in (Bhandari et al., 2018). $\square$

The i.i.d sampling scheme assumptions are also used in (Nachum et al., 2019; Sutton et al., 2012; Liu et al., 2020; Dalal et al., 2018).

**Lemma 9.** *If* $\epsilon \le \sqrt{\frac{\lambda_{\min}-1}{4H_3^2 H_8}}$, *We always have* $||\kappa_1 k_2 - k_1\kappa_2|| \le 2(H_8 H_{10} H_7 + H_1 H_2 H_3)\epsilon$, $||(k_2 + \kappa_2)k_2|| \le 2H_3 H_8\epsilon$.

*Proof.* For $||\kappa_1 diag^o(k_2) - k_1\kappa_2||$, we have:

$$||\kappa_1 k_2 - k_1\kappa_2||$$
$$= ||\left(\left(Q_w^{\pi_\theta}(s,a) - Q^{\pi_\theta}(s,a)\right)\left(\frac{\partial}{\partial\theta^\top}\frac{\partial}{\partial w}Q_w^{\pi_\theta}(s,a)\right)\right)\frac{\partial Q_w^{\pi_\theta}(s,a)}{\partial w}\frac{\partial Q_w^{\pi_\theta}(s,a)^\top}{\partial w}$$
$$- \frac{\partial}{\partial\theta^\top}\frac{\partial}{\partial w}Q_w^{\pi_\theta}(s,a)\left(\left(Q_w^{\pi_\theta}(s,a) - Q^{\pi_\theta}(s,a)\right)\frac{\partial Q_w^{\pi_\theta}(s,a)}{\partial w}\frac{\partial Q_w^{\pi_\theta}(s,a)^\top}{\partial w}\right)||$$
$$\le ||\frac{\partial Q_w^{\pi_\theta}(s,a)}{\partial w}\frac{\partial Q_w^{\pi_\theta}(s,a)^\top}{\partial w}||\,||\left(Q_w^{\pi_\theta}(s,a) - Q^{\pi_\theta}(s,a)\right)\left(\frac{\partial}{\partial\theta^\top}\frac{\partial}{\partial w}Q_w^{\pi_\theta}(s,a)\right)||$$
$$+2||\left(\frac{\partial}{\partial w}Q_w^{\pi_\theta}(s,a)\frac{\partial w}{\partial\theta}\right)^\top\left(\frac{\partial}{\partial w}Q_w^{\pi_\theta}(s,a)\right)||\,||\left(Q_w^{\pi_\theta}(s,a) - Q^{\pi_\theta}(s,a)\right)\frac{\partial^2}{\partial w^2}Q_w^{\pi_\theta}(s,a)||$$
$$\le 2H_8 H_7\epsilon + 2H_1 H_2 H_3\epsilon = 2(H_8 H_7 + H_1 H_2 H_3)\epsilon$$

For $||[\kappa_2 + k_2]^{-1}||$, we have:

$$||[\kappa_2 + k_2]^{-1}|| \overset{(a)}{=} ||\kappa_2^{-1} - \kappa_2^{-1}k_2[k_2 + \kappa_2]^{-1}||$$
$$\le 2\epsilon H_3 + 2\epsilon H_3 2\epsilon H_3||[k_2 + \kappa_2]^{-1}||$$

After some simplification, we have:

$$||[k_2 + \kappa_2]^{-1}|| \le \frac{G_1}{G_2}$$

(a) follows Lemma 4, where $G_1 = 2\epsilon H_3$, $G_2 = 1 - 2\epsilon H_3 2\epsilon H_3$.

Therefore, putting the above inequality together, we have:

$$||[(k_2 + \kappa_2)k_2]^{-1}|| \le ||[k_2 + \kappa_2]^{-1}||\,||k_2^{-1}||$$
$$\le \frac{G_1}{G_2}||k_2^{-1}||$$
$$\le \frac{G_1}{G_2\lambda_{\min}} \overset{(b)}{\le} 2H_3\epsilon.$$

(b) is based on the assumption $\epsilon \le \sqrt{\frac{\lambda_{\min}-1}{4H_3^2}}$, which induces $\frac{\epsilon}{1 - 2\epsilon H_3 2\epsilon H_3} \le \lambda_{\min}\epsilon$.

$\square$

**Theorem 3.** $||\zeta(\boldsymbol{\theta}) - \hat{\zeta}(\boldsymbol{\theta})||$ *is bounded by* $5H_{10}H_3H_8(H_8H_{10}H_7 + H_1H_2H_3)H\frac{1}{T^2}$.

*Proof.* Using the lemmas mentioned above, we have:

$$
\begin{aligned}
||\zeta(\boldsymbol{\theta}) - \hat{\zeta}(\boldsymbol{\theta})|| &= ||\left[\left(\frac{(k_1+\kappa_1)}{(k_2+\kappa_2)} - \frac{k_1}{k_2}\right)\frac{\partial J(\theta,w)}{\partial w}, 0\right]|| \\
&\leq ||\left(\frac{(k_1+\kappa_1)}{(k_2+\kappa_2)} - \frac{k_1}{k_2}\right)||||\frac{\partial J(\theta,w)}{\partial w}|| \\
&= ||(\kappa_1 k_2 - k_1\kappa_2)(k_2 + \kappa_2)k_2^{-1}||||\frac{\partial J(\theta,w)}{\partial w}|| \\
&\leq ||((k_2) + \kappa_2)(k_2||||(((k_2 + \kappa_2)k_2^{-1}||||\frac{\partial J(\theta,w)}{\partial w}|| \\
&\leq 5H_{10}H_3(H_{10}H_7 + H_1H_2H_3)\epsilon^2 \\
&\leq 5H_{10}H_3(H_{10}H_7 + H_1H_2H_3)H\frac{1}{T^2}
\end{aligned}
$$

$\square$

Where the third inequality is based on Lemma 7 and 9, and the last inequality is based on Lemma 8.

## F    ALMOST SURELY AVOID SADDLE POINTS.

The proof relies on Theorem 4 in (Fiez et al., 2020), as well as Corollary 23 in (Mazumdar et al., 2020).

Firstly recall that the definitions of critical points, strict saddle points, as well as fixed points (for our methods) in (Fiez et al., 2020; Chasnov et al., 2020; Ratliff et al., 2016):

**Definition 1.** *A point is a critical point if* $\zeta(\boldsymbol{\theta}) = 0$.

**Definition 2.** *A point is a strict saddle point if* $\zeta(\boldsymbol{\theta}) = 0$, *and its some eigenvalues are less than 0, while others are larger than 0.*

**Definition 3.** *A point is a locally asymptotically stable equilibrium if* $\zeta(\boldsymbol{\theta}) = 0$, *and all its eigenvalues are larger than 0.*

**Lemma 10.** $||\zeta(\boldsymbol{\theta}_1) - \zeta(\boldsymbol{\theta}_2)||$ *is Lipschitz continuous.*

*Proof.* Firstly we observe the following inequalities:

$$
\begin{aligned}
||\frac{\partial}{\partial w_1}\frac{\partial L(\theta_1,w_1)}{\partial w_1}|| &\leq ||\frac{\partial}{\partial w_1}\left(Q_{w_1}^{\pi_{\theta_1}}(s,a) - Q^{\pi}(s,a)\right)\frac{\partial Q_{w_1}^{\pi_{\theta_1}}(s,a)}{\partial w_1}|| \\
&= ||\frac{\partial}{\partial w_1}\left[\left(Q_{w_1}^{\pi_{\theta_1}}(s,a) - Q^{\pi}(s,a)\right)\frac{\partial Q_{w_1}^{\pi_{\theta_1}}(s,a)}{\partial w_1}\right]|| \\
&= ||\frac{\partial^2 Q_{w_1}^{\pi_{\theta_1}}(s,a)}{\partial w_1^2}\left(Q_{w_1}^{\pi_{\theta_1}}(s,a) - Q^{\pi}(s,a)\right) + \frac{\partial Q_{w_1}^{\pi_{\theta_1}}(s,a)}{\partial w_1}\frac{\partial Q_{w_1}^{\pi_{\theta_1}}(s,a)}{\partial w_1}|| \\
&\leq \frac{2H_3 R_{\max}}{1-\gamma} + H_4^2
\end{aligned}
$$

$$
||\frac{\partial}{\partial\theta}\left(\frac{\partial J(\theta,w)}{\partial\theta} - k_1(\boldsymbol{\theta})\, k_2^{-1}(\boldsymbol{\theta})\,\frac{\partial J(\theta,w)}{\partial w}\right)||
$$

$$
\begin{aligned}
&\leq ||\frac{\partial}{\partial\theta}\frac{\partial Q_w^{\pi_\theta}(s,a)}{\partial\theta} - \frac{\partial}{\partial\theta}\{\left[\left(\frac{\partial}{\partial w}Q_w^{\pi_\theta}(s,a)\right)^{\top}\left(\frac{\partial}{\partial w}Q_w^{\pi_\theta}(s,a)\frac{\partial w}{\partial\theta}\right)\right]^{\top} \\
&\quad \left(\frac{\partial}{\partial w}Q_w^{\pi_\theta}(s,a)\frac{\partial}{\partial w}Q_w^{\pi_\theta}(s,a)^{\top}\right)^{-1}\frac{\partial Q_w^{\pi_{\theta_1}}(s,a)}{\partial w}\}|| \\
&= ||\frac{\partial^2 Q_w^{\pi_\theta}(s,a)}{\partial\theta^2} - \frac{\partial}{\partial\theta}\left\{\left(\frac{\partial}{\partial w}Q_w^{\pi_\theta}(s,a)\frac{\partial w}{\partial\theta}\right)^{\top}\left(\frac{\partial}{\partial w}Q_w^{\pi_\theta}(s,a)\right)\left(\frac{\partial}{\partial w}Q_w^{\pi_\theta}(s,a)^{\top}\right)^{-1}\right\}|| \\
&= ||\frac{\partial^2 Q_w^{\pi_\theta}(s,a)}{\partial\theta^2} - \frac{\partial}{\partial\theta}\left\{\frac{\partial Q_w^{\pi_\theta}(s,a)}{\partial\theta}\left(\frac{\partial}{\partial w}Q_w^{\pi_\theta}(s,a)\right)\left(\frac{\partial}{\partial w}Q_w^{\pi_\theta}(s,a)^{\top}\right)^{-1}\right\}|| \\
&\leq H_3 + H_3H_4 + H_1H_2H_4H_5
\end{aligned}
$$

$$
\begin{aligned}
||\frac{\partial^2 L(\theta,w)}{\partial\theta\partial w}|| &\leq ||\frac{\partial}{\partial\theta}\left(Q_{w_1}^{\pi_{\theta_1}}(s,a) - Q^{\pi}(s,a)\right)\frac{\partial Q_{w_1}^{\pi_{\theta_1}}(s,a)}{\partial w_1}|| \\
&= ||\left(\frac{\partial}{\partial\theta}Q_{w_1}^{\pi_{\theta_1}}(s,a)\right)\frac{\partial Q_{w_1}^{\pi_{\theta_1}}(s,a)}{\partial w_1} + \left(Q_{w_1}^{\pi_{\theta_1}}(s,a) - Q^{\pi}(s,a)\right)\frac{\partial}{\partial\theta}\frac{\partial Q_{w_1}^{\pi_{\theta_1}}(s,a)}{\partial w_1}|| \\
&\leq H_1H_3H_4^2 + \frac{2H_3 R_{\max}}{1-\gamma}H_4
\end{aligned}
$$

$$\|\frac{\partial}{\partial w}\left(\frac{\partial J(\theta,w)}{\partial \theta} - k_1(\boldsymbol{\theta})\, k_2^{-1}(\boldsymbol{\theta})\, \frac{\partial J(\theta,w)}{\partial w}\right)\|$$

$$\leq \|\frac{\partial}{\partial w}\left(\frac{\partial Q_w^{\pi_\theta}(s,a)}{\partial \theta}\right) - \frac{\partial}{\partial w}\{\left[\left(\frac{\partial}{\partial w}Q_w^{\pi_\theta}(s,a)\right)^\top \left(\frac{\partial}{\partial w}Q_w^{\pi_\theta}(s,a)\frac{\partial w}{\partial \theta}\right)\right]^\top$$

$$\left(\frac{\partial}{\partial w}Q_w^{\pi_\theta}(s,a)\frac{\partial}{\partial w}Q_w^{\pi_\theta}(s,a)^\top\right)^{-1} \frac{\partial Q_w^{\pi_{\theta_1}}(s,a)}{\partial w}\}\|$$

$$= \|\frac{\partial}{\partial w}\left(\frac{\partial Q_w^{\pi_\theta}(s,a)}{\partial \theta}\right) - \frac{\partial}{\partial w}\left\{\frac{\partial Q_w^{\pi_\theta}(s,a)}{\partial \theta}\left(\frac{\partial}{\partial w}Q_w^{\pi_\theta}(s,a)\right)\left(\frac{\partial}{\partial w}Q_w^{\pi_\theta}(s,a)^\top\right)^{-1}\right\}\|$$

$$= H_4 + H_1 H_2 H_4 H_6$$

Therefore, based on the fact that for any continuous function $f$ if $\sup\|\nabla_x f(x)\| \leq L$, then $f(x)$ is L-lipschitz continuous, we have:

$$\|w(\boldsymbol{\theta_1}) - w(\boldsymbol{\theta_2})\|$$

$$= \|\left[\begin{matrix}\left(\frac{\partial J(\theta_1,w_1)}{\partial \theta_1} - k_1(\boldsymbol{\theta_1})\, k_2^{-1}(\boldsymbol{\theta_1})\, \frac{\partial J(\theta_1,w_1)}{\partial w_1}\right) - \left(\frac{\partial J(\theta_2,w_2)}{\partial \theta_2} - k_1(\boldsymbol{\theta_2})\, k_2^{-1}(\boldsymbol{\theta_2})\, \frac{\partial J(\theta_2,w_2)}{\partial w_2}\right) \\ \tau\frac{\partial L(\theta_1,w_1)}{\partial w_1} - \tau\frac{\partial L(\theta_2,w_2)}{\partial w_2}\end{matrix}\right]\|$$

$$\leq \tau\|\frac{\partial L(\theta_1,w_1)}{\partial w_1} - \frac{\partial L(\theta_2,w_2)}{\partial w_2}\|$$

$$+ \|\left(\frac{\partial J(\theta_1,w_1)}{\partial \theta_1} - k_1(\boldsymbol{\theta_1})\, k_2^{-1}(\boldsymbol{\theta_1})\, \frac{\partial J(\theta_1,w_1)}{\partial w_1}\right) - \left(\frac{\partial J(\theta_2,w_2)}{\partial \theta_2} - k_1(\boldsymbol{\theta_2})\, k_2^{-1}(\boldsymbol{\theta_2})\, \frac{\partial J(\theta_2,w_2)}{\partial w_2}\right)\|$$

$$\leq \max\{ H_4 + H_1 H_2 H_4 H_6,\ H_3 + H_3 H_4 + H_1 H_2 H_4 H_5\}\|\boldsymbol{\theta_1} - \boldsymbol{\theta_2}\|$$

$$+ \max\left\{\frac{2H_3 R_{\max}}{1-\gamma} + H_4^2,\ H_1 H_3 H_4^2 + \frac{2H_3 R_{\max}}{1-\gamma}\right\}\|\boldsymbol{\theta_1} - \boldsymbol{\theta_2}\|$$

$$\leq 2\max\{ H_4 + H_1 H_2 H_4 H_6,\ H_3 + H_3 H_4 + H_1 H_2 H_4 H_5,$$

$$\frac{2H_3 R_{\max}}{1-\gamma} + H_4^2,\ H_1 H_3 H_4^2 + \frac{2H_3 R_{\max}}{1-\gamma}\}\|\boldsymbol{\theta_1} - \boldsymbol{\theta_2}\| \leq L\|\boldsymbol{\theta_1} - \boldsymbol{\theta_2}\|,$$

where $L := 2\max\{ H_4 + H_1 H_2 H_4 H_6,\ H_3 + H_3 H_4 + H_1 H_2 H_4 H_5,$
$\frac{2H_3 R_{\max}}{1-\gamma} + H_4^2,\ H_1 H_3 H_4^2 + \frac{2H_3 R_{\max}}{1-\gamma}\}$. $\qquad\square$

Now, we can finally prove our claim, the framework follows Theorem 4 in (Fiez et al., 2020):

**Theorem 4.** *If $\alpha L < 1$, our method converges to the strict saddle points of both $\zeta(\boldsymbol{\theta})$ and $\hat{z\hat{e}ta}(\boldsymbol{\theta})$ on a set of measure zero.*

*Proof.* We firstly prove that $g(\boldsymbol{\theta})$ is *invertible*, which can be done by contradictions. Consider $\boldsymbol{\theta_1} \neq \boldsymbol{\theta_2}$, and suppose $g(\boldsymbol{\theta_1}) = g(\boldsymbol{\theta_2})$ so that $\boldsymbol{\theta_1} - \boldsymbol{\theta_2} = \alpha w(\boldsymbol{\theta_1}) - \alpha w(\boldsymbol{\theta_2})$. According to Lemma 10, $\alpha\|w(\boldsymbol{\theta_1}) - w(\boldsymbol{\theta_2})\| \leq \alpha L\|\boldsymbol{\theta_1} - \boldsymbol{\theta_2}\| < \|\boldsymbol{\theta_1} - \boldsymbol{\theta_2}\|$, giving a contradiction. Therefore, it is invertible.

Now, observe that $\frac{\partial g(\boldsymbol{\theta})}{\partial \boldsymbol{\theta}} = I - \alpha J_\mathcal{S}(\boldsymbol{\theta})$, and if $\frac{\partial g(\boldsymbol{\theta})}{\partial \boldsymbol{\theta}}$ is invertible, then according to the inverse function theorem, we have $g(\boldsymbol{\theta})$ is local diffeomorphism. Hence, it suffices to show that $I - \alpha J_\mathcal{S}(\boldsymbol{\theta})$ does not have an eigenvalue of 0 (or $\frac{\partial g(\boldsymbol{\theta})}{\partial \boldsymbol{\theta}}$ does not have an eigenvalue of 1). Indeed, letting $\rho(A)$ be the spectral radius of a matrix $A$, we know that $\rho(A) \leq \|A\|$ for any square matrix $A$ and induced operator norm $\|\cdot\|$ so that $\rho(\alpha J_\mathcal{S}(x)) \leq \|\alpha J_\mathcal{S}(x)\| \leq \alpha \sup_\theta \|J_\mathcal{S}(\boldsymbol{\theta})\| < \alpha L < 1,$

The above implies that all eigenvalues of $\alpha J_\mathcal{S}(\boldsymbol{\theta})$ have absolute value less than 1. Since $g(\boldsymbol{\theta})$ is injective by the preceding argument, its inverse is well-defined . Also, since $g(\boldsymbol{\theta})$ is a local diffeomorphism on $\Theta \bigotimes \Sigma$, it follows that $g(\boldsymbol{\theta})$ is smooth on $\Theta \bigotimes \Sigma$. Thus, $g(\boldsymbol{\theta})$ is a diffeomorphism.

Consider all critical points of $\boldsymbol{\theta}$, given by $\ker(w)$. For each $u \in \ker(w)$, let $B_u$, where $u$ indexes the point, be the open ball derived from the center manifold theorem, and let $B = \cup_u B_u$. Since $\Theta \bigotimes \Sigma \subseteq \mathbb{R}^m$, Lindelöf's lemma indicates that every open cover has a countable sub-cover. That is, for a countable set of critical points $\{u_i\}_{i=1}^\infty$ with $u_i \in \ker(w)$, we have that $B = \cup_{i=1}^\infty B_{u_i}$.

Starting from some point $\boldsymbol{\theta_0} \in \Theta \bigotimes \Sigma$, if our methods converges to a strict saddle point,

then there exists a $t_0$ and index such that $g_t(\boldsymbol{\theta}) \in B_{u_i}$. Applying the center manifold theorem and using that $g(\boldsymbol{\theta}) \subseteq \Theta \bigotimes \Sigma$, we get that $g(\boldsymbol{\theta}_0) \in W_{loc}^{cs} \cap \Theta \bigotimes \Sigma$.

Using the fact that $g(\boldsymbol{\theta})$ is invertible, we can iteratively construct the sequence of sets defined by $W_1(u_i) = g^{-1}\left(W_{loc}^{cs} \cap \Theta \bigotimes \Sigma\right)$ and $W_{k+1}(u_i) = g^{-1}(W_k(u_i) \cap \Theta \bigotimes \Sigma)$. Then we have that $\boldsymbol{\theta}_0 \in W_t(u_i)$ for all $t \geq t_0$. The set $\Theta_0 \bigotimes \Sigma_0 = \cup_{i=1}^{\infty} \cup_{t=0}^{\infty} W_t(u_i)$ contains all the initial points in X such that our method converges to a strict saddle.

Since $u_i$ is a strict saddle, $I - \alpha J_{\mathcal{S}}(\boldsymbol{\theta})$ has an eigenvalue greater than 1. This implies that the co-dimension of the unstable manifold is strictly less than m—i.e., $\dim(W_{loc}^{cs}) < m$. Hence, $W_{loc}^{cs} \cap X$ has Lebesgue measure zero.

Using again that $g(\boldsymbol{\theta})$ is a diffeomorphism, $g^{-1} \in C^1$ so that it is locally Lipschitz and locally Lipschitz maps are null set preserving. Hence, $W_K(u_i)$ has measure zero for all k by induction so that $\Theta_0 \bigotimes \Sigma_0 = \cup_{i=1}^{\infty} \cup_{t=0}^{\infty} W_t(u_i)$ is a measure zero set since it is a countable union of measure zero sets.

$\square$

## G  ENVIRONMENTS SETUP

We list some important statstics about the environments as shown in Table 6 and the detailed dynamics of each environments can be found in (Duan et al., 2016; Tunyasuvunakool et al., 2020).

Moreover, we also list hyper-parameters of our methods and baselines as shown in Tables 4 and 5.

Here, we would like to discuss more details on why SDDPG-BD is slow comparing with FSDDPG-BD. Theoretically speaking, since it needs to compute the partial of the Hessian $\frac{\partial^2 Q_w^{\pi_\theta}(s,a)}{\partial w\theta}$, its convergence rate is slow than ours as mentioned in Sec. 4.3.

But the main reason makes SDDPG-BD slow is that the current deep learning framework (e.g., PyTorch (Paszke et al., 2019)) is hard to implement $diag^o(\frac{\partial^2 L(\theta,w)}{\partial w^2})$ in parallel through GPU. That is, we need to calculate the elements of Hessian matrices on diagonal block matrix. However, the torch.autograd.grad[11] function does not support calculating the gradient of a series of gradients in parallel, meaning that we need to loop them one by one. This makes the training process slow.

The RL platform is based on OpenAI's Spinning UP (Achiam, 2018). The codes for DDPG and TD3 are from `https://spinningup.openai.com/`.

## H  EXTRA EXPERIMENTS

We conduct the full version of SDDPG on two of the simplest environments in MuJoCo: Inverted-DoublePendulum, as well as InvertedPendulum. Results (Fig. 6) reveal that the SDDPG converges slowly than ours, indicating that the removing the TD-related terms do help improve the convergence rate of Stackelberg learning.

We also show toy example on the best response landscapes. We find that SDDPG can always keep in the best response region.

As shown in Fig. 8, we can find that SDDPG-CG has smaller losses for the critic, meaning that it does keep $w$ in the best response area. For DDPG, the losses of the $Q$ values increase dramatically, revealing that it may drifts far form the best response area. For ours, the loss of FSDDPG is between SDDPG-CG and DDPG. This coincide with the results in Fig. 1, which also supports the assumption that the TD-related term pulls the agent to remain in best repsonse area in Sec. 4.

---

[11]`https://pytorch.org/docs/stable/generated/torch.autograd.grad.html`

Table 4: The hyper-parameters of SDDPG (ours), SDDPG, and DDPG.

| | SDDPG (ours) | SDDPG | DDPG |
|---|---|---|---|
| Critic Learning Rate | $10^{-3}$ | $10^{-3}$ | $10^{-3}$ |
| Actor Learning Rate | $10^{-3}$ | $10^{-3}$ | $10^{-3}$ |
| Target Update Rate | $5 \times 10^{-3}$ | $5 \times 10^{-3}$ | $5 \times 10^{-3}$ |
| Batch Size | 100 | 100 | 100 |
| Discount Factor | 0.99 | 0.99 | 0.99 |
| Exploration Policy | $\mathcal{N}(0, 0.1)$ | $\mathcal{N}(0, 0.1)$ | $\mathcal{N}(0, 0.1)$ |
| Network Sizes for Actor | $16 \times 16$ | $16 \times 16$ | $16 \times 16$ |
| Other Useful Parameters | block diagonal rate =1 | − | − |

Table 5: The hyper-parameters of STD3 (ours), TD3, and SDDPG-CG, FSDDPG-CG.

| | FSTD3 (ours) | TD3 | SDDPG-CG | FSDDPG-CG |
|---|---|---|---|---|
| Critic Learning Rate | $10^{-3}$ | $10^{-3}$ | $10^{-3}$ | $10^{-3}$ |
| Actor Learning Rate | $10^{-3}$ | $10^{-3}$ | $10^{-3}$ | $10^{-3}$ |
| Target Update Rate | $5 \times 10^{-3}$ | $5 \times 10^{-3}$ | $5 \times 10^{-3}$ | $5 \times 10^{-3}$ |
| Batch Size | 100 | 100 | 100 | 100 |
| Discount Factor | 0.99 | 0.99 | 0.99 | 0.99 |
| Exploration Policy | $\mathcal{N}(0, 0.1)$ | $\mathcal{N}(0, 0.1)$ | $\mathcal{N}(0, 0.1)$ | $\mathcal{N}(0, 0.1)$ |
| Network Sizes for Actor | $16 \times 16$ | $16 \times 16$ | $16 \times 16$ | $16 \times 16$ |
| Other useful Hyper-Parameters | block diagonal rate =1 | − | CG iterations=5 | CG iterations=5 |

Table 6: Some statistics about the environments.

| Environments | State Dimension | Action Dimension |
|---|---|---|
| Ant | 125 | 8 |
| HalfCheetah | 20 | 6 |
| Humanoid | 102 | 10 |
| InvertedDoublePendulum | 6 | 1 |
| InvertedPendulum | 3 | 1 |
| Swimmer | 13 | 2 |
| Reacher | 6 | 2 |
| Walker | 24 | 6 |

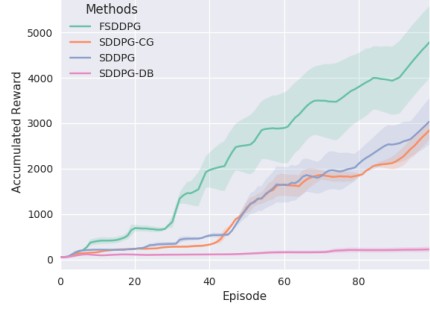
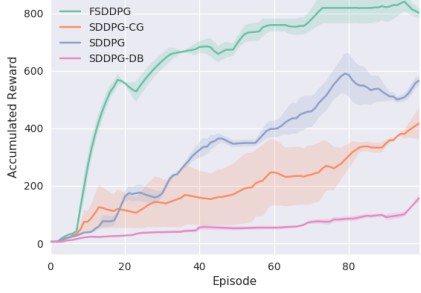

(a) InvertedDoublePendulum.
(b) InvertedPendulum.

Figure 6: Ablations on different Hessian approximation approaches and SDDPG approaches.

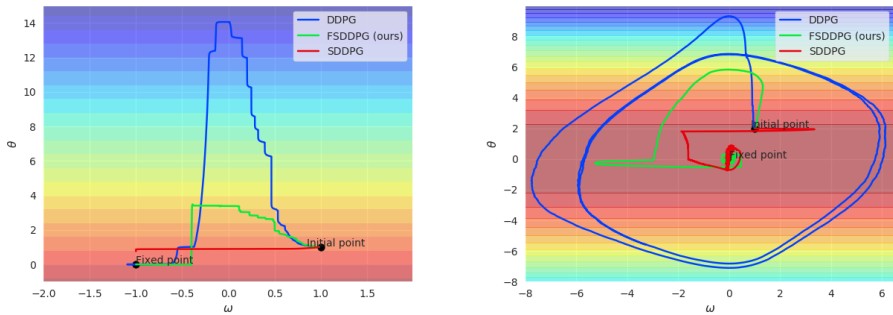

Figure 7: The trajectories of $\theta$ and $w$ for toy example 1 and toy example 2 (The contours are for $J(\theta, w^*)$ value).

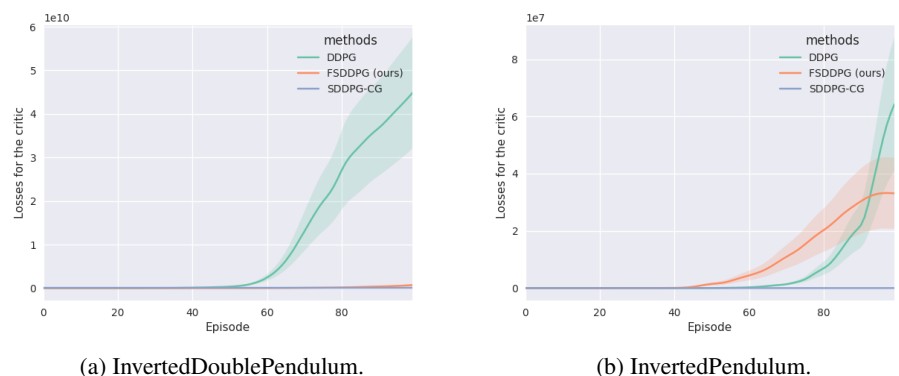

(a) InvertedDoublePendulum.

(b) InvertedPendulum.

Figure 8: Ablations on losses of the critics for different methods.

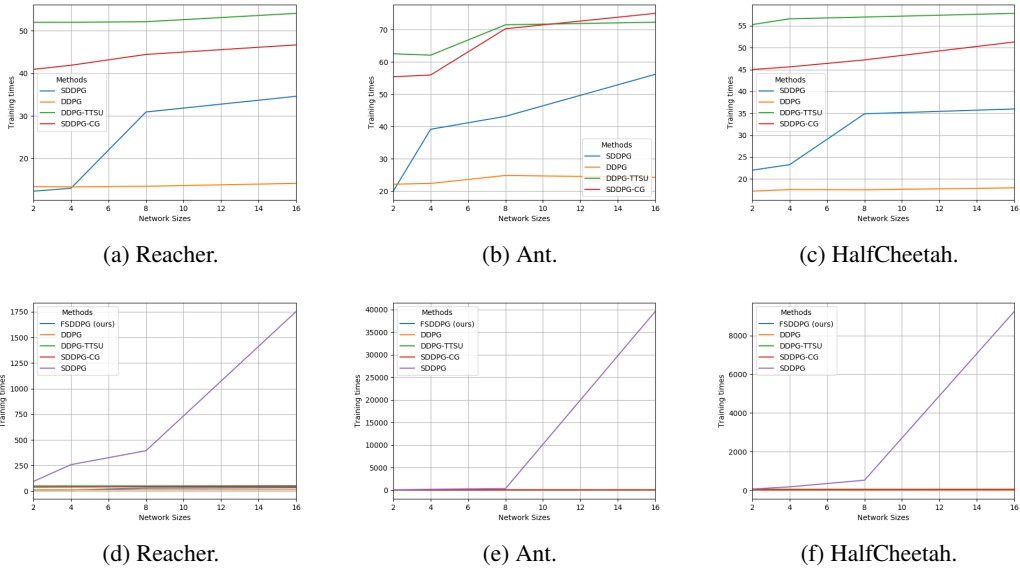

(a) Reacher.

(b) Ant.

(c) HalfCheetah.

(d) Reacher.

(e) Ant.

(f) HalfCheetah.

Figure 9: Average running time per episode of different methods. The first row (a-c) is the training time per episode of different methods without SDDPG. The second row (d-f) is the training time per episode of all the methods.

## I  SOME IMPORTANT SOURCE CODES

To make our reader know more about our algorithm. We put some important part of the source Codes here. One can implement our FSDDPG by just putting the following code to replace the corresponding functions in `https://github.com/openai/spinningup/blob/master/spinup/algos/pytorch/ddpg/ddpg.py`.

```python
def compute_loss_q(data):
    """Calculates the loss for the critic"""
    o, a, r, o2, d = data['obs'].to(device), data['act'].to(device),
                                      data['rew'].to(device), \
                    data['obs2'].to(device), data['done'].to(device)

    q = ac.q(o,a)

    # Bellman backup for Q function
    with torch.no_grad(): #stop-gradient operator for semi-gradient
                                      technique.
        q_pi_targ = ac_targ.q(o2, ac_targ.pi(o2))
        backup = r + gamma * (1 - d) * q_pi_targ

    # MSE loss against Bellman backup
    loss_q = ((q - backup)**2).mean()

    # Useful info for logging
    loss_info = dict(QVals=q.detach().cpu().numpy())

    return loss_q, loss_info

def calculate_s_actor_loss(data):
    """Calculates the Stackelberg gradient for the actor"""
    states = data['obs'].to(device)
    actions_pred = ac.pi(states)
    q = -ac.q(states, actions_pred).mean()
    dq_dws = autograd.grad(outputs=q, inputs=ac.q.parameters(),
                    create_graph=True)
    dq_dthetas = autograd.grad(outputs =q,
                            inputs = ac.pi.parameters(),
                            create_graph=True)

    # vectorize
    dq_dw_vec = torch.zeros(size=[1]).to(device)
    dq_dtheta_vec = torch.zeros(size=[1]).to(device)

    for dq_dw in dq_dws:
        dq_dw = torch.flatten(dq_dw)
        dq_dw_vec = torch.cat((dq_dw_vec,dq_dw),dim=0)
    dq_dw_vec = dq_dw_vec[1:].unsqueeze(0) #delete the first  element

    for dq_dtheta in dq_dthetas:
        dq_dtheta = torch.flatten(dq_dtheta)
        dq_dtheta_vec = torch.cat((dq_dtheta_vec,dq_dtheta),dim=0)
    dq_dtheta_vec = dq_dtheta_vec[1:].unsqueeze(0) #delete the first
                                      element

    k_1 = 2*torch.transpose(dq_dw_vec,dim0=0,dim1=1)@dq_dtheta_vec#
                                      Theta*w

    #deal with the diagonal block
    k_2_dia = 2 * dq_dw_vec * dq_dw_vec  # w*w
    rec_k_2_dia = torch.reciprocal(k_2_dia + 1e-5)

    #calculate the gradient for actor
    dq_dtheta_stack = torch.transpose(dq_dtheta_vec,
            dim0=0, dim1=1) \
```

```
              + torch.transpose(k_1, dim0=0, dim1=1) * \
                rec_k_2_dia @ torch.transpose(dq_dw_vec, dim0=0, dim1=1)
        dq_dtheta_stack = dq_dtheta_stack[:,0]

        for i,para in enumerate(ac.pi.parameters()):
            para.grad = dq_dtheta_stack[param_index[i]:param_index[i+1]].
                                          reshape(para.shape)
```

```python
    def update(data):
        # First run one gradient descent step for Q.
        q_optimizer.zero_grad()
        loss_q, loss_info = compute_loss_q(data)
        loss_q.backward()
        q_optimizer.step()

        # Manaully reset gradient
        for p in ac.pi.parameters():
            p.requires_grad = False
            p.requires_grad = True

        # SDDPG Next run one gradient descent step for pi.
        calculate_s_actor_loss(data)
        pi_optimizer.step()
```

