# OpenReview forum: "Fast Deterministic Stackelberg Actor-Critic"
_ICLR.cc/2022/Conference — ICLR 2022 Submitted_

### Official Review · Reviewer_jvHe · 2021-10-22

**Correctness:** 3
**Technical Novelty And Significance:** 2
**Empirical Novelty And Significance:** 3
**Recommendation:** 3
**Confidence:** 4

**Main Review:**

Strength: This paper seems to provide numerical experiments on Mujoco that are comparable to the state-of-the-art.

Weakness:

1. The formulation of Stackelberg game between the actor and critic seems problematic. In specific, the actor $\pi_{\theta}$ is the leader and the critic $Q_{\omega}$ is the follower. The follower's loss of the mean-squared temopral difference error
\begin{align}
L(\theta, \omega) = \mathbb{E} _ {s \sim \rho_{\theta}, a\sim \pi_{\theta}} \big  [ ( Q_{\omega} (s,a) - Q^{\pi_{\theta} } (s,a))^2 \bigr ] ,
\end{align}
where $\rho_{\theta}$ is the visitation measure or stationary distribution induced by the policy $\pi_{\theta}$. Then, when calculating the indirect gradient term in Eq. (3), we  need to handle the partial direvative $\partial ^2 L(\theta, \omega) / \partial \theta \partial \omega$, which involves taking the gradient with respect to $(s,a) \sim \rho_{\theta} \otimes \pi_{\theta}$ and needs to be handled using policy gradient thoerem. This work seems to neglect this matter by using a fixed sampling distribution
$\rho$, which is problematic.

2. More importantly, I feel that the issue raised in this work:
> Most AC methods perform stochastic gradient descent on
the actor and critic simultaneously. This can be regarded as performing Gradient Descent Ascent
(GDA) (Zheng et al., 2021; Wen et al., 2021), which is known to suffer from convergence to limit
cycles or bad locally optimal solutions (Wang et al., 2019; Jin et al., 2020; Balduzzi et al., 2018).
Moreover, Yang et al. (2019) show that the critic is biased and may not converge when the actor and
critic are updated simultaneously with similar learning rates.
seems not well justified. First, it seems [Yang et al. (2019)] does not provide a proof showing biased critic leads to divergence. Second, there are various recent works showing that actor-critic converges in finite-time. See [1,2,3,4]. It seems unclear what advantage such a Stackelberg view brings to the analysis.

3. It seems that the Stackelberg formulation of actor-critic used in this work is proposed in [Zheng et al., 2021]. The main contribution of this work is to (i) additionally use deterministic policy gradient and drop a term involving TD error, and (ii) use block diagonal approximation in the calculation of the inverse Hessian.

4. The theoretical results and the proof seem hard to understand. It seems Theorem 1 directly follows from [Fiez et al., 2020] and convex analysis. Moreover, the assumption of positive-definiteness seems very strong. It would be nice to provide examples on which the assumptions hold. In addition, how is the stepsize separation $\tau$ chosen and how does it affect the theory?

References:

[1] A Finite Time Analysis of Two Time-Scale Actor Critic Methods - Yue Wu, Weitong Zhang, Pan Xu, Quanquan Gu

[2] Improving Sample Complexity Bounds for (Natural) Actor-Critic Algorithms - Tengyu Xu, Zhe Wang, Yingbin Liang

[3] A Two-Timescale Framework for Bilevel Optimization: Complexity Analysis and Application to Actor-Critic - Mingyi Hong, Hoi-To Wai, Zhaoran Wang, Zhuoran Yang

[4] An Improved Analysis of (Variance-Reduced) Policy Gradient and Natural Policy Gradient Methods - Yanli Liu, Kaiqing Zhang, Tamer Basar, Wotao Yin

**Summary Of The Paper:**

This work proposes a variant of the Stackelberg actor-critic algorithm, which treats actor-critic as a Stackelberg game with the actor being the leader and critic being the follower. Based on Stackelberg AC, this paper proposes to adopt deterministic policies in order to simplify the computation of the implicit gradient. Additionally, the authors propose approximation schemes including dropping small-order terms and matrix inversion via block diagonal matrix approximation. The computational complexity of gradient computation is established. Moreover, under relatively strong assumptions, the proposed method is shown to be convergent. Experiments on Mujoco are conducted to demonstrate the efficacy of the method.

**Summary Of The Review:**

a. The Stackelberg view of actor-critic seems not well motivated.

b. There exists a small issue in the implicit gradient involving the sampling distribution.

c. The theoretical arguments depend on strong and ungrounded assumptions and the proofs are hard to follow.

---

> ### Author Response · Authors · 2021-11-22
> **Response to Reviewer jvHe**
>
> Thank you very much for your detailed and valuable suggestions!
>
> ---
> **Q1:  This work seems to neglect this matter by using a fixed sampling distribution, which is problematic.**
>
> We use the off-policy deterministic policy gradient scheme as in (Lillicrap et al., 2016).
> Thus, $\displaystyle \rho $ is the stationary distribution induced by the
> behavior policy $\displaystyle \beta $,
> and is not related to $\displaystyle \pi _{\theta }$.
> Data is obtained
> according to this behavior distribution $\rho $.
> The same setting is also used in [5] and [Silver et al., 2014].
> In this revised version, we added more details on this in the first paragraph of
> Sec. 3.
>
> ---
> **Q2: Seems [Yang et al. (2019)] does not provide a proof showing biased critic leads to divergence. Second, there are various recent works showing that actor-critic converges in finite-time. See [1, 2, 3, 4]. It seems unclear what advantage such a Stackelberg view brings to
> the analysis. We agree that  [Yang et al. (2019)]  does not provide a proof.
> In  Sec. 1 of this revised version, we changed the sentence to "Moreover,
> as mentioned in [Yang et al. (2019)] , the GDA training scheme is fragile, which could lead to biased critic and may not converge".**
>
> We agree that
> [Yang et al. (2019)]
> does not provide a proof.
> In Sec. 1 of
> this revised version, we changed the sentence to "Moreover,
> as mentioned in [Yang et al. (2019)] , the GDA training scheme is fragile, which could lead to biased critic and may not converge".
>
> References [1,2,3,4] are on the two time scale update scheme (TTSU),
> in which the inner loop stepsize is smaller than that in the outer loop.
> Therefore, the actual training time
> is long
> since the inner loop stepsize is small.
>
> Our empirical results in Fig. 3 (Sec. 6) and Fig. 9 (Appendix H) also show that
> DDPG-TTSU is slower than DDPG (Discussions are in Sec. 6.2).
> The Stackelberg scheme can make training faster since it uses one-time scale update.
> Also, comparing with the vanilla one-time scale update GDA, the Stackelberg scheme is more stable.
> More details can be found in the first paragraph of Sec. 1 and the first
> paragraph of Sec. 2.
> ---
> **Q3: It seems that the Stackelberg formulation of actor-critic used in this work is proposed in [Zheng et al., 2021]**
>
> Besides the two contributions mentioned by the reviewer,
> there are other differences with
> [Zheng et al., 2021]. Specifically,
> [Zheng et al., 2021] uses
>
>
> \begin{equation*}
>  E_{s\sim \rho (\cdot ),a\sim \beta, s'\sim p(\cdot|s,a) }\left[\left( Q_{w}^{\pi_{\theta}} (s,a)-r( s,a) -\gamma Q_{w}^{\pi _{\theta }} (s',\pi _{\theta} (s')\right)^{2}\right]
> \end{equation*}
>
> as the loss of critic,
> which differs from the critic's loss in DDPG.
> On the other hand,
> we follow the critic loss in
> DDPG:
>
>
> \begin{eqnarray*}
> \hat{L}(\theta,w) =   E_{s\sim \rho (\cdot ),a\sim \beta }[( Q_{w}^{\pi_{\theta }} (s,a) -\perp[Q^{\pi_{\theta }} (s,a)])^{2}]
> \end{eqnarray*}
> \begin{eqnarray*}
> \approx   E_{s\sim \rho (\cdot ),a\sim \beta,s'\sim p(\cdot|s,a) }[( Q_{w}^{\pi_{\theta }} (s,a)-r( s,a) -\gamma \perp[Q_{w}^{\pi_{\theta }} (s',\pi _{\theta } (s')])^{2}],
> \end{eqnarray*}
>
> where
> $\perp(\cdot)$ is the stop-gradient operator, and $\beta$ is the behavior policy.
> The resultant derivations and update schemes are thus also different from [Zheng et al., 2021].
> ---
> **Q4: The theoretical results and the proof seem hard to understand.**
>
> In this revised version, we added
> 1) Explanations to
> the stop-gradient operator
> in the proofs of Proposition 1 of Appendix A.
>
> 2) Remark 1 in Appendix A to discuss the different derivatives between
> DPG and ours.
>
> 3) A conclusion at the end of Appendix B to show how Lemma 4 can guide us to select
> the diagonal degree $o$.
>
> 4) More explanations in the proofs of Theorems 2 and 3.
>
> 5) Updated Corollary 1 and Theorem 1 in Sec. 5, and added Proposition 2 in
> Appendix D to match the revised update scheme.
>
> ---
> **Q5 It seems Theorem 1 directly follows from [Fiez et al., 2020].**
>
> It mainly follows Theorem 4 in [Fiez et al., 2020].
> But they differ in the
> definitions of the dynamics $ \xi (\boldsymbol{\theta })$ and its Jacobian
> $J_{\mathcal{S}} (\boldsymbol{\theta } ) $ (defined at the beginning of Sec. 5)
> which are specific to our settings.
> ---
> **Q6: The assumption of positive-definiteness seems very strong.**
>
> This assumption is introduced to Stackelberg learning in [Fiez et al., 2020], and also used in [Wang et al., 2019].
>
>
>  ---
> **Q7: How is the stepsize separation $\tau$  chosen?**
>
> The choice of $\tau$ is the same as in [Fiez et al., 2020].
>
>  ---
> References
>
> [5] Degris, Thomas, Martha White, and Richard S. Sutton. "Off-policy actor-critic." ICML. 2012.

---

> > ### Comment · Reviewer_jvHe · 2021-11-23
> > **futher response**
> >
> > I would like to thank the authors for responding to my comments.
> >
> > **Re: Q1**
> > - It seems that setting the sampling distribution **does not** resolve your problem. In the definition of $L$, the expectation is taken with respect to $s\sim \rho$ and $a\sim \pi(\cdot | s)$. What is this policy $\pi$? As seen from the text, it should be the current policy $\pi_{w}$. Then you need to additionally consider the gradient involving this term when taking the gradient w.r.t. $w$. Moreover, defining $L$ using $\rho$, instead of the stationary distribution induced by $\pi_w$ seems to deviate from the existing literature of policy evaluation. When $\rho$ does not have sufficient coverage, it seems $L$ can have other global minimizers in addition to $Q^{\pi_w}$.
> >
> > **Re: Q4**
> > - How is $\perp$ defined mathematically? It seems not to change the value of $Q^{\pi}$, but somehow it changes the gradient. Also, your equation following "On the other hand, we follow the critic loss in DDPG:" seems inaccurate. When replacing the mean-squared error by the mean-squared Bellman error, there will be a bias, and the bias might be large.

---

> > > ### Author Response · Authors · 2021-11-24
> > > **Further Response to Reviewer jvHe (Part 1/2)**
> > >
> > > We really appreciate your further comments!
> > >
> > > We now explain your concerns and answer your questions point by point.
> > >
> > > ---
> > > **Q8: In the definition of $ L$, the expectation is taken with respect to $s\sim \rho $ and $a\sim \pi ( \cdot |s)$. What is this policy $\pi $? As seen from the text, it should be the current policy $\pi _{\theta }$.
> > > Then you need to additionally consider the gradient involving this term when taking the gradient w.r.t $\theta $.**
> > >
> > > i) We agree that in $L(\theta,w)$ and $J(\theta,w)$, the term $a\sim \pi ( \cdot |s)$ is actually $a\sim \pi _{\theta }( \cdot |s)$. We will revise $J(\theta ,w)$ and $L(\theta ,w)$ as:
> > >
> > > $J(\theta ,w)=E_{s\sim \rho (s),a\sim \pi_{\theta } (\cdot \mid s)}\left[ Q_{w}^{\pi_{\theta }} (s,a)\right] ,L(\theta ,w)=E_{s\sim \rho (s),a\sim \pi_{\theta } (\cdot \mid s)}\left[\left( Q_{w}^{\pi_{\theta }} (s,a)-\perp \left[ Q^{\pi_{\theta }} (s,a)\right]\right)^{2}\right]$.
> > >
> > > ii) Regarding the comment "Then you need to additionally consider the gradient involving this term when taking the gradient w.r.t $\theta $".
> > >
> > > Note that the equations mentioned above are for the stochastic policy, different from the deterministic policy setting we focus on. As mentioned in Page 3, Sec. 3, under the deterministic policy setting, $\pi_\theta(\cdot|s)$ becomes $\delta(a - \pi _{\theta }( s))$, where $\delta$ is a Dirac-Delta function, and $\pi _{\theta }( s)$ is the deterministic policy. Consequently, $J(\theta ,w)$ as well as $L(\theta ,w)$  degenerate to:
> > >
> > >  $ L( \theta ,\omega ) \ =\ E_{s\sim \rho (s)}\left[\left( Q_{w}^{\pi_{\theta }} (s,a)-\perp \left[ Q^{\pi_{\theta }} (s,a)\right]\right)^{2}\right] ,$ $J( \theta ,\omega ) \ =\ E_{s\sim \rho (s)}\left[ Q_{w}^{\pi_{\theta }} (s,a)\right]$,
> > >
> > >  s.t. $\ a=\pi _{\theta }( s)$.
> > >
> > >  The proof can be done by using the property of the Dirac-Delta function:
> > >
> > >  $\int_A \pi_\theta(a|s) g(s,a) da = \int_A \delta(\pi_\theta(s)-a) g(s,a) da = g(s,\pi_\theta(s))$,
> > >
> > >  where $g(s,a)$ is a function with input $s,a$. By setting $g(s,a)$ as $Q_{w}^{\pi_{\theta }} (s,a)-\perp [ Q^{\pi_{\theta }} (s,a)]$ and $Q_{w}^{\pi_{\theta }} (s,a)$ respectively, we can obtain the degenerated $L$ and $J$.
> > >
> > > Thus, we do not need to consider the extra term in $\pi_\theta(\cdot|s)$.
> > > The gradient results in Lemma 1 as well as Proposition 1 in Appendix A are exact.
> > >
> > > ---
> > > **Q9: Moreover, defining $L$ using $\rho$, instead of the stationary distribution induced by  $\pi_\theta$ seems to deviate from the existing literature of policy evaluation. When $\rho $ does not have sufficient converage, it seems $L$ can have other global minimizers
> > > in addition to $Q^{\pi _{\theta }}$.**
> > >
> > >
> > > Our off-policy setting follows Eq. (3) in [5] and Eq. (14) in [Silver et al., 2014],
> > > which are also widely used in various DPG family, e.g., [Lillicrap et al., 2016] and [6,7].
> > > Besides the DPGs, similar off-policy settings are also used in [15,16].
> > >
> > > ---
> > > **Q10: How is $\perp $ defined mathematically?**
> > >
> > > The definition of $\perp $ follows [8,9,10].
> > > Formally speaking, $\perp $ is defined as:
> > >
> > > $ \forall i\in \\{ 1,n \\} ,\ \nabla_{x_{i}} \perp ( f( x_{1} ,x_{2} ,\dotsc ,x_{i} \dotsc ,x_{n})) \ \equiv \ 0,$
> > >
> > > where $f$ can be any differential function.
> > >
> > > We will add the formal definition to the main text.
> > >
> > > ---
> > > **Q11: It seems not to change the value of $Q^{\pi }$, but somehow it changes the gradient.
> > > Also, your equation following "On the other hand, we follow the critic loss in DDPG:" seems inaccurate.**
> > >
> > > Our critic's loss $L(\theta,w)$ is equivalent to the critic's loss used in DDPG (Eq. (4) in [Lillicrap et al., 2016]).
> > >
> > > What the reviewer concerns about seems to be whether the gradient between $\hat{L}(\theta,w)$ and Eq. (4) in [Lillicrap et al., 2016] are equal. The answer is yes.
> > >
> > > Note that [Lillicrap et al., 2016] also leverage the semi-gradient technique to their critic's loss (mentioned in the sentence below Eq. (5) in [Lillicrap et al., 2016]), and consequently for DDPG,
> > >
> > > $\nabla_w [r(s,a)+\gamma Q_{w}^{\pi _{\theta }} (s',\pi _{\theta } (s')] = 0$. So the gradient of the critic's loss w.r.t $w$ is:
> > >
> > > $2E_{s\sim \rho (s),a\sim \beta ,s'\sim p(\cdot |s,a)} [(Q_{w}^{\pi_{\theta }} (s,a)-r(s,a)-\gamma Q_{w}^{\pi_{\theta }} (s',\pi_{\theta } (s'))\nabla_{w} Q_{w}^{\pi_{\theta }} (s,a)],$
> > >
> > > For ours, we can easily check that:
> > >
> > > $ \nabla_{w}\hat{L}( \theta ,w)=\ 2E_{s\sim \rho (s),a\sim \beta ,s'\sim p(\cdot |s,a)} [(Q_{w}^{\pi_{\theta }} (s,a)-r(s,a)-\gamma \perp [Q_{w}^{\pi_{\theta }} (s',\pi_{\theta } (s')])\nabla_{w} Q_{w}^{\pi_{\theta }} (s,a)],$
> > >
> > > which is equal to the gradient of the critic's loss for DDPG.
> > >
> > > Moreover, many widely-adopted DDPG implementations (e.g., [12] and [Achiam, 2018]) also apply the stop-gradient operator to obtain the semi-gradient of the critic's loss (e.g., lines 166-170 in [13] as well as lines 169-192 in [14]), which is the same as our $\hat{L}( \theta ,w)$.

---

> > > > ### Author Response · Authors · 2021-11-24
> > > > **Further Response to Reviewer jvHe (Part 2/2)**
> > > >
> > > > (Continued)
> > > >
> > > > ---
> > > > **Q12: "When replacing the mean-squared error by the mean-squared Bellman error, there will be a bias, and the bias might be large.**
> > > >
> > > > As discussed in our reply to Q11, the semi-gradient technique we used (replacing the mean-squared error by the mean-squared Bellman error) follows DDPG [Lillicrap et al., 2016].
> > > >
> > > > In fact, this technique are widely adopted in various methods, e.g., [6,16,17], as well as classical methods mentioned in Sec. 9.3 and Sec. 11.1 (Sutton & Barto, 2018).
> > > >
> > > >
> > > > Moreover, we are not the first one that use semi-gradient to Stackelberg Actor Critic. (Wen et al., 2021) also use the semi-gradient method to calculate the gradient for both actor and critic in Sec. 5.2 and 5.3.  Specifically, as mentioned in Sec. 5.2, under some assumptions, when using the semi-gradient for the critic, the Stackelberg gradient of the policy can still be equal to its true policy gradient (Wen et al., 2021).
> > > >
> > > > ---
> > > > **In conclusion, our off-policy settings and the definition of our critic's loss are all directly follow DDPG.**
> > > >
> > > > References:
> > > >
> > > > [6] Pan, Ling, Qingpeng Cai, and Longbo Huang. "Softmax deep double deterministic policy gradients." NeurIPS, 2020.
> > > >
> > > > [7] Lowe, Ryan, et al. "Multi-Agent Actor-Critic for Mixed Cooperative-Competitive Environments." NeurIPS, 2017.
> > > >
> > > > [8] Foerster, Jakob, et al. "DICE: The infinitely differentiable monte carlo estimator." ICML, 2018.
> > > >
> > > > [9] van den Oord, Aaron, Oriol Vinyals, and Koray Kavukcuoglu. "Neural discrete representation learning." NeurIPS, 2017
> > > >
> > > > [10] Tian, Yuandong, Xinlei Chen, and Surya Ganguli. "Understanding self-supervised learning dynamics without contrastive pairs." ICML, 2021.
> > > >
> > > > [12] Liang, Eric, et al. "RLlib: Abstractions for distributed reinforcement learning." ICML, 2018.
> > > >
> > > > [13] https://github.com/openai/spinningup/blob/master/spinup/algos/tf1/ddpg/ddpg.py#L166-L170
> > > >
> > > > [14] https://github.com/ray-project/ray/blob/39c598bab0929f007a49daa7262a7c9ee6492221/rllib/agents/ddpg/ddpg_tf_policy.py#L169-L192
> > > >
> > > > [15] Imani, Ehsan, Eric Graves, and Martha White. "An off-policy policy gradient theorem using emphatic weightings." NeurIPS 2018.
> > > >
> > > >
> > > > [16] Sutton, Richard S., A. Rupam Mahmood, and Martha White. "An emphatic approach to the problem of off-policy temporal-difference learning." JMLR (2016).
> > > >
> > > > [17] Haarnoja, Tuomas, et al. "Soft actor-critic: Off-policy maximum entropy deep reinforcement learning with a stochastic actor." ICML 2018.

---

### Official Review · Reviewer_WR2J · 2021-11-01

**Correctness:** 3
**Technical Novelty And Significance:** 2
**Empirical Novelty And Significance:** 2
**Recommendation:** 5
**Confidence:** 3

**Main Review:**

Strengths: This paper is well-written and organized. The motivation and intuition of the proposed method are clear and easy to follow. Various experiments are presented to compare the proposed method with baselines.

Weaknesses: My concerns about this paper are three-folds. Firstly, the novelty. The theoretical analysis in Section 5 mainly follows Theorem 5 in (Fiez et al., 2020), the authors should clearly state the connections of these two works. Second, the theoretical analysis is pretty weak. They only show a faster convergence rate, which is an expected and relatively weak result. They don't show any theoretical results for stability, which is stated as one of their two main contributions (faster convergence rate and maintained stability). Third, in terms of the experiments, stability is mainly illustrated and validated only on toy examples, hence less convincing.


**Summary Of The Paper:**

This paper proposes a method to accelerate Stackelberg Actor-Critic by ignoring the terms that contain Hessian in the indirect gradient term and applying block-diagonal approximation technique to remaining inverse terms. They prove a faster convergence rate of the proposed method to a fixed point. Experiments are conducted to validate the fast convergence and stability.

**Summary Of The Review:**

In all, although this paper is clear to follow and nicely presented, I believe the theoretical analysis part is weak. The fast convergence rate somehow lacks significant novelty. The stability of the proposed method is less convincing.

---

> ### Author Response · Authors · 2021-11-22
> **Response to Reviewer WR2J**
>
> Many thanks for your insightful and useful comments!
>
> ---
> **Q1: The theoretical analysis in Section 5 mainly follows Theorem 5in (Fiez et al., 2020).**
>
> Only the proof in Theorem 1 follows (Fiez et al., 2020).The proofs for
> Corollary 1 and Theorem 2 are very different.
>
> ---
> **Q2: Connections of our paper and (Fiez et al., 2020).**
>
> In the theoretical analysis, we have used tools from (Fiez et al., 2020)
> and e.g.,  (Henderson & Searle, 1981; Reddi et al., 2018).
> Note that
> the proposed method and relaxation techniques we used are novel and not from (Fiez et al., 2020).
>
> ---
> **Q3: They only show a
> faster convergence rate, which is an expected and relatively weak result.
> They don't show any theoretical results for stability.**
>
> To our best knowledge, we are the first to analyze faster convergence for approximated
> Stackelberg training.
>
> Regarding stability,
> indeed, we do have provided
> analysis.
> As mentioned at the end of Sec. 5,
> we put them in
> Appendix F
> because of lack of space.
> Specifically,
> Theorem 4 there
> shows that the proposed method can almost surely avoid saddle
> points, which are similar to the
> stability results in
> Theorem 4 of (Fiez et al., 2020) and Corollary 23 of (Mazumdar et al., 2020).
>
> ---
> **Q4: In terms of the experiments, stability is mainly illustrated and validated only on toy examples**
>
> Limited cycles are likely to be periodic in the low-dimensional space of
> neural network
> parameters
> [1],
> and so
> the stability problem can be more easily observed.
> In high-dimensional spaces,
> orbits of the limited cycles are "strange"
> and can be
> neither cyclic nor quasi-periodic
> (Sec. 4.1 of [1]).
> Existing works (such as
> [2,3,4] and [Fiez et al., 2020]) also
> only use toy examples to visualize the
> stability problem.
>
> ---
> References
>
> [1] Fuchs, Armin. Nonlinear dynamics in complex systems. Berlin: Springer, 2014.
>
> [2] Daskalakis, Constantinos, et al. "Training GANs with Optimism." ICLR 2018.
>
> [3] Fiez, Tanner, and Lillian Ratliff. "Local Convergence Analysis of Gradient Descent Ascent with Finite Timescale Separation." ICLR 2021.
>
> [4] Zhang, Guojun. "Understanding Minimax Optimization in Modern Machine Learning." PhD thesis, University of Waterloo (2021).

---

### Official Review · Reviewer_7evM · 2021-11-01

**Correctness:** 4
**Technical Novelty And Significance:** 3
**Empirical Novelty And Significance:** 4
**Recommendation:** 8
**Confidence:** 2

**Main Review:**

This paper is well-organized, and the discussion of the literature background is thorough. The proposed method is novel and sound, which has solid theoretical analysis and empirical grounding. Overall, I recommend accepting this paper.

Minor comments:

- Several curves in Figure 3 do not seem to converge, e.g, in InvertedDoublePendulum and Walker. It would be better to include a longer training horizon to establish a rigorous comparison.

- In Table 1, how does the scale of $\epsilon$ compare to $\frac{1}{nm}$ and $\frac{1}{o}$ in practice?


**Summary Of The Paper:**

This paper aims to establish a computational efficient Stackelberg training scheme for deep-learning-based actor-critic methods. The proposed approach,  Fast Stackelberg Deep Deterministic Policy Gradient (FSDDPG), considers the block diagonal approximation technique to reduce the training complexity while improving the convergence rate. This paper conducts both theoretical analysis and empirical evaluation to demonstrate the strength of the proposed approach.

**Summary Of The Review:**

To my knowledge, the method proposed by this paper is novel and tackles an important problem. I recommend acceptance.

---

> ### Author Response · Authors · 2021-11-22
> **Response to Reviewer 7evM**
>
> Thank you very much for your helpful and valuable comments!
>
> ---
> **Q1: Several curves in Figure 3 do not seem to converge. ...
> include a longer training horizon.**
>
> As suggested, we increased
> the training horizon to 210 episodes for InvertedDoublePendulum and 240 episodes
> for Walker. The updated results are shown in Fig. 3.
> As can be seen, the proposed FSDDPG still outperforms DDPG and SDDPG-CG, and the
> proposed FSTD3 still outperforms STD3-CG and TD3.
>
> ---
> **Q2: How does the scale of  $\epsilon$ compare to $\frac{1}{n m}$
> and  $\frac{1}{o} $ in practice?**
>
> We do not choose
> $\epsilon$ directly as
> its value is often hard to pick. A large $\epsilon$  may lead to
> bad solution,
> while using a small $\epsilon$  is computationally expensive.
>
> In the experiments, instead of setting
> $\epsilon$,
> we follow (Zheng et al., 2021; Wen et al., 2021) and set a maximum on the number
> of CG iterations $I$ (which is equal to 5), which is larger than $o=1$.

---

### Official Review · Reviewer_6tEQ · 2021-11-02

**Correctness:** 2
**Technical Novelty And Significance:** 2
**Empirical Novelty And Significance:** 2
**Recommendation:** 3
**Confidence:** 4

**Main Review:**


I found this paper in general well written and the method presented seems to be novel. The authors analyzed many different Hessian approximation scenarios and I believe that this line of work can be relevant in order to make the Stackelberg approach feasible for larger neural networks.

Unfortunately, I found a few issues with the theory related to DDPG and the way experiments have been conducted, which might violate most of the proposed theoretical results.

The main issue with this paper is that it considers the approximated off-policy policy gradient of DDPG, which appears here in equation (4) as if it was the true policy gradient.

I want to discuss two cases and explain why equation (4) is not correct:

- In the on-policy case, the expectation of the left side of equation (4) should be over the distribution on the initial state of the agent, while on the right side it becomes the (improper) discounted state visitation under the current policy. This is because the gradient $\nabla_{\theta}Q^{\pi_{\theta}}(s,a)$ (here a is any action, not the deterministic one) can be iteratively decomposed (see Theorem 1 in [1] and  Theorem 1 in [2]).

- In the off-policy case, however, since we do not have samples from the stationary distribution under the current policy, the expectation on the left side of equation (4) is taken over the discounted state visitation under the behavioral policy, here called $\rho(s)$. If this is the case, the gradient  $\nabla_{\theta}Q^{\pi_{\theta}}(s,a)$ is difficult to compute and it is usually dropped (see Eq. 15 in [1] and comment below, see Section 2.2 and Appendix B Errata in [3]). Therefore the gradient is only an approximation of the true off-policy policy gradient. This is what happens in off-policy DPG, DDPG and TD3. So, in order to have equality in equation (4), the term  $\nabla_{\theta}Q^{\pi_{\theta}}(s,a)$ must be added. Formally, equation 4 should be: $\nabla_{\theta} \mathbb{E}_{s \sim \rho(\cdot)} [Q^{\pi_{\theta}}(s, \pi_{\theta}(s))] = \mathbb{E}_{s \sim \rho(\cdot)} [\nabla_{\theta}\pi_{\theta}(s) \nabla_a Q^{\pi_{\theta}}(s,a)|_{a=\pi_{\theta}(s)} + \nabla_{\theta}Q^{\pi_{\theta}}(s,a)|_{a=\pi_{\theta}(s)}]$.

This issue is present in most of the proofs, where equation (4) is used ignoring the additional term, so the results of proposition 1, Theorem 1, Theorem 2, Corollary 1, should not hold in the off-policy setting. Note that the experiments performed are off-policy and use a replay buffer containing past trajectories.

In the problem formulation, it is not clear how $\rho(s)$ is defined. The authors claim it is the "discounted state distribution of s", but under which policy? The definition of the action-value function should not depend on the initial state.

The authors define $Q_w^{\pi_{\theta}}(s,a)$ to be equal to $V_w^{\pi_{\theta}} (s)$, because they always consider the deterministic action of the policy. This can be seen also in the minimization of the TD error in equation (1), where the expectation is never taken over the actions in the replay buffer, but only over the states ($\nabla_w \mathbb{E}_{s \sim \rho(\cdot)} [(Q^{\pi_{\theta}}_w(s, \pi_{\theta}(s)) - Q^{\pi_{\theta}}(s, \pi_{\theta}(s)) )^2]$). Note how this differs from DDPG, where the target value function is deterministic, but the learned value function is evaluated on a set of noisy actions. This is relevant here because when taking the second order gradient, the authors must use the chain rule for both terms in the temporal difference loss, while this would not be the case in standard DDPG. Note that in the experiments the authors are instead using the standard DDPG approach and learn a full action-value function, sampling state-actions pairs from the replay buffer. This is evident also in Table 4 where a noisy version of the policy is used for exploration. How is this affecting the theoretical results?

In the proof of Proposition 1, why is the term $\nabla_{\theta^T} Q^{\pi_{\theta}}(s,a)$ disappearing?

I would like to note that this paper is building on the work of Zheng et. al. [4], which appeared on ArXiV only 3 days before the first ICLR submission deadline. I was not able to verify if [4] has been peer-reviewed, but given the issues above, I believe that [4] might contain similar theoretical problems.

[1]David Silver, Guy Lever, Nicolas Heess, Thomas Degris, Daan Wierstra, and Martin Riedmiller. Deterministic policy gradient algorithms. In Proceedings of the 31st International Conference on International Conference on Machine Learning - Volume 32, ICML’14, pages I–387–I–395.JMLR.org, 2014.

[2]Sutton, R. S., McAllester, D. A., Singh, S. P., & Mansour, Y. (2000). Policy gradient methods for reinforcement learning with function approximation. In Advances in neural information processing systems (pp. 1057-1063).

[3]Thomas Degris, Martha White, and Richard S. Sutton. Off-policy actor-critic. In Proceedings of the 29th International Conference on International Conference on Machine Learning, ICML’12, pages 179–186, USA, 2012.Omnipress.

[4]Zheng, L., Fiez, T., Alumbaugh, Z., Chasnov, B., & Ratliff, L. J. (2021). Stackelberg Actor-Critic: Game-Theoretic Reinforcement Learning Algorithms. arXiv preprint arXiv:2109.12286.


**Summary Of The Paper:**

This paper improves upon Stackelberg Deep Deterministic Policy Gradients by proposing a set of strategies on how to deal with the Hessian part of the gradient. This should overcome the major limitations of this class of methods, which are the great time complexity and the slow converging rate with respect to the standard actor-critic framework. The authors provide a formal justification on why removing parts of the Hessian and using a block-diagonal approximation is still achieving convergence. Time complexity is analyzed for many variations of the algorithm and the experimental session provides mixed results.

**Summary Of The Review:**

While I find the paper well written and potentially relevant for this field, there is a major flaw in how the policy gradient is computed. It is not clear to me how the theoretical results can be fixed in the off-policy case, while for on-policy learning some additional work might be able to keep the results valid. Unfortunately, the experiments conducted are in the off-policy setting, hence, it is not clear what we can conclude from them. Given the issues above, I propose to reject this paper in the current version.

---

> ### Author Response · Authors · 2021-11-22
> **Response to Reviewer 6tEQ (Part 1/3)**
>
> Many thanks for your insightful and constructive advice!
>
> ---
> **Q1: The proposed method is on-policy or off-policy?**
>
> In this paper, following (Lillicrap et al., 2016), we only consider the off-policy setting.
>
> ---
> **Q2: How $\displaystyle \rho ( s)$ is defined?**
>
> We use the off-policy setting. So, following (Lillicrap et al., 2016),
> $\rho( s) $ is the discounted state distribution induced by the behavior policy $\beta$.
> More details
> are added
> in the first paragraph of Sec. 3.
>
> ---
> **Q3: In order to have equality in equation (4), the term
> $\nabla _{\theta } Q^{\pi _{\theta }} (s,a)$
> must be added.**
>
> i)
> Note that in our setting, the critic $Q_{w}^{\pi_{\theta }}$, and the actor $\pi_{\theta }$ are neural networks with trainable parameters $w$ and $\theta$ respectively (mentioned in the first paragraph in Sec. 3). Therefore,
> the actor and critic are differentiable, and
>   $\nabla_{\theta } Q_{w}^{\pi_{\theta }} (s,\pi_{\theta } (s))$ is well-defined. By the Leibniz integral rule,  our off-policy policy gradient can be obtained as:
>
>   \begin{equation*}
> \nabla_{\theta} E_{\rho (s)}\left[ Q_{w}^{\pi_{\theta}} (s,\pi_{\theta } (s))\right] =E_{\rho(s)}\left[\nabla_{\theta } Q_{w}^{\pi_{\theta}} (s,\pi_{\theta } (s))\right]
>  =E_{\rho (s)}\left[ \nabla_{\theta } \pi_{\theta } (s)\nabla_{a} Q_w^{\pi_{\theta }} (s,a)|{a=\pi_{\theta}(s)}\right],
>   \end{equation*}
>
>  which does not contain the term $\nabla _{\theta } Q^{\pi _{\theta }} (s,a)$. Detailed proofs can be found in Lemma 1, Appendix A.
>
> As pointed out by the reviewer, the off-policy DPG should be:
>
> $
> \nabla_{\theta }E_{\rho (s)}[Q^{\pi_{\theta }}(s,\pi_{\theta } (s))] = E_{\rho (s)}\left[\nabla_{\theta } \pi_{\theta } (s)\nabla_{a} Q^{\pi_{\theta }} (s,a)+\nabla_{\theta } Q^{\pi_{\theta }} (s,a)|a=\pi_{\theta } (s)\right].
> $
>
> While this is true,
> $ \nabla_{\theta }E_{\rho (s)}\left[ Q_{w}^{\pi_{\theta
>  }} (s,\pi_{\theta } (s))\right]$ is different from $\nabla_{\theta}E_{\rho (s)}[Q^{\pi_{\theta }}(s,\pi_{\theta } (s))]$, where $Q_{w}^{\pi_{\theta
>  }} (s,\pi_{\theta } (s))$ is a differentiable function  w.r.t $\theta$.
>
>
> ii) In fact, the derivation approach in [1] cannot be used on our neural-network-based AC.
> First, recall the two key steps in off-policy DPG:
> \begin{equation*}
>      \nabla _{\theta } Q^{\pi _{\theta }} (s,\pi _{\theta } (s))=\nabla _{\theta
> }\int _{A} [\pi (a|s)Q^{\pi _{\theta }} (s,a)]da \overset{(b)}{=}\int _{A} \nabla
> _{\theta } [\pi (a|s)Q^{\pi _{\theta }} (s,a)]da,
> \end{equation*}
>
> where $\pi(a|s)$ is the Dirac-Delta function defined in Sec. 3. In order to use
> the integral rule of interchanging the integration and differentiation ((b) in
> the above), the critic $Q_{w}^{\pi _{\theta }} (s,a) $ should be continuous
> w.r.t. $\displaystyle \theta $, $\displaystyle \forall \ a\in A$ [7]. However,
> in our case, it is only differentiable w.r.t. $\theta$ when $a=\pi_\theta(s)$.
> Thus, we cannot interchange $\nabla _{\theta }$ and $\int _{A}$. In this revised
> version, we have added more details about this in Remark 1,
> Appendix A.
>
>
> We also added a proof in Lemma 2, Appendix A, showing that our derived method to $\nabla_{\theta }E_{\rho(s)}[ Q_{w}^{\pi_{\theta }} (s,\pi_{\theta } (s))]$ is the same as the
>  derived method in [1] when $Q_{w}^{\pi _{\theta }} (s,a) $ is continuous
>  w.r.t. $\displaystyle \theta $, $\displaystyle \forall \ a\in A$.
>
> ---
> **Q4:  The definition of the action-value function should not depend
> on the initial state.**
>
> In the original submission, we followed the trajectory distribution in Eq.
> (2) of [6],
> which depends on the initial state distribution.
> As suggested,
> we rewrote the first paragraph of Sec. 3
> in this revised version.
> We
> now follow (Lillicrap et al., 2016; Silver et al., 2014)
> and the action-value function is defined as $Q^{\pi}(s,a) := \mathbb{E}_\pi[\sum_k
> \gamma^k r_k|s,a]$,  which
> does not depend on the initial state.

---

> > ### Author Response · Authors · 2021-11-22
> > **Response to Reviewer 6tEQ (Part 2/3)**
> >
> > (Continued)
> >
> > ---
> > **Q5:
> > The TD loss in this paper differs from DDPG, where the target
> > value function is deterministic, but the learned value function is evaluated on a
> > set of noisy actions.**
> >
> > We  agree that  the
> > TD loss
> > ($L(\theta,w) = E_{\rho(s)}\left[(Q_{w}^{\pi_{\theta }}(s, \pi_\theta(s))-\perp[Q^{\pi_{\theta}}(s, \pi_\theta(s))])^{2}\right]$)
> > used in our theoretical analysis
> > differs from the loss used in our implementation
> > ($\hat{L}(\theta,w)=E_{s\sim \rho(s), a\sim\pi(\cdot|s)}\left[(Q_{w}^{\pi_{\theta }}(s,
> > a)-\perp[Q^{\pi_{\theta}}(s, a)])^{2}\right]$),
> > where $\perp(\cdot)$ is the stop-gradient operator.
> > Note that $\hat{L}(\theta,w)$ is also the critic's loss for DDPG.
> > Therefore, as suggested, in the analysis  (Sec. 3) in this revised version, we revised the loss of the critic from $L(\theta,w)$ to $\hat{L}(\theta,w)$
> > so that the theoretical update scheme matches with the implementation
> > in Appendix I.
> >
> >
> > Specifically, we use $\nabla_w L( \theta ,w )$ to compute the
> > indirect gradient term $
> > \left(\frac{\partial
> > ^{2} L(\theta ,w)}{\partial \theta \partial w}\right)\left(\frac{\partial ^{2}
> > L(\theta ,w)}{\partial w^{2}}\right)^{-1}\frac{\partial J(\theta ,w)}{\partial
> > w}
> > $,
> > and use $\hat{L}( \theta ,w )$ to update the
> > critic.
> >
> > Thus, the revised update scheme is (Eq. (4), Sec. 3):
> >
> > $
> >  \theta  \leftarrow \theta +\alpha \frac{\partial J( \theta ,w^{*} (\theta
> > ))}{\partial \theta} , w\leftarrow w-\alpha_{w}\frac{\partial \hat{L}
> > (\theta ,w)}{\partial w},
> > $
> >
> > where
> > $\frac{\partial J\left( \theta ,w^{*} (\theta )\right)}{\partial \theta }
> > =\frac{\partial J(\theta ,w)}{\partial \theta } -\left(\frac{\partial ^{2}
> > L(\theta ,w)}{\partial \theta \partial w}\right)\left(\frac{\partial ^{2}
> > L(\theta ,w)}{\partial w^{2}}\right)^{-1}\frac{\partial J(\theta ,w)}{\partial
> > w}$.
> >
> > ---
> > **Q6: The authors must use the chain rule for both terms in the temporal
> > difference loss, while this would not be the case in standard DDPG.**
> >
> > We use
> > semi-gradient descent (Sutton & Barto, 2018) in $L(\theta,w) $ and $\hat{L}(\theta,w) $, which
> > drops the gradient of the second term ($Q^{\pi_{\theta}}(s, \pi_\theta(s))$). So, we only need to take the chain rule for $Q_{w}^{\pi_{\theta }}(s, \pi_
> > \theta(s))$. We added more details on this in Sec. 4.1, Lemma 3 in Appendix A, and Proposition 1, in Sec. 4.
> >
> > ---
> > **Q7: how this changes affect the theoretical results?**
> >
> > With the mentioned changes to the
> > update scheme above, in this revised version, we have also changed the definitions of the dynamics and the
> > Jacobian for FSDDPG
> > and SDDPG at the beginning of Sec. 5 and Appendix D, accordingly.
> >
> > For theoretical results
> > that
> > are do not involve the dynamics and the Jacobian
> > (Proposition 1 in Appendix A,  Lemmas 4-6 in Appendix B, and results in Tab. 1), they
> > remain unchanged.
> >
> > For the others
> > that involve the dynamics and the Jacobian
> > (Theorems 1-2 in Appendix D, Theorem 3 in Appendix E, and Theorem 4 in Appendix F),
> > by using the revised definitions of dynamics and the Jacobian for FSDDPG and SDDPG, and using the same assumptions to the new
> > dynamics and the Jacobian, we can still obtain the same results.
> >
> > Since the revised learning scheme is different from that in [Fiez et al., 2020],
> > we also added a new proposition (Proposition 2
> > in Appendix D)
> > to show the convergence of SDDPG.
> >
> >
> > For Corollary 1 in Appendix D, we now have three update schemes (Eq. (4)
> > for SDDPG, Eq. (7) for FSDDPG, and Eq. (2) for the Stackelberg dynamics in [Fiez
> > et al., 2020]). The original corollary only shows the relationship between SDDPG
> > and FSDDPG. Thus, we update it by showing that the optimal solution of SDDPG is
> > also the optimal solution of Eq. (2) and FSDDPG.

---

> > > ### Author Response · Authors · 2021-11-22
> > > **Response to Reviewer 6tEQ (Part 3/3)**
> > >
> > > (Continued)
> > >
> > > ---
> > > **Q8: In the proof of Proposition 1, why is the term $\displaystyle \nabla
> > > _{\theta } Q^{\pi _{\theta }} (s,\pi _{\theta } (s))$ disappearing?**
> > >
> > > We use
> > > semi-gradient method (Sutton & Barto, 2018) on $L$, which
> > > drops the gradient of the second term ($Q^{\pi_{\theta}}(s, \pi_\theta(s))$).
> > > Thus,
> > >
> > > $
> > >   \nabla_{\theta } Q^{\pi_{\theta }} (s,\pi_{\theta } (s)) = \nabla_{\theta } \perp [Q^{\pi_{\theta }} (s,\pi_{\theta } (s))] = 0,
> > > $
> > >
> > > where
> > >   $\perp$ is the stop-gradient operator.
> > > In this revised version, we added more details on this in Sec. 4.1 and Proposition 1.
> > >
> > > The reason for using the semi-gradient method to $L(\theta, w)$ is that
> > > $ \nabla_{\theta} Q^{\pi_{\theta}} (s,\pi_{\theta } (s))$ is not always well-defined.
> > > That is, $ \nabla_{\theta} Q^{\pi_{\theta}} (s,\pi_{\theta } (s))$
> > > exists only when
> > > the assumptions in Appendix A of [1]
> > > are satisfied. However, these assumptions may be strong since they require
> > > the rewards and transition functions to be differentiable w.r.t. $\theta $.
> > > In many popular environments (such as [Todorov et al., 2012] and the environment used in the
> > > experiments), they are non-differentiable, which also makes $\frac{\partial}{\partial \theta} E_{\rho(s)}\left[(Q_w^{\pi_{\theta}}(s, a)-Q^{\pi_{\theta}}(s, \pi_{\theta}( s)))^{2}\right]$ non-differentiable (Detailed proof can be found in Lemma 3, Appendix A). Thus, we use the semi-gradient technique here to
> > > make $ \nabla_{\theta}  L(\theta,w)$ well-defined.
> > >
> > >
> > > ---
> > > **Q9: "The work of Zheng et. al. [4]".**
> > >
> > > This paper is
> > > motivated from
> > > a preliminary version of [4], which was published in AAAI 2021 workshop [5].
> > > In this revised version, we now cite
> > > both [5] and [4] in Sec. 1.
> > >
> > > ---
> > > References
> > >
> > > [5] "Stackelberg Actor-Critic: A Game-Theoretic Perspective." AAAI Workshop on Reinforcement Learning in Games (2021).
> > >
> > > [6] Levine, Sergey. "Reinforcement learning and control as probabilistic inference: Tutorial and review." arXiv preprint arXiv:1805.00909 (2018).
> > >
> > > [7] https://planetmath.org/differentiationundertheintegralsign

---

> > > > ### Comment · Reviewer_6tEQ · 2021-11-29
> > > > **Response to authors**
> > > >
> > > > I thank the authors for their reply. I am not convinced by the argument of the authors for dropping the $\nabla_\{\theta} Q^{\pi_{\theta}}(s,a) $ term. Even if the maximization objective is defined using directly the function approximator for Q, this function approximator is implicitly a function of the policy parameters $\theta$, hence that part of the gradient should not be ignored. There is a large area of RL dealing with the issues caused by considering that term. The most popular ideas are using emphatic weights [1] or considering the distribution shift induced by the target policy [2].
> > > >
> > > > I think that defining the RL objective directly in terms of the function approximator has no valid justification since it can be shown [2,3] that the objective function maximized, even if Q is linear, is not the true RL objective
> > > >
> > > > [1] Richard S Sutton, A Rupam Mahmood, and Martha White. An emphatic approach to the problem of off-policy temporal-difference learning. The Journal of Machine Learning Research, 17(1): 2603–2631, 2016
> > > >
> > > > [2] Yao Liu, Adith Swaminathan, Alekh Agarwal, and Emma Brunskill. Off-policy policy gradient with state distribution correction. arXiv preprint arXiv:1904.08473, 2019
> > > >
> > > > [3] [3]Thomas Degris, Martha White, and Richard S. Sutton. Off-policy actor-critic. In Proceedings of the 29th International Conference on International Conference on Machine Learning, ICML’12, pages 179–186, USA, 2012.Omnipress

---

> > > > > ### Author Response · Authors · 2021-11-30
> > > > > **Further Response to Reviewer 6tEQ**
> > > > >
> > > > > Thank you so much for your further comments!
> > > > >
> > > > >
> > > > >
> > > > > Here are our responses:
> > > > >
> > > > > ---
> > > > >
> > > > > **Q10: Even if the maximization objective is defined using directly the function approximator for $\displaystyle Q$, this function approximator is implicitly a function of the policy parameters $\displaystyle \theta $, hence that part $\displaystyle \nabla _{\theta } Q^{\pi _{\theta }} (s,a)$ of the gradient should not be ignored. There is a large area of RL dealing with the issues caused by considering that term. The most popular ideas are using emphatic weights [1] or considering the distribution shift induced by the target policy [2].**
> > > > >
> > > > >
> > > > >
> > > > > We have answered similar questions in our responses to Q3 and Q7.
> > > > > The following we highlight some important parts to make the reviewer better know our ideas:
> > > > >
> > > > > i) Under our deterministic policy and function approximator critic setting,
> > > > >
> > > > >  $\nabla_{\theta } E_{\rho (s)}\left[ Q_{w}^{\pi_{\theta }}( s,\pi_{\theta } (s))\right] =E_{\rho (s)}\left[ \nabla_{\theta } \pi_{\theta } (s)\nabla_{a} Q_{w}^{\pi_{\theta }} (s,a)\mid a=\pi_{\theta } (s)\right]$
> > > > >
> > > > > is exact, according to the Lemma 1 in Appendix A. Thus, our gradient does not contain the term $\nabla _{\theta } Q^{\pi _{\theta }} (s,a)$.
> > > > >
> > > > >
> > > > > Regarding [1,2], they are orthogonal to our work. They mainly focus on off-policy policy gradient under the stochastic policy setting,
> > > > > while we focus on the deterministic policy and the function approximator critic setting.
> > > > >
> > > > > In our settings, due to the properties of the Dirac-Delta function, our derived gradient for
> > > > > $ \nabla_{\theta } E_{\rho (s)}\left[ Q_{w}^{\pi_{\theta }}( s,\pi_{\theta } (s))\right]$ is different from the off-policy policy gradient in [1,2].
> > > > > The reason has been discussed in Remark 1, Appendix A.
> > > > >
> > > > >
> > > > > ii) The reason  $\nabla_{\theta } Q^{\pi_{\theta }} (s,a)$ can not be used in Proposition 1 is that
> > > > > under our experiment settings (the rewards and transition functions are not differentiable w.r.t. $\displaystyle \theta $),
> > > > > $\nabla_{\theta } Q^{\pi_{\theta }} (s,a)$ is not well-defined and does not exist according to Lemma 3 in Appendix A.
> > > > > Thus, we leverage the semi-gradient method to avoid this problem.
> > > > >
> > > > > ---
> > > > > **Q11: I think that defining the RL objective directly in terms of the function approximator has no valid justification since it can be shown [2,3] that the objective function maximized, even if Q is linear, is not the true RL objective.**
> > > > >
> > > > > Defining the RL objective directly in terms of the function approximators is also used Eqs.(15-16) in [5] and Eqs. (36-37) in [8]. Also, it seems that [2,3] do not claim our function approximator objective can not be the true RL objective.
> > > > >
> > > > > In fact, if all the state $s$ and action $a$ have probability to be sampled, when $w^{*}$ minimizes the critic's loss, i.e.,
> > > > >
> > > > > $\hat{L} (\theta ,w^{*})$
> > > > >
> > > > > $ = E_{s\sim \rho (s), a\sim \beta(\cdot|s)}[ (Q_{w^{*}}^{\pi _{\theta }} (s,a)- Q^{\pi _{\theta }} (s,a) )^{2}] =0,$
> > > > >
> > > > > we have $\forall s,a$, $\displaystyle Q_{w^{*}}^{\pi _{\theta }} (s,a)=Q^{\pi _{\theta }} (s,a)$. Thus,
> > > > >
> > > > > $\max J( \theta ,w^*) =\max E_{\rho ( s)}\left[ Q_{w*}^{\pi_\theta} (s,a)\right] =\max E_{\rho ( s)}\left[ Q^{\pi_{\theta }} (s,a)\right] .$
> > > > >
> > > > > Therefore, it is equal to maximize the discount total reward, which is the RL objective.
> > > > >
> > > > > Our toy examples in Sec. 6.1 also show that our method does converge to the fixed points that maximize the reward functions.
> > > > >
> > > > > ---
> > > > > References:
> > > > >
> > > > > [8] Wen, Junfeng, et al. "Characterizing the Gap Between Actor-Critic and Policy Gradient." ICML, 2021.

---

### Author Response · Authors · 2021-11-22
**Response to all reviewers**

For all the reviewers:

We thank again all the reviewers for their constructive and valuable comments. We have uploaded a new version and summarize the changes below. Important changes are highlighted in blue.
Please check the details of the responses to the questions raised by reviewers in separated replies.

Summary of changes:

1. We added more details and discussion on the definition of $\rho$ in Sec. 3.

2. We added Lemma 1, Lemma 2 and Remark 1 to explain more on our off-policy
gradient: including why it is different from the original off-policy DPG, and under what conditions they are equal.

3. We revised the loss of critic
from $L(\theta,w)$ to $\hat{L}(\theta,w)$ to match our implementation.
We modify the corresponding proofs to make them consistent with the revised
update scheme. We also added Corollary 1 in Appendix D to provide more results on
the relationship among the update schemes.

4. We have used longer training episodes for InvertedDoublePendulum and Walker in Fig. 3, Sec. 5.

We hope our responses could address the reviewers' concerns. More discussions and suggestions on our paper are always welcomed!

---

### Decision · Program_Chairs · 2022-01-20

**Decision:**

Reject

**Comment:**

This paper aims to make Stackelberg Deep Deterministic Policy Gradients practical and efficient. The main contributions are an analysis which suggests terms involving the Hessian can be dropped and a block-diagonal approximation to an expensive matrix inversion.

Several reviewers who voted for rejection expressed concerns about the soundness of the theoretical arguments. The response provided by the authors did help alleviate some of the reviewers’ concerns but still left significant doubts. While some of the remaining concerns could be due to a misunderstanding of the deterministic setting it is up to the authors to convince the reviewers that their arguments are sound. Given the current scores and the low confidence of the reviewer voting for acceptance, I recommend rejecting the paper in its current form.